# Distributional Consistency Loss: Beyond Pointwise Data Terms in Inverse Problems

**George Webber & Andrew J. Reader**
School of Biomedical Engineering and Imaging Sciences
King's College London
`{george,andrew}.{webber,reader}@kcl.ac.uk`

## Abstract

Recovering true signals from noisy measurements is a central challenge in inverse problems spanning medical imaging, geophysics, and signal processing. Current solutions nearly always balance prior assumptions regarding the true signal (regularization) with agreement to noisy measured data (data-fidelity). Conventional data-fidelity loss functions, such as mean-squared error (MSE) or negative log-likelihood, seek pointwise agreement with noisy measurements, often leading to overfitting to noise. In this work, we instead evaluate data-fidelity collectively by testing whether the observed measurements are statistically consistent with the noise distributions implied by the current estimate. We adopt this aggregated perspective and introduce *distributional consistency (DC) loss*, a data-fidelity objective that replaces pointwise matching with distribution-level calibration. DC loss acts as a direct and practical plug-in replacement for standard data consistency terms: i) it is compatible with modern unsupervised regularizers that operate without paired measurement–ground-truth data, ii) it is optimized in the same way as traditional losses, and iii) it avoids overfitting to measurement noise without early stopping or the use of priors. Its scope naturally fits many practical inverse problems where the measurement-noise distribution is known and where the measured dataset consists of many independent noisy values. We demonstrate efficacy in two key example application areas: i) in image denoising with deep image prior, using DC instead of MSE loss removes the need for early stopping and achieves higher PSNR; ii) in medical image reconstruction from Poisson-noisy data, DC loss reduces artifacts in highly-iterated reconstructions and enhances the efficacy of hand-crafted regularization. These results position DC loss as a statistically grounded, performance-enhancing alternative to conventional fidelity losses for an important class of unsupervised noise-dominated inverse problems.

## 1 Introduction

Reconstructing a true signal from a single set of noisy measurements is a prevalent challenge in scientific computing (Vogel, 2002; Tarantola, 2005; Aster et al., 2018), with applications spanning medical imaging (Bertero et al., 2021; McCann et al., 2017; Ribes and Schmitt, 2008; Louis, 1992), remote sensing (Efremenko and Kokhanovsky, 2021; Baret and Buis, 2008), time-series denoising (Gubbins, 2004), astronomical data analysis (Craig and Brown, 1986; Lucy, 1994), and geophysical inversion (Parker, 1994; Linde et al., 2015). In these problems, the reconstruction objective typically combines a data-fidelity term with a regularizer: data-fidelity enforces consistency with the measurement process and noise model, while regularization encodes prior structure in the signal.

While regularization has evolved substantially (Habring & Holler, 2024), the underlying data-fidelity terms have changed little. Standard objectives such as mean squared error (MSE) or negative log-likelihood (NLL) penalize residuals point-by-point. Optimizing these pointwise objectives encourages matching individual noise realizations rather than ensuring that the measurements are statistically compatible with the model. While early stopping or discrepancy-based rules can mitigate this behavior, they require explicit tuning and do not change the objective itself. As a result, under a noisy realization, the ground-truth signal is not a minimizer of the pointwise data term, and regularization must compensate for noise fitting rather than focusing solely on structure.

We address this limitation directly by reframing the data-fidelity objective. We introduce distributional consistency (DC) loss, which evaluates whether the measurements are collectively consistent with the noise distributions implied by a candidate reconstruction. Each measurement is mapped to its percentile value under the cumulative distribution function (CDF) of its predicted noise model, and the resulting collection of percentile values is compared to the uniform distribution expected under a correct model (Figure 1). Instead of reducing individual residuals, DC loss tests whether the entire noisy dataset behaves like a typical draw from the assumed noise process.

DC loss applies whenever the noise distribution is known or can be estimated, including heteroscedastic settings such as Poisson counts or Gaussian noise with known variance. It requires only one noise realization per measurement but many independent measurements overall. This setting is satisfied in common inverse problems such as pixel-wise imaging (Fan et al., 2019), dense time-series sampling (Gubbins, 2004), and tomographic reconstruction (Arridge et al., 2019).

CONTRIBUTIONS:

- We introduce DC loss as a new principled data-fidelity term built from distributional calibration. Evaluating DC loss is simple and compatible with auto-differentiation frameworks.

- We analyze how DC loss compares to pointwise objectives: for parameter estimates that are far from the solution, DC loss exhibits MSE/NLL-like convergence, while near the solution it removes the incentive to fit noise, enabling stable prolonged optimization.

- Across the settings we study, DC loss improves practical reconstructions: (a) Deep Image Prior (DIP) denoising (Ulyanov et al., 2020) with clipped Gaussian noise, where DC loss removes the need for early stopping and yields higher peak PSNR than MSE; (b) PET image reconstruction under a Poisson model (Gourion and Noll, 2002), where DC loss reduces noise artifacts at high iteration numbers and pairs well with TV regularization, achieving superior noise–detail trade-offs at much smaller regularization strengths.

- We demonstrate the real-world applicability of DC loss through experiments on real 3D PET brain data.

## 2 BACKGROUND

### 2.1 PROBLEM FORMULATION

We consider inverse problems where a latent signal is mapped through a known forward model and corrupted by stochastic noise drawn from a known distribution. No paired measurement–ground-truth data are available for learning, though a pre-trained regularizer may be provided.

Let unknown parameters $\boldsymbol{\theta}^* \in \Theta$ be mapped by a known forward operator $f : \Theta \to \mathbb{R}^N$ to a noise–free signal (or mean data) $\mathbf{y} = f(\boldsymbol{\theta}^*) = (y_1, \ldots, y_N)$. Measurements $\mathbf{m} = (m_1, \ldots, m_N)$ are then modeled as noisy draws from known, per–index likelihoods tied to that signal,

$$m_i \sim \mathcal{D}_i(y_i) = \mathcal{D}_i\big(f(\boldsymbol{\theta}^*)_i\big), \tag{1}$$

where each family $\mathcal{D}_i$ has $y_i$ as its single unknown parameter (e.g., Poisson rate, Gaussian mean with known variance). Our goal is to recover an estimate $\hat{\boldsymbol{\theta}}$ close to $\boldsymbol{\theta}^*$, given $\mathbf{m}$, $f$, and $\{\mathcal{D}_i\}_{i=1}^N$.

Classical data–fidelity terms enforce *pointwise* agreement between $f(\hat{\boldsymbol{\theta}})$ and $\mathbf{m}$, which can promote overfitting in expressive models by tracking individual noise realizations. In Section 3, we instead assess *distributional* agreement: given a candidate $\hat{\mathbf{y}} = f(\hat{\boldsymbol{\theta}})$, we ask whether the measurements $\mathbf{m}$ are statistically consistent with the predicted noise models $\{\mathcal{D}_i(\hat{y}_i)\}_{i=1}^N$.

### 2.2 RELATED WORK

While most progress in inverse problems has focused on regularization, some works have considered the design of data-fidelity terms. Robust variants such as the Huber and Student's t losses (Mohan et al., 2015; Kazantsev et al., 2017) can reduce outlier influence but do not prevent noise-fitting under a correct noise model. Stein's unbiased risk estimate (Metzler et al., 2018) penalizes the divergence of the estimator to control its sensitivity to noise, effectively regularizing the architecture rather than

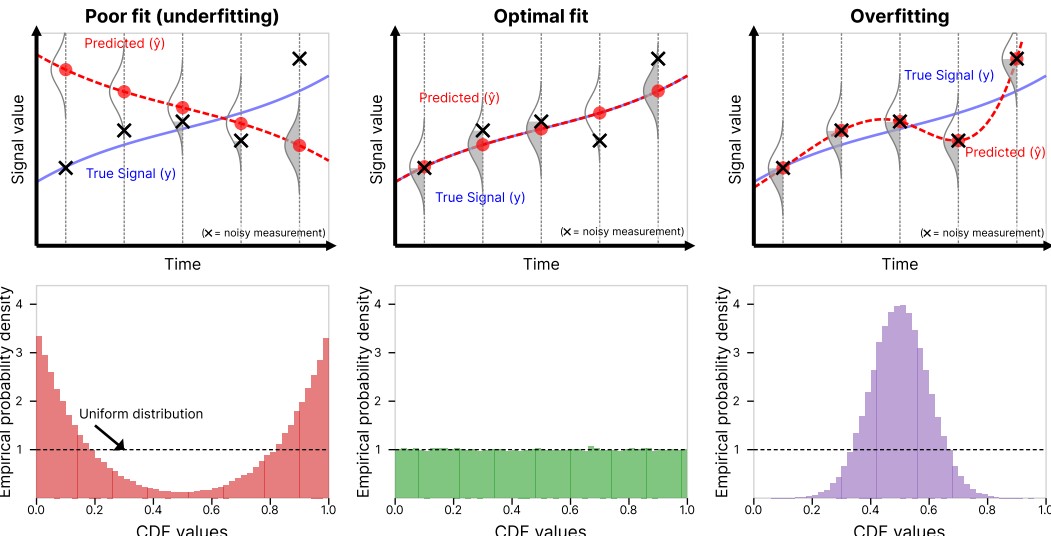

Figure 1: Illustration of the proposed DC loss. Top: predicted signals illustrating under-, well- and over-fitting to noisy measurements. Bottom: empirical CDF-value histograms for each case (assuming many predicted points). Poor fit histograms skew towards 0 or 1; good fits give uniform histograms; overfitting gives a sharp peak near 0.5. DC loss penalizes departures from uniformity.

altering the underlying incentive to match the observed noise. DC loss operates in a different regime from Noise2Noise (Lehtinen et al., 2018), which requires multiple noisy observations, or end-to-end trained approaches (Heckel, 2025), which require paired measurement–ground-truth data.

DC loss is instead linked most strongly to classical goodness-of-fit tests such as the Kolmogorov–Smirnov (K-S) and Cramér–von Mises (CvM) criteria (Kolmogorov, 1933; Cramér, 1928) as well as related work in emission tomography (Llacer et al., 1989). The K-S and CvM formulations provide post-hoc assessments of goodness-of-fit using $L^\infty$ and $L^2$ discrepancies between empirical and theoretical distributions. DC loss may be viewed as a optimization-friendly analogue of these tests: by changing the metric, we obtain a smooth, differentiable objective suitable for optimization rather than post-hoc assessment. This avoids the limitations of post-hoc stopping criteria in iterative reconstruction, such as spatially variant convergence and ambiguity in determining when a reconstruction is complete. This mechanism parallels the stabilization achieved by semi-convergence and early-stopping methods (Elfving et al., 2014) but embeds it directly in the loss, allowing data fidelity and regularization to interact synergistically.

## 3 METHOD

### 3.1 FROM POINTWISE TO DISTRIBUTIONAL CONSISTENCY

The *Probability Integral Transform* (PIT) provides a simple way to test whether a model's predicted distributions are statistically consistent with observed data. For each measurement $m_i$ and its corresponding predicted noise distribution $\mathcal{D}_i(\hat{y}_i)$, we compute the cumulative probability

$$s_i = F_i(m_i \mid \hat{y}_i) = \mathbb{P}_{c \sim \mathcal{D}_i(\hat{y}_i)}(c \le m_i).$$

Each value $s_i$ represents the *percentile position* of $m_i$ within its predicted distribution. The PIT says, that if the model is correctly calibrated, these percentiles are uniformly distributed on $[0, 1]$: about 10% of data fall below the 10th percentile, 50% below the median, and so on[1]. Departures from uniformity therefore indicate systematic mismatch between the model and data.

Figure 1 illustrates this idea. For a predicted signal $\hat{\mathbf{y}}$ (red curve), we evaluate the CDF of each noise model $\mathcal{D}_i(\hat{y}_i)$ (vertical Gaussians) at the corresponding measurement $m_i$ (black crosses). Collecting

---

[1]See Appendix A.1 for a full mathematical statement of this result.

---

**Algorithm 1** Generalized Distributional Consistency Loss

---

**Require:** Estimate $\hat{\boldsymbol{\theta}} \in \Theta$, observed data $\mathbf{m}$, number of measurements $N$, CDFs $F_1, \cdots, F_N$ corresponding to noise distributions $\mathcal{D}_1, \cdots, \mathcal{D}_N$, forward operator $f : \Theta \rightarrow \mathbb{R}^N$

1: $\hat{\mathbf{y}} \leftarrow f(\hat{\boldsymbol{\theta}})$
2: $\mathbf{s} \leftarrow (F_1(m_1|\hat{y}_1), \ldots, F_N(m_N|\hat{y}_N))$         ▷ Evaluate CDF of model at observed data
3: $\mathbf{r} \leftarrow \text{logit}(\mathbf{s})$                    ▷ Apply logit transform
4: Sample $\mathbf{u} \sim \text{Logistic}(0, 1)^N$        ▷ Generate $N$ samples from target distribution
5: Sort $\mathbf{r}$ and $\mathbf{u}$ in ascending order
6: $\mathcal{L}_{\text{DC}}(\hat{\boldsymbol{\theta}}) \leftarrow \frac{1}{N} \sum_{i=1}^{N} |r_i - u_i|$             ▷ Compute Wasserstein-1 distance
7: **return** $\mathcal{L}_{\text{DC}}(\hat{\boldsymbol{\theta}})$

---

these $s_i$ values yields an empirical distribution whose shape reflects how well the model explains the data:

- **Underfitting:** a histogram with peaks near 0 or 1 shows that most measurements lie in improbable regions of their predicted distributions $\mathcal{D}_i(\hat{y}_i)$.

- **Well-calibrated:** a uniform histogram shows that each quantile contains its expected share of data.

- **Overfitting:** a histogram with a sharp peak near 0.5 shows that most measurements lie near the center of their predicted noise distribution.

This motivates the concept of *distributional consistency*: a prediction is distributionally consistent when the empirical distribution of its CDF values is uniform. The next subsection formalizes this intuition into a differentiable loss.

## 3.2 DISTRIBUTIONAL CONSISTENCY LOSS FORMULATION

Given a current parameter estimate $\hat{\boldsymbol{\theta}}$, we evaluate each measurement $m_i$ under its predicted noise distribution $\mathcal{D}_i(f(\hat{\boldsymbol{\theta}})_i)$ via its cumulative probability

$$s_i := F_i(m_i \mid f(\hat{\boldsymbol{\theta}})_i) = \mathbb{P}_{c \sim \mathcal{D}_i(f(\hat{\boldsymbol{\theta}})_i)}(c \leq m_i). \tag{2}$$

When the model is well-calibrated, the PIT guarantees that these percentiles $s_i$ follow a uniform distribution on $[0, 1]$. We therefore aim to make the empirical distribution of $\mathbf{s} = (s_1, \ldots, s_N)$ as close to uniform as possible.

However, directly matching $\mathbf{s}$ to $\text{Uniform}[0, 1]$ often leads to vanishing gradients: when a predicted mean is far from $m_i$, $s_i$ saturates near 0 or 1, giving $\frac{\partial s_i}{\partial \theta_j} \approx 0$. To address this, we apply the logit transform (inverse sigmoid)

$$r_i = \text{logit}(s_i) = \ln\left(\frac{s_i}{1 - s_i}\right), \tag{3}$$

which stretches the endpoints to $\pm\infty$ and preserves gradient sensitivity. This maps the uniform target to a $\text{Logistic}(0, 1)$ distribution.[2] To measure the discrepancy between the empirical $\{r_i\}$ and the $\text{Logistic}(0, 1)$ reference, we compute the Wasserstein-1 (Earth Mover's) distance:

$$\mathcal{L}_{\text{DC}}(\hat{\boldsymbol{\theta}}) = \frac{1}{N} \sum_{i=1}^{N} |r_i - u_i|, \tag{4}$$

where $\{u_i\}_{i=1}^N$ are sorted samples from $\text{Logistic}(0, 1)$.

This formulation encourages $\hat{\boldsymbol{\theta}}$ to produce predictions whose implied noise distributions make the observed data statistically typical, rather than merely close in value, offering a principled alternative to pointwise losses. Algorithm 1 summarizes the steps taken to calculate DC loss.

---

[2]See Appendix A.2 for a derivation.

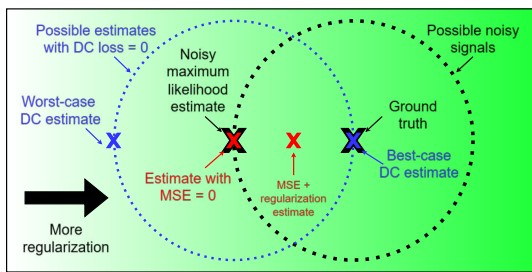

Figure 2: Interaction of DC loss and MSE loss with regularization. Crosses denote the predicted noise-free values $\hat{\mathbf{y}}$ implied by different parameter settings. With MSE loss, the estimate is pulled toward the noisy maximum-likelihood estimate (MLE); adding regularization forces a compromise between data-fidelity and regularity. With DC loss, many estimates that explain the data attain (near-)zero loss, forming a manifold around the MLE; adding regularization then simply selects among these data-consistent solutions rather than trading fidelity for regularity.

## 4 RELATION TO POINTWISE ERROR FUNCTIONS

### 4.1 EARLY-STAGE OPTIMIZATION BEHAVIOR

To avoid numerical precision difficulties, we may wish to use an approximation that directly estimates $r_i$, avoiding the need to compute the CDF value $s_i$ and then apply the logit as separate steps (i.e. combining Steps 1 and 2 in Algorithm 1). For example, when $\mathcal{D}_i(\hat{y}_i) = \mathcal{N}(\hat{y}_i, \sigma^2)$, if $\frac{|m_i - \hat{y}_i|}{\sigma}$ is large then $s_i = \mathbb{P}_{c \sim \mathcal{N}(\hat{y}_i, \sigma^2)}(c \leq m_i)$ may be exactly 0 or 1 under float precision. To resolve this issue, we may calculate $r_i = \mathrm{logit}(s_i)$ in one step, using appropriate approximations for $s_i$ and logit. When $s_i \approx 1$ (i.e. $m_i \gg \hat{y}_i$), using the Laplace tail approximation to a Gaussian distribution we obtain:

$$r_i = \mathrm{logit}(s_i) \approx \frac{(m_i - \hat{y}_i)^2}{2\sigma^2} + \ln\left(\frac{m_i - \hat{y}_i}{\sigma}\right) + \ln(\sqrt{2\pi}) . \tag{5}$$

Note that when predicted value $\hat{y}_i$ is much smaller than the measured value $m_i$, $r_i$ is dominated by the squared term in the approximation. A derivation and the $s_i \approx 0$ case are given in Appendix C.

Now consider the DC loss contribution for index $i$ and the gradient it gives to $\hat{y}_i$ when $\hat{y}_i$ is far from $y_i$. The per-sample term is $|r_i - u_i|$, where $r_i = \mathrm{logit}(F_i(m_i \mid \hat{y}_i))$ and $u_i$ is the corresponding sorted reference value from $\mathrm{Logistic}(0, 1)$. When $\frac{|m_i - \hat{y}_i|}{\sigma}$ is large, $|r_i|$ dominates the bounded $u_i$, so $|r_i - u_i| \approx |r_i|$ and the gradient w.r.t. $\hat{y}_i$ is governed by $\partial r_i / \partial \hat{y}_i$. Using the Gaussian tail approximation in equation 5, $\partial r_i / \partial \hat{y}_i \approx -(m_i - \hat{y}_i)/\sigma^2$. Thus, **far from the solution, DC loss provides essentially the same pointwise update direction as MSE on the noisy measurement**, with differences emerging only near the data-consistent regime.

A similar result holds for Poisson noise, with DC loss exhibiting similar behavior to the negative Poisson log-likelihood function when $\hat{y}_i$ is far from the true signal value $y_i$ (see Appendix C).

### 4.2 LATE-STAGE OPTIMIZATION BEHAVIOR & INTERACTION WITH REGULARIZATION

We have shown that DC loss behaves similarly to MSE/NLL when a given estimate is far from the maximum likelihood estimate. Now, we consider how DC loss behaves close to the true solution.

Consider the simple case where the forward operator $A$ is $\mathbf{I}$ and the noise model is Gaussian noise with variance $\sigma^2$. Then, the ground truth noise-free measurements $\mathbf{y}$ are exactly the true parameters $\boldsymbol{\theta}^*$. Let the noise vector be $\mathbf{n} \sim \mathcal{N}(\mathbf{0}, \sigma^2)$. Then measurements $\mathbf{m}$ are simply $\mathbf{m} = \boldsymbol{\theta}^* + \mathbf{n}$.

In this setting, a "worst-case" for DC loss minimization occurs when the optimizer attributes the noise with the *wrong sign*, i.e. predicting $-\mathbf{n}$ instead of $\mathbf{n}$. Because DC loss enforces distributional (not pointwise) agreement, this estimate attains the same DC loss as the ground truth. Then,

$$\hat{\boldsymbol{\theta}}_{\text{worst-DC}} = \mathbf{m} - (-\mathbf{n}) = \mathbf{m} + \mathbf{n} = \boldsymbol{\theta}^* + 2\mathbf{n} . \tag{6}$$

Hence, in the worst case, our $L_2$ error is $||2\mathbf{n}||_2$. Our best case is, of course, **zero** $L_2$ error. In comparison, using MSE as the data-fidelity term leads directly to the noisy maximum likelihood

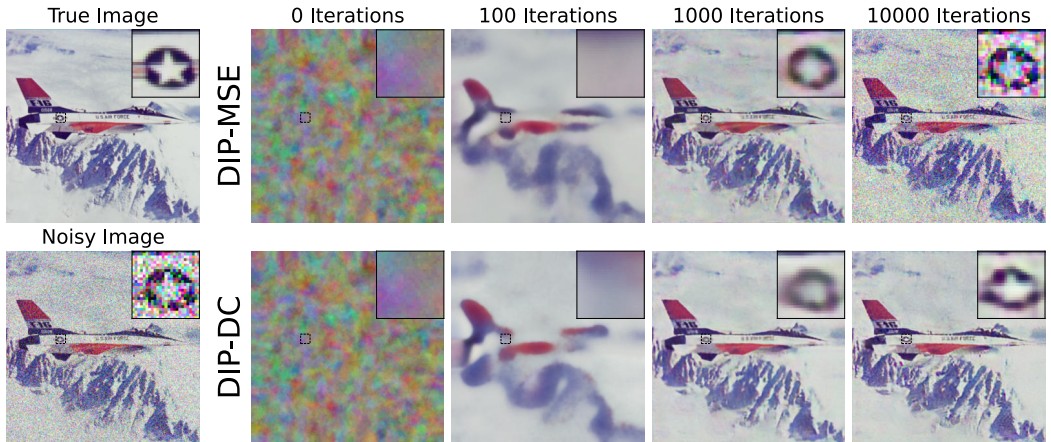

Figure 3: Images denoised with DIP over time ($\sigma = \frac{75}{255}$). Top row: DIP with MSE loss; bottom row: DIP with DC loss. DIP-MSE converges towards the noisy target; DIP-DC remains stable and does not display the same late-iteration degradation.

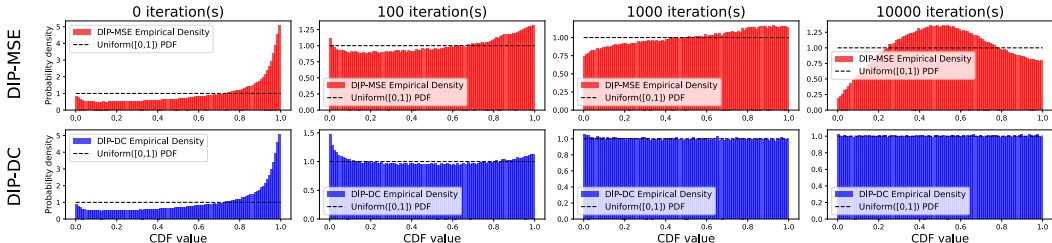

Figure 4: Histograms of the CDF values of measured data ($s_i$ values from Section 3.2) from performing DIP with MSE (top) and DC loss (bottom) as iteration increases ($\sigma = \frac{75}{255}$).

estimate, and a guaranteed $L_2$ error of $||\mathbf{n}||_2$. Figure 2 summarizes the conclusion from this analysis, namely that DC loss *can but is not guaranteed* to deliver a lower error estimate than MSE loss, and that the quality of estimate depends also on the prior assumptions introduced and optimization path.

In the following section, we demonstrate this behavior empirically in different unsupervised inverse problem settings.

# 5 APPLICATIONS

## 5.1 DEEP IMAGE PRIOR (DIP) FOR GAUSSIAN DENOISING

DIP (Ulyanov et al., 2020) is a popular method for unsupervised image denoising. In DIP, a randomly initialized convolutional neural network is given a fixed pure-noise input and trained, with an MSE objective, to reproduce the noisy target image. The network's inductive bias encourages the recovery of structured content, enabling unsupervised denoising, provided early stopping of training iterations is used to prevent overfitting the noise in the target image (Wang et al., 2023).

Instead of the standard MSE loss, we investigated using our DC loss with DIP to avoid overfitting without necessitating early stopping. In our experiments, we followed the method (including the neural architecture) of Ulyanov et al. (2020). Appendix E gives the full experimental details.

In Figure 3, we show the result of the DIP method using our DC loss (DIP-DC) and MSE (DIP-MSE). It is clear that after 1,000 iterations DIP-MSE begins to overfit to noise. By 10,000 iterations, this has resulted in severe degradation of the image quality, with the noise spikes from the noisy image visible in the DIP-MSE image. In contrast, the DIP-DC image suffers no such degradation.

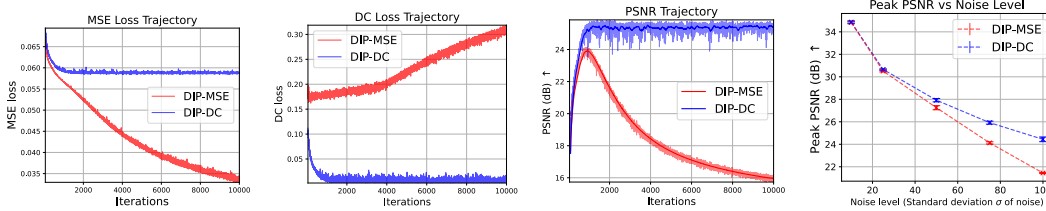

(a) Loss trajectories of MSE loss during DIP.

(b) Loss trajectories of DC loss during DIP.

(c) PSNR over time for $\sigma = \frac{75}{255}$ case.

(d) Peak PSNR achieved as $\sigma$ varies.

Figure 5: DIP-MSE and DIP-DC losses and metrics. Error bars were calculated as $1.96\times$ standard deviation on metrics over 5 random initializations of noise and network parameters.

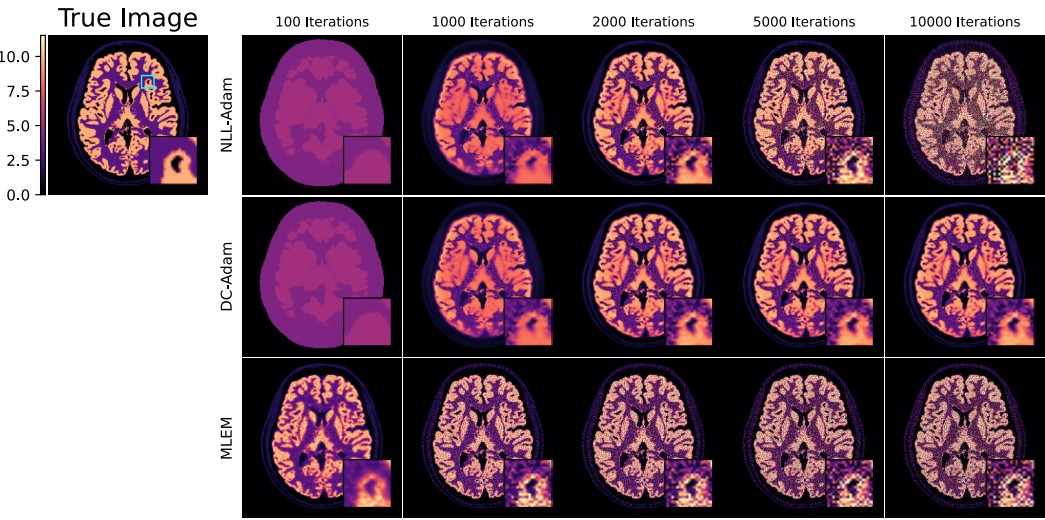

Figure 6: PET image reconstructions formed with three iterative algorithms (NLL-Adam, DC-Adam and MLEM) shown with increasing iteration number. The proposed DC loss (row 2) clearly avoids the noise overfitting with NLL as an objective, whether with Adam optimization (row 1) or with the MLEM optimization algorithm (row 3).

This effect is also seen in the histograms of $s_i$ values (Figure 4) and loss trajectories (Figure 5) for each case. Figure 5a demonstrates that the MSE loss tends to zero for DIP-MSE, while the MSE loss converges to a non-zero value for DIP-DC. Similarly, in Figure 5b, DC loss tends towards zero for DIP-DC, and converges to a value away from zero for DIP-MSE.

Further, in Figure 5c we plot PSNR during optimization. As is standard in DIP, MSE requires carefully optimised early stopping to avoid performance degradation. DIP-DC, by contrast, reaches its optimum PSNR and then plateaus without such intervention. Importantly, we compare against optimally early-stopped DIP-MSE (i.e., peak PSNR), and still observe that **DIP-DC achieves higher peak PSNR**. Figure 5d shows this trend persists across noise levels, with larger gains at higher $\sigma$. We hypothesize that MSE begins fitting noise before finer details can fully benefit from the inductive prior, whereas DC reduces this incentive and stabilizes optimization near the data-consistent regime.

In Appendix E, we show further results and trends with varied $\sigma$ values, as well as results evaluating the effect of misspecifying the noise model.

## 5.2 TOMOGRAPHIC IMAGE RECONSTRUCTION FROM POISSON NOISY MEASUREMENTS

Medical imaging features many noisy inverse problems, of which a notable example is *positron emission tomography* (PET). In PET image reconstruction, a linear forward model $\mathbf{A}$ maps the

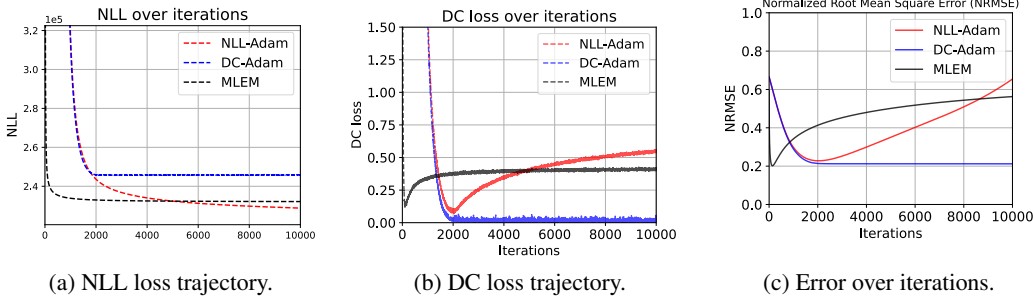

(a) NLL loss trajectory.  (b) DC loss trajectory.  (c) Error over iterations.

Figure 7: PET reconstruction losses and metrics. As iterations increase, only DC loss avoids noise-chasing behavior that leads NLL-Adam and MLEM towards worse error.

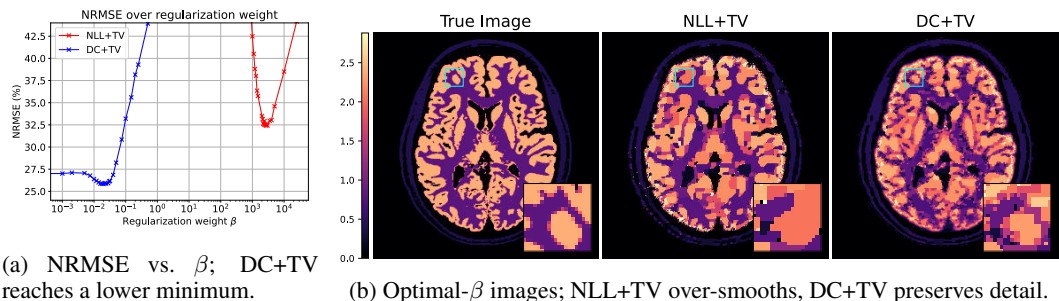

(a) NRMSE vs. $\beta$; DC+TV reaches a lower minimum.  (b) Optimal-$\beta$ images; NLL+TV over-smooths, DC+TV preserves detail.

Figure 8: Edge-preserving TV-regularized PET image reconstruction with different data terms. Under NLL, the regularizer must both suppress noise-fitting *and* impose structure; with DC loss, noise-fitting is already controlled, so the regularizer can concentrate on promoting desirable properties. This yields lower error and better detail at smaller $\beta$ for DC+TV than for NLL+TV.

unknown activity image $\hat{\boldsymbol{\theta}}$ to the ideal sinogram $\mathbf{y} = \mathbf{A}\hat{\boldsymbol{\theta}}$. The detector then records counts $\mathbf{m} \sim \mathrm{Poisson}(\mathbf{y})$, introducing Poisson measurement noise that reconstruction algorithms must handle.

Clinical PET scans meet DC loss's assumptions of a known noise model and a large number of independent measurements. We investigate the use of DC loss for PET image reconstruction from noisy measurement data. Existing unregularized algorithms optimize an image estimate with respect to the negative Poisson log-likelihood (NLL):

$$\mathrm{NLL}(\hat{\boldsymbol{\theta}}|\mathbf{m}) = \sum_{i=1}^{N} -m_i \ln([\mathbf{A}\hat{\boldsymbol{\theta}}]_i) + [\mathbf{A}\hat{\boldsymbol{\theta}}]_i + \ln(m_i!) \ . \tag{7}$$

Clinically-used algorithms for PET reconstruction, such as MLEM (maximum-likelihood expectation-maximization) (Shepp and Vardi, 1982), use early stopping on the number of iterations to avoid overfitting, which manifests as noise spikes and other image artifacts (Jonsson et al., 1998).

Using a BrainWeb phantom (Cocosco et al., 1997), we modeled the PET reconstruction process in 2D with a single ring of a cylindrical PET scanner (Schramm and Thielemans, 2023). We then optimized an image $\hat{\boldsymbol{\theta}}$ with respect to the NLL and DC loss using Adam (Kingma and Ba, 2015) (with learning rate $5 \times 10^{-3}$), as well as using the clinically-relevant MLEM algorithm for comparison.

In Figure 6, we show the visual results from each algorithm over 10,000 iterations. We see that both MLEM and NLL-Adam overfit to noise in the image, with varying convergence rates. We further see that DC-Adam does not overfit the noise in the data. This is also observed in Figure 7a, where the NLL of the DC-Adam image plateaus after 2,000 iterations, and in Figure 7b where the NLL-Adam and MLEM images do not converge to zero DC loss. In Figure 7c, we see that DC-Adam converges at the minimum error of NLL-Adam, although MLEM achieves a slightly better minimum error than both (likely due to the non-negativity constraint in the MLEM algorithm).

In this setting the problem is less over-parameterized than in Sections 5.1 and Appendix D, so perfectly fitting the noise is harder. Even so, DC loss delivers a clear advantage: it **converges to its**

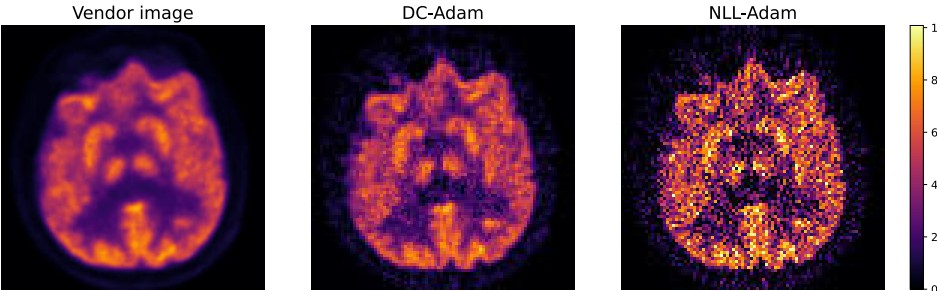

Figure 9: An axial slice from reconstruction of a real 3D PET dataset with DC-Adam or NLL-Adam, demonstrating the stable late-iteration behavior of DC loss as observed in synthetic experiments.

**best solution and stays there**, rather than briefly peaking and then chasing noise. In practice this yields stable late-iteration behavior without early stopping, fewer noise artifacts in highly iterated images, and better agreement with the assumed noise model. By contrast, conventional fidelity terms tend to overfit as iterations progress. Additional setup details and experiments on the impact of over-parameterization are provided in Appendix F.

### 5.2.1 REGULARIZATION WITH DC LOSS

Regularization is commonly used in medical image reconstruction to compensate for the noise–fitting tendency of standard data-fidelity terms. We therefore augmented the PET image reconstruction objective with a regularization penalty, which introduces the regularization strength hyperparameter $\beta$

$$\ell(\hat{\boldsymbol{\theta}}, \mathbf{m}) = NLL(\hat{\boldsymbol{\theta}}|\mathbf{m}) + \beta \cdot TV(\hat{\boldsymbol{\theta}}) . \tag{8}$$

For this experiment, we reconstructed from $4\times$ lower count data while using an edge-preserving variant of total variation (TV) as the regularization term (details in Appendix F.4).

DC+TV achieved better quantitative accuracy and stronger noise suppression at its optimum $\beta$ than NLL+TV, as shown by Figure 8. Very notably, even with little to no regularization, DC loss still delivers low NRMSE results. In contrast, as shown in Figure 8b, NLL+TV required substantially greater regularization to counter noise fitting; the reconstruction with lowest NRMSE is noticeably smoother and less detailed than its DC+TV counterpart. Notably, the optimal $\beta$ for DC+TV also was orders of magnitude smaller than for NLL+TV (Figure 8a); this aligns with our hypothesis that DC loss doesn't place regularization in opposition to data-fidelity (see Figure 2).

By curbing noise fitting at the data-fidelity term, DC loss enables a more favorable trade-off: weaker regularization suffices to suppress noise without sacrificing detail.

### 5.2.2 REAL PET FEASIBILITY STUDY

To assess the practical feasibility of DC loss on real measurement data, we applied it to a clinical 3D PET brain scan acquired on a Siemens Biograph mMR system. The dataset comprises over 70 million lines of response, and reconstruction uses a full physical forward model including attenuation, detector normalization, scatter, and randoms. An axial slice comparing the vendor reconstruction with late-iteration DC-Adam and NLL-Adam reconstructions (10,000 iterations, 21 subsets) is shown in Fig. 9; full reconstruction details are provided in Appendix F.5.

In this clinical dataset, DC loss yields a stable late-iteration reconstruction, whereas NLL-Adam displays the characteristic high-frequency amplification commonly observed at large iteration counts. This experiment therefore demonstrates that DC loss is operationally feasible on real clinical PET data and behaves as expected under realistic measurement and modelling conditions.

This experiment is intended as a first demonstration of real-world applicability. In practice, the regimes where overfitting becomes most problematic are lower-dose acquisitions or high-resolution reconstructions, where stronger priors are typically used. Those settings provide natural next targets for evaluating DC loss in even more challenging scenarios.

# 6    DISCUSSION

Across the settings we have studied, DC loss initially tracks pointwise losses but then shifts trajectory and converges once the measurements are statistically explained, at which stage, in contrast, further updates with pointwise losses mostly chase noise rather than signal. This behavior matches the theory in Section 4: once the set of CDF values is close to the uniform target, DC no longer rewards per-sample noise fitting.

Crucially, combining DC with regularization, whether implicit (e.g., DIP) or explicit (e.g., edge-preserving TV), yields stronger results than pointwise losses, because DC reduces the tension between data-fidelity and regularization; the fidelity term stops pulling toward noise, and the regularizer can preserve structure instead of fighting noise fitting.

As discussed in Section 4.2, DC loss effectively defines an equivalence class of solutions: any prediction that induces an (approximately) uniform distribution of CDF values across measurements attains low DC loss. This flattens the objective near good solutions. While this guards against noise-chasing, it does not by itself distinguish between plausible and implausible signals within the data-consistent regime; architectural priors or explicit penalties provide that structure.

We expect DC loss to be most useful (a) in over-parameterized regimes where many viable solutions exist and noise fitting is a concern, and (b) when paired with priors that encourage interpretable or realistic outputs. In the main text, we have shown DC loss integrating well with DIP and handcrafted priors. We further demonstrate applicability to 1D noisy deconvolution in Appendix D and noisy image deblurring with plug-and-play priors (Venkatakrishnan et al., 2013) in Appendix G; we also anticipate compatibility with score-based generative models for inverse problems (Song et al., 2022).

## 6.1    LIMITATIONS AND SCOPE

DC loss assumes independent measurements and a known (or estimable) noise model per measurement, and it benefits from a large enough set of measurements to reliably assess collective residual behavior. These conditions hold in many large-scale inverse problems (e.g., imaging, tomography, spatial sensing) but may limit applicability in small-data regimes or when noise is poorly characterized. Likewise, DC is not designed for settings where the ill-posedness of the forward operator, rather than noise, is the main challenge, such as inpainting or problems with large null spaces, where overfitting noisy measurements is not a primary concern. For discrete noise models (e.g., Poisson), exact uniformity of CDF values need not hold; the randomized probability integral transform could be employed (Appendix A.3). Computationally, DC loss adds overhead time relative to pointwise methods (Appendix H), although in our experiments this was not a bottleneck.

Finally, because DC loss tolerates a family of statistically valid solutions, it does not, on its own, enforce structural properties such as regularity, sparsity, or coherence. As with pointwise losses, priors or regularizers remain important to steer reconstructions toward plausible signals.

## 6.2    FUTURE WORK

We focused on DC loss paired with non-learned regularization to isolate the behavior of the fidelity term itself. Appendix G shows that similar benefits also appear in a learned plug-and-play setting, suggesting that DC loss should extend naturally to learned priors. Further work should explore this integration more fully and study DC loss across additional operators, noise models, and problems.

# 7    CONCLUSION

We introduced distributional consistency (DC) loss, which reframes the data term in inverse problems as statistical consistency with the noise model across many independent measurements in a single noisy dataset. The central advance is that DC loss removes the incentive for noise-chasing in the data term, so optimization no longer rewards fitting a particular noise realization and regularization can focus on structure rather than suppressing noise. Empirically, in DIP denoising and PET image reconstruction, DC loss reduced reliance on early stopping and improved noise–detail trade-offs, consistent with this picture. DC loss is simple to implement, compatible with standard priors, and provides a practical data-fidelity term for many inverse problems with known noise models.

## REPRODUCIBILITY

To ensure the reproducibility of this work, further experimental details are listed in Appendices D.2, E.1, F.1 and F.4. Additionally, Appendix C details the approximations used to enable practical implementation of DC loss, with algorithm pseudocodes given as Algorithms 2, 3 and 4. Appendices A and B also give more mathematical derivations used to support the formulation of DC loss.

Source code for the experiments is available at: `https://github.com/GeorgeWebber/Distributional-Consistency-Loss`.

## ACKNOWLEDGEMENTS

The authors acknowledge support from the EPSRC CDT in Smart Medical Imaging [EP/S022104/1] and a GSK Studentship.

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

APPENDIX

ORGANIZATION

This appendix is organized as follows.

- Appendix A presents additional mathematical details supporting the derivation of DC loss.
- Appendix B is more mathematically involved, and concerns the implications between whether low DC loss implies low signal error and vice versa.
- Appendix C derives tail approximations used in the practical implementation of DC loss, and gives the pseudocodes for implementing DC loss.
- Appendix D presents an illustrative example using DC loss to mitigate noise-fitting in 1D signal deconvolution, with results that support the conclusions of the main paper in an additional application area.
- Appendix E relates experimental details and further experimental results for the DIP section of the main paper, including the effect of a mismatched noise model.
- Appendix F presents analogous content for the PET section of the main paper.
- Appendix G presents an example application of DC loss to a plug-and-play framework.
- Appendix H presents data on the time efficiency of DC loss.
- Appendix I declares the usage of LLMs as required by ICLR.

## A  MATHEMATICAL DERIVATION OF DISTRIBUTIONAL CONSISTENCY LOSS

### A.1  PROBABILITY INTEGRAL TRANSFORM (PIT)

We will justify the statement in Section 3.2 that if the model parameters $\hat{\boldsymbol{\theta}}$ are close to the true parameters $\boldsymbol{\theta}^*$, then the values $s_i := F_i(f(\hat{\boldsymbol{\theta}}), m_i)$ are approximately uniformly distributed on $[0, 1]$. This relies on the classical probability integral transform (Pearson, 1938), which we state and prove below for convenience.

**Lemma A.1** (Probability Integral Transform). *Let $Z$ be a continuous random variable with CDF $F_Z$. Then the transformed variable*

$$U := F_Z(Z) \tag{9}$$

*is uniformly distributed on the interval $[0, 1]$.*

*Proof.* Let $u \in [0, 1]$. We compute the CDF of $U$:

$$\mathbb{P}(U \leq u) = \mathbb{P}(F_Z(Z) \leq u) \tag{10}$$
$$= \mathbb{P}(Z \leq F_Z^{-1}(u)) \tag{11}$$
$$= F_Z(F_Z^{-1}(u)) = u \ . \tag{12}$$

Here, $F_Z^{-1}(u)$ denotes the (generalized) inverse CDF of $Z$, which exists because $F_Z$ is continuous and strictly increasing. The last equality follows from the definition of the inverse CDF.

Thus, the distribution function of $U$ is $\mathbb{P}(U \leq u) = u$, which is the CDF of the uniform distribution on $[0, 1]$. Therefore, $U \sim \text{Uniform}[0, 1]$. □

**Application.**  In our context, for each $i$, let $m_i$ be a measurement assumed to be drawn from the continuous model distribution $\mathcal{D}_i(f_i(\boldsymbol{\theta}^*))$, which has CDF $F_i(\boldsymbol{\theta}^*, \cdot)$. Then the probability integral transform implies:

$$s_i := F_i(\boldsymbol{\theta}^*, m_i) \sim \text{Uniform}[0, 1] \ . \tag{13}$$

If $\hat{\boldsymbol{\theta}} \approx \boldsymbol{\theta}^*$, then $s_i = F_i(\hat{\boldsymbol{\theta}}, m_i)$ is approximately uniformly distributed as well (assuming $f$ and $\mathcal{D}$ are sufficiently well-behaved). This statement is made more precise in Appendix B.

A.2 LOGIT TRANSFORM OF UNIFORM DISTRIBUTION

Let $U \sim \text{Uniform}(0, 1)$ and define

$$Z := \text{logit}(U) = \log\left(\frac{U}{1-U}\right) . \tag{14}$$

We show that $Z \sim \text{Logistic}(0, 1)$ by deriving its CDF directly.

By definition of $Z$, we compute its CDF using the transformation method:

$$F_Z(z) = \mathbb{P}(Z \leq z) = \mathbb{P}\left(\log\left(\frac{U}{1-U}\right) \leq z\right) . \tag{15}$$

Apply the monotonic function $\exp$ to both sides:

$$F_Z(z) = \mathbb{P}\left(\frac{U}{1-U} \leq e^z\right) . \tag{16}$$

Solve the inequality for $U$:

$$\frac{U}{1-U} \leq e^z \quad \Longleftrightarrow \quad U \leq \frac{e^z}{1+e^z} . \tag{17}$$

Thus,

$$F_Z(z) = \mathbb{P}\left(U \leq \frac{e^z}{1+e^z}\right) . \tag{18}$$

Since $U \sim \text{Uniform}(0, 1)$, we have

$$F_Z(z) = \frac{e^z}{1+e^z} = \frac{1}{1+e^{-z}} . \tag{19}$$

The CDF of $Z$ matches that of a standard $\text{Logistic}(0, 1)$ distribution. Therefore,

$$\text{logit}(U) \sim \text{Logistic}(0, 1) . \tag{20}$$

A.3 APPROXIMATION OF THE CDF OF A DISCRETE RANDOM VARIABLE

The probability integral transform in subsection A.1 applies to continuous noise. For discrete noise, $F_Z$ is a step function, so the naive PIT is not exactly uniform. In this work we *use the naive approach*, which is often adequate in practice; we also note a simple randomized variant that would make the transform exactly uniform if desired.

**(A) Naive (non-randomized) PIT can be close to uniform when support is rich.** Let $Z$ be a discrete random variable with CDF $F_Z$ and pmf $p_Z$, and let $m_i \stackrel{\text{i.i.d.}}{\sim} Z$. Define $s_i = F_Z(m_i)$ and the empirical CDF

$$\hat{F}_s(u) = \frac{1}{N}\sum_{i=1}^{N} \mathbf{1}\{s_i \leq u\}. \tag{21}$$

As $N \to \infty$, $\hat{F}_s$ converges to the distribution of $F_Z(Z)$, which places mass at the jump values of $F_Z$. When $Z$ has many support points with relatively balanced probabilities, the mass of $F_Z(Z)$ is spread across many levels in $[0, 1]$, and the empirical histogram of $\{s_i\}$ can look visually close to uniform (e.g. see Figure F1). This heuristic supports the empirical usefulness of DC loss with discrete noise even without randomization.

**(B) Randomized PIT makes the transform exactly uniform (discrete case).** Define an independent $U \sim \text{Uniform}[0, 1]$ and the *randomized PIT*

$$S^{\text{rnd}} := F_Z(Z^-) + U\,p_Z(Z), \qquad \text{where } F_Z(z^-) = \lim_{t\uparrow z} F_Z(t). \tag{22}$$

A direct calculation shows $S^{\mathrm{rnd}} \sim \mathrm{Uniform}[0, 1]$. Indeed, for any $u \in [0, 1]$,

$$\mathbb{P}(S^{\mathrm{rnd}} \leq u) = \sum_z \mathbb{P}(Z = z)\,\mathbb{P}\Big(F_Z(z^-) + U\,p_Z(z) \leq u \,\Big|\, Z = z\Big) \tag{23}$$

$$= \sum_z p_Z(z)\, \frac{\big(u - F_Z(z^-)\big)_+}{p_Z(z)}\, \mathbf{1}\{u \leq F_Z(z)\} \;=\; u, \tag{24}$$

where $(\cdot)_+$ is the positive part and we used that the jumps of $F_Z$ partition $[0, 1]$. Applying the (monotone) logit map gives

$$R^{\mathrm{rnd}} \;:=\; \mathrm{logit}\big(S^{\mathrm{rnd}}\big) \;\sim\; \mathrm{Logistic}(0, 1). \tag{25}$$

Therefore, at the population level the DC loss is exactly zero.

**Practical takeaway.** Our experiments used the naive PIT for discrete noise (for simplicity). If exact uniformity (and hence exact logisticity after the logit map) is desired at the population level, one could instead apply the randomized PIT above with no other changes to the framework.

## B    RELATIONSHIP BETWEEN DC LOSS AND ERROR

**Notation.**    Notation follows the main text. We briefly restate the pieces used below. Let $\boldsymbol{\theta}^*$ denote the ground-truth parameter, $f$ the forward operator, and $\mathbf{y} = f(\boldsymbol{\theta}^*) \in \mathbb{R}^N$ the true signal vector. For any candidate parameter $\hat{\boldsymbol{\theta}}$, write $\hat{\mathbf{y}} = f(\hat{\boldsymbol{\theta}})$. For each $i \in \{1, \ldots, N\}$, the observation $m_i$ (the $i$th component of $\mathbf{m} \in \mathbb{R}^N$) is drawn from a parametric family $\mathcal{D}_i(y_i)$ with continuous CDF $F_i(\cdot \mid y_i)$. Define the probability integral transform (PIT)

$$s_i(\hat{\boldsymbol{\theta}}) = F_i\big(m_i \mid \hat{y}_i\big) \in (0, 1), \tag{26}$$

and set

$$r_i(\hat{\boldsymbol{\theta}}) = \mathrm{logit}\big(s_i(\hat{\boldsymbol{\theta}})\big) \in \mathbb{R}. \tag{27}$$

Let the empirical residual-score measure be

$$\hat{P}_{\{r_i(\hat{\boldsymbol{\theta}})\}} = \frac{1}{N} \sum_{i=1}^N \delta_{r_i(\hat{\boldsymbol{\theta}})}. \tag{28}$$

Independently, draw $Z_1, \ldots, Z_M \overset{\text{i.i.d.}}{\sim} \mathrm{Logistic}(0, 1)$ and define the empirical logistic measure

$$\hat{P}_M^{\log} = \frac{1}{M} \sum_{j=1}^M \delta_{Z_j}. \tag{29}$$

The (empirical two-sample) *DC loss* is

$$\mathcal{L}_{N,M}(\hat{\boldsymbol{\theta}}) = \mathsf{W}_1\big(\hat{P}_{\{r_i(\hat{\boldsymbol{\theta}})\}}, \hat{P}_M^{\log}\big). \tag{30}$$

Signal error[3] is measured by the average squared error:

$$\mathrm{Err}_2(\hat{\boldsymbol{\theta}}) = \frac{1}{N} \sum_{i=1}^N \big(\hat{y}_i - y_i\big)^2. \tag{31}$$

### B.1    ZERO SIGNAL ERROR $\Rightarrow$ NEAR-ZERO EMPIRICAL DC LOSS

**Proposition B.1.** *If* $\mathrm{Err}_2(\hat{\boldsymbol{\theta}}) = 0$, *then* $\hat{y}_i = y_i$ *for all* $i$, *and the following hold:*

(a) Expectation identity. *For each* $i$ *and any bounded measurable test function* $f$,

$$\mathbb{E}\big[f\big(r_i(\hat{\boldsymbol{\theta}})\big)\big] = \mathbb{E}_{Z \sim \mathrm{Logistic}(0,1)}[f(Z)]. \tag{32}$$

(b) Empirical two-sample convergence. *As* $N \to \infty$ *and* $M \to \infty$,

$$\mathcal{L}_{N,M}(\hat{\boldsymbol{\theta}}) = \mathsf{W}_1\big(\hat{P}_{\{r_i(\hat{\boldsymbol{\theta}})\}}, \hat{P}_M^{\log}\big) \xrightarrow{a.s.} 0. \tag{33}$$

*In particular, this holds if* $M = N$ *and both tend to infinity.*

(c) Two-sample rate in expectation. *There exists a universal constant* $C < \infty$ *such that for all* $N, M \geq 1$,

$$\mathbb{E}\big[\mathcal{L}_{N,M}(\hat{\boldsymbol{\theta}})\big] \leq C\left(\frac{1}{\sqrt{N}} + \frac{1}{\sqrt{M}}\right). \tag{34}$$

*Proof.* Since $\mathrm{Err}_2(\hat{\boldsymbol{\theta}}) = 0$, we have $\hat{y}_i = y_i$ for every $i$. By the probability integral transform (with $F_i$ continuous),

$$s_i(\hat{\boldsymbol{\theta}}) = F_i(m_i \mid y_i) \sim \mathrm{Uniform}[0, 1]. \tag{35}$$

Because $\mathrm{logit} : (0, 1) \to \mathbb{R}$ is measurable and strictly increasing, the image of a uniform variable is standard logistic; hence

$$r_i(\hat{\boldsymbol{\theta}}) = \mathrm{logit}\big(s_i(\hat{\boldsymbol{\theta}})\big) \sim \mathrm{Logistic}(0, 1), \tag{36}$$

---

[3]In this section, signal error is taken to mean the error on the forward-modeled estimate.

which yields (a): for any bounded measurable $f$,

$$\mathbb{E}\big[f\big(r_i(\hat{\boldsymbol{\theta}})\big)\big] = \mathbb{E}_{Z \sim \text{Logistic}(0,1)}[f(Z)]. \tag{37}$$

For (b), write $P^{\log}$ for the population logistic distribution and apply the triangle inequality:

$$\mathsf{W}_1\big(\hat{P}_{\{r_i(\hat{\boldsymbol{\theta}})\}}, \hat{P}_M^{\log}\big) \le \mathsf{W}_1\big(\hat{P}_{\{r_i(\hat{\boldsymbol{\theta}})\}}, P^{\log}\big) + \mathsf{W}_1\big(P^{\log}, \hat{P}_M^{\log}\big). \tag{38}$$

Conditionally on $\{\hat{y}_i = y_i\}$, the sample $\{r_i(\hat{\boldsymbol{\theta}})\}_{i=1}^N$ is i.i.d. from $P^{\log}$ with finite first moment, and similarly $\{Z_j\}_{j=1}^M$ is i.i.d. from $P^{\log}$ and independent. By standard results for empirical measures on $\mathbb{R}$,

$$\mathsf{W}_1\big(\hat{P}_{\{r_i(\hat{\boldsymbol{\theta}})\}}, P^{\log}\big) \xrightarrow{\text{a.s.}} 0 \quad \text{as } N \to \infty, \tag{39}$$

$$\mathsf{W}_1\big(P^{\log}, \hat{P}_M^{\log}\big) \xrightarrow{\text{a.s.}} 0 \quad \text{as } M \to \infty, \tag{40}$$

hence $\mathcal{L}_{N,M}(\hat{\boldsymbol{\theta}}) \xrightarrow{\text{a.s.}} 0$ as $N, M \to \infty$.

For (c), take expectations and use known one-dimensional bounds for the empirical Wasserstein-1 distance (via the quantile representation and the Dvoretzky–Kiefer–Wolfowitz inequality):

$$\mathbb{E}\big[\mathsf{W}_1\big(\hat{P}_{\{r_i(\hat{\boldsymbol{\theta}})\}}, P^{\log}\big)\big] \le \frac{C}{\sqrt{N}}, \qquad \mathbb{E}\big[\mathsf{W}_1\big(P^{\log}, \hat{P}_M^{\log}\big)\big] \le \frac{C}{\sqrt{M}}. \tag{41}$$

Combining with the triangle inequality gives

$$\mathbb{E}\big[\mathcal{L}_{N,M}(\hat{\boldsymbol{\theta}})\big] \le C\left(\frac{1}{\sqrt{N}} + \frac{1}{\sqrt{M}}\right). \tag{42}$$

$\square$

## B.2 SMALL SIGNAL ERROR $\Rightarrow$ SMALL DC LOSS

We now argue that when the forward-model error is small, the empirical DC loss must also be small. Intuitively, if $\hat{y}_i$ is close to $y_i$ for each $i$, then the corresponding PIT values $s_i(\hat{\boldsymbol{\theta}})$ are close to the uniform law, hence their logit-transform $r_i(\hat{\boldsymbol{\theta}})$ is close to the logistic law, leading to small Wasserstein distance.

**Proposition B.2.** *Suppose that for each $i$ the CDF $F_i(\cdot \mid \eta)$ is Lipschitz-continuous in $\eta$, with constant $\kappa_i \ge 0$ in the sense that*

$$\big|F_i(m \mid \eta) - F_i(m \mid y_i)\big| \le \kappa_i |\eta - y_i| \qquad \text{for all } m \text{ in the support.} \tag{43}$$

*Then the residual scores $r_i(\hat{\boldsymbol{\theta}})$ satisfy*

$$\big|\text{logit}(F_i(m_i \mid \hat{y}_i)) - \text{logit}(F_i(m_i \mid y_i))\big| \le L_i |\hat{y}_i - y_i|, \tag{44}$$

*where $L_i$ is a finite constant depending on $\kappa_i$ and on the boundedness of the logistic derivative on the relevant interval.*[4]

*Consequently, writing $\Delta_i = \hat{y}_i - y_i$, the empirical residual-score measure obeys*

$$\mathsf{W}_1\big(\hat{P}_{\{r_i(\hat{\boldsymbol{\theta}})\}}, \hat{P}_{\{r_i(\boldsymbol{\theta}^*)\}}\big) \le \frac{1}{N}\sum_{i=1}^N L_i |\Delta_i|. \tag{45}$$

**Corollary B.3.** *If $\text{Err}_2(\hat{\boldsymbol{\theta}}) = \frac{1}{N}\sum_i \Delta_i^2$ is small, then the expected DC loss is also small. In particular,*

$$\mathbb{E}\big[\mathcal{L}_{N,M}(\hat{\boldsymbol{\theta}})\big] \le C\left(\sqrt{\text{Err}_2(\hat{\boldsymbol{\theta}})} + \frac{1}{\sqrt{M}}\right), \tag{46}$$

*where $C$ is a constant depending on the average Lipschitz constants $\frac{1}{N}\sum_i L_i$ and the logistic reference distribution.*

*Proof sketch.* The Lipschitz assumption transfers small perturbations in the signal values $\hat{y}_i - y_i$ to small perturbations in the PIT values, and hence (by smoothness of the logit map on compact intervals) to the residual scores. Averaging and using the definition of the Wasserstein-1 distance yields the stated bound. Taking expectations and combining with the standard empirical-process bound for $\hat{P}_M^{\log}$ relative to its population limit then gives the corollary. $\square$

---

[4]Since logit has derivative $1/(s(1-s))$, it is Lipschitz on any compact sub-interval $[\epsilon, 1-\epsilon] \subset (0,1)$. In practice, tail values contribute negligibly to Wasserstein-1, so this suffices.

### B.3 DOES SMALL DC LOSS IMPLY SMALL ERROR?

The converse direction is more delicate. In general, while small DC loss does imply some bounds on the signal error, it is *not* true that as DC loss tends to zero that signal error must also tend to zero. DC loss is a *global* distributional criterion: it requires the logistic residual values to be statistically indistinguishable from the logistic distribution, but this is only in aggregate and doesn't place a strong constraint on individual data points.

For example, in the Gaussian-noise identity model discussed in Section 4.2, the estimate $\hat{\boldsymbol{\theta}} = \boldsymbol{\theta}^* + 2\mathbf{n}$ produces residuals with the same distributional law as the true $\boldsymbol{\theta}^*$, and hence yields nearly zero DC loss, despite incurring $L_2$ error $\|\hat{\mathbf{y}} - \mathbf{y}\|_2 = \|2\mathbf{n}\|_2$. Thus low DC loss can coexist with high[5] estimation error.

To obtain an implication of the form

$$\mathcal{L}(\hat{\boldsymbol{\theta}}_j) \to 0 \quad \Rightarrow \quad \mathrm{Err}_2(\hat{\boldsymbol{\theta}}_j) \to 0 \quad \text{for a sequence of estimates } (\hat{\boldsymbol{\theta}}_j),$$

one would need to impose additional conditions, such as: (i) identifiability of the forward model, (ii) additional constraints to DC loss introducing uniqueness for the distributional match, or (iii) the use of regularization to select among the manifold of DC-loss minimizers.

In short, while small signal error always forces small DC loss, the reverse is not automatic: DC loss controls distributional consistency, not pointwise closeness, and only under stronger structural assumptions can the two be shown to coincide exactly.

**Remark (role of regularization).** DC loss enforces a *distributional* match: many parameter settings can produce logit residual values that look logistic, so DC loss by itself may not pick a unique solution. Regularization (implicit or explicit) is therefore essential: not as a penalty that trades off data-fidelity (as with MSE/NLL), but as a *selection rule* within the set of DC-consistent solutions (see Figure 2). This lens explains why DC loss can yield either higher or lower estimation error than MSE, depending on the regularizer and optimization path. *In practice*, across the applications we studied, DC loss performed reliably with standard forward models and regularizers. A plausible explanation is that, far from the MLE, DC loss behaves similarly to pointwise losses (see Section 4.1), guiding optimization toward comparable estimates; near the solution, regularization selects among the (near-)zero–DC loss candidates in a way that aligns with prior structure.

### B.4 FROM SIGNAL ERROR TO PARAMETER ERROR

In this appendix we have measured error in the signal space, via

$$\mathrm{Err}_2(\hat{\boldsymbol{\theta}}) = \frac{1}{N} \sum_{i=1}^{N} (\hat{y}_i - y_i)^2, \qquad \hat{\mathbf{y}} = f(\hat{\boldsymbol{\theta}}), \ \mathbf{y} = f(\boldsymbol{\theta}^*).$$

This choice is natural because both the DC loss and pointwise losses are defined directly on the forward-modeled data, and it allows us to unlink the choice of forward model from the error analysis. Nevertheless, one ultimately seeks control of the parameter error $\|\hat{\boldsymbol{\theta}} - \boldsymbol{\theta}^*\|$.

A simple link can be stated under mild conditions on the forward operator $f$. If $f : \Theta \to \mathbb{R}^N$ is injective and locally bi-Lipschitz around $\boldsymbol{\theta}^*$, then there exists a constant $C > 0$ such that

$$\|\hat{\boldsymbol{\theta}} - \boldsymbol{\theta}^*\| \leq C \|\hat{\mathbf{y}} - \mathbf{y}\|_2 = C \sqrt{N \, \mathrm{Err}_2(\hat{\boldsymbol{\theta}})}. \tag{47}$$

Thus small signal error implies small parameter error. In particular, our results showing that small signal error leads to small DC loss extend, under such regularity assumptions, to the conclusion that parameter estimates with low error also have low DC loss.

---

[5]Note DC loss will never reward an estimate that produces residuals that are *extremely* far from the true signal, as it still penalizes residuals that are many $\sigma$ from the noisy signal value. As $\sigma$ tends to zero, the bound that DC loss places on signal error will also tend to zero.

## C  EVALUATING DC LOSS IN PRACTICE: CALCULATING THE LOGIT FUNCTION OF A CDF VALUE

In subsection 4.1, we motivated the direct approximation of $\mathrm{logit}(s)$ to avoid numerical precision difficulties sequentially calculating $s$ and applying the logit function. In this appendix, we derive appropriate tail approximations to $\mathrm{logit}(s)$ for the cases of the Gaussian, Poisson and clipped Gaussian distributions. These approximations are useful in settings where $s$ is close to 0 or 1 and separately evaluating the logit function and CDF is numerically unstable.

### C.1  GAUSSIAN DISTRIBUTION

#### C.1.1  APPROXIMATING LOGIT$(s)$ FOR THE TAILS OF A GAUSSIAN DISTRIBUTION

Consider drawing a measurement $m$ from a Gaussian distribution $\mathcal{N}(\hat{y}, \sigma^2)$. Let

$$s = \Phi\left(\frac{m - \hat{y}}{\sigma}\right) , \tag{48}$$

where $\Phi$ denotes the standard Gaussian CDF, and

$$\mathrm{logit}(s) = \log\left(\frac{s}{1 - s}\right) . \tag{49}$$

We derive asymptotic approximations for $\mathrm{logit}(s)$ in the extreme tails, where $s \approx 1$ or $s \approx 0$. Define the standardized variable:

$$z := \frac{m - \hat{y}}{\sigma} . \tag{50}$$

RIGHT TAIL: $z \gg 0$ (I.E., $s \approx 1$)

Using a classical Laplace approximation for the upper tail of the Gaussian CDF,

$$1 - \Phi(z) \approx \frac{1}{z\sqrt{2\pi}} e^{-z^2/2}, \quad \text{as } z \to \infty , \tag{51}$$

we get:

$$\mathrm{logit}(s) = \log\left(\frac{s}{1 - s}\right) = -\log\left(\frac{1 - s}{s}\right) \approx -\log(1 - s) \approx \frac{z^2}{2} + \log z + \log\sqrt{2\pi} . \tag{52}$$

Thus,

$$\mathrm{logit}\left(\Phi\left(\frac{m - \hat{y}}{\sigma}\right)\right) \approx \frac{1}{2}\left(\frac{m - \hat{y}}{\sigma}\right)^2 + \log\left(\frac{m - \hat{y}}{\sigma}\right) + \log\sqrt{2\pi} , \quad \text{as } m \gg \hat{y} . \tag{53}$$

LEFT TAIL: $z \ll 0$ (I.E., $s \approx 0$)

For large negative $z$, we use the lower tail approximation:

$$\Phi(z) \approx \frac{1}{|z|\sqrt{2\pi}} e^{-z^2/2}, \quad \text{as } z \to -\infty . \tag{54}$$

Then,

$$\mathrm{logit}(s) = \log(s) - \log(1 - s) \approx \log(s) \approx -\frac{z^2}{2} - \log|z| - \log\sqrt{2\pi} . \tag{55}$$

So,

$$\mathrm{logit}\left(\Phi\left(\frac{m - \hat{y}}{\sigma}\right)\right) \approx -\frac{1}{2}\left(\frac{m - \hat{y}}{\sigma}\right)^2 - \log\left|\frac{m - \hat{y}}{\sigma}\right| - \log\sqrt{2\pi} , \quad \text{as } m \ll \hat{y} . \tag{56}$$

---

**Algorithm 2** Distributional Consistency Loss for Gaussian Noise

---

**Require:** Parameter estimate $\hat{\boldsymbol{\theta}}$, observed data $\mathbf{m}$, standard deviation $\sigma$, number of measurements $N$, tail threshold $\tau$
1: $\hat{\mathbf{y}} \leftarrow f(\hat{\boldsymbol{\theta}})$         $\triangleright$ Predicted signal
2: Initialize empty vector $\mathbf{z}$ of length $N$
3: **for** $i = 1$ to $N$ **do**
4:     $z_i \leftarrow \frac{m_i - \hat{y}_i}{\sigma}$         $\triangleright$ Standardized residual
5:     **if** $z_i > \tau$ **then**
6:        $r_i \leftarrow \frac{z_i^2}{2} + \log(z_i) + \log\sqrt{2\pi}$         $\triangleright$ Right tail approximation
7:     **else if** $z_i < -\tau$ **then**
8:        $r_i \leftarrow -\frac{z_i^2}{2} - \log(|z_i|) - \log\sqrt{2\pi}$         $\triangleright$ Left tail approximation
9:     **else**
10:       $s_i \leftarrow \Phi(z_i)$         $\triangleright$ Evaluate Gaussian CDF
11:       $r_i \leftarrow \log\left(\frac{s_i}{1 - s_i}\right)$         $\triangleright$ Standard logit transform
12:     **end if**
13: **end for**
14: Sample $\mathbf{u} \sim \text{Logistic}(0, 1)^N$         $\triangleright$ Generate $N$ samples from reference distribution
15: Sort $\mathbf{r}$ and $\mathbf{u}$ in ascending order
16: $L \leftarrow \frac{1}{N}\sum_{i=1}^{N}|r_i - u_i|$         $\triangleright$ Wasserstein-1 distance
17: **return** $L$

---

SUMMARY

Let $z = \frac{m - \hat{y}}{\sigma}$, then:

$$
\text{logit}\left(\Phi(z)\right) \approx \begin{cases} \dfrac{z^2}{2} + \log z + \log\sqrt{2\pi}\,, & z \gg 0, \\[2mm] -\dfrac{z^2}{2} - \log|z| - \log\sqrt{2\pi}\,, & z \ll 0 \\[2mm] \log\left(\dfrac{\Phi(z)}{1 - \Phi(z)}\right) & \text{otherwise}\,. \end{cases} \tag{57}
$$

### C.1.2 PSEUDOCODE FOR CALCULATING DISTRIBUTIONAL CONSISTENCY LOSS WITH GAUSSIAN DISTRIBUTION

Algorithm 2 gives a full pseudocode for calculating the distributional consistency loss when $\mathcal{D}_i(\hat{y}) = \mathcal{N}(\hat{y}, \sigma^2)$.

### C.2 CLIPPED GAUSSIAN CDF

We extend our logit-CDF approximation to the case of a Gaussian distribution clipped to the interval $[0, 1]$. This arises in applications where measurements are inherently bounded, such as normalized image intensities. To handle this, we define a smoothed approximation to the CDF that matches the Gaussian in the interior but transitions linearly near the boundaries to maintain continuity and gradient stability.

Let $\Phi(m; \hat{y}, \sigma)$ be the Gaussian CDF centered at $\hat{y}$ with standard deviation $\sigma$, and let $\epsilon > 0$ define the width of two linear regions near 0 and 1 respectively. We define a modified CDF $\tilde{F}(m|\hat{y})$ for $m \in [0, 1]$ by:

$$
\tilde{F}(m|\hat{y}) = \begin{cases} \Phi(\epsilon; \hat{y}, \sigma) \cdot \dfrac{m}{\epsilon}, & 0 \le m < \epsilon, \\[2mm] \Phi(m; \hat{y}, \sigma), & \epsilon \le m \le 1 - \epsilon, \\[2mm] \Phi(1 - \epsilon; \hat{y}, \sigma) + (1 - \Phi(1 - \epsilon; \hat{y}, \sigma)) \cdot \dfrac{m - (1 - \epsilon)}{\epsilon}, & 1 - \epsilon < m \le 1\,. \end{cases} \tag{58}
$$

This construction ensures a smooth approximation to a clipped Gaussian CDF while avoiding discontinuities in the derivative.

For measured values that are exactly 0 or 1, we uniformly resample the values to be within $[0, \epsilon]$ or $[1 - \epsilon, 1]$, respectively. This avoids degenerate gradients and provides a consistent treatment of boundary values.

To compute the logit of the CDF, we apply:

$$\text{logit}(\tilde{F}(m|\hat{y})) = \log\left(\frac{\tilde{F}(m|\hat{y})}{1 - \tilde{F}(m|\hat{y})}\right) . \tag{59}$$

In the linear regions near 0 and 1, $\tilde{F}(m|\hat{y})$ behaves approximately linearly with respect to $m$, and no special tail approximation is needed beyond standard numerical clamping (we clamp to a small interval $[\epsilon_{\text{hard}}, 1 - \epsilon_{\text{hard}}]$ to avoid instability). In the central region of $m$ values, the logit is computed directly, with the tail approximations given in Equation 57 where $m$ and $\hat{y}$ are far apart.

This approach allows for stable and differentiable evaluation of $\text{logit}(\tilde{F}(m|\hat{y}))$ across the full range of $m \in [0, 1]$, including at the boundaries.

### C.2.1 PSEUDOCODE FOR CALCULATING DISTRIBUTIONAL CONSISTENCY LOSS WITH CLIPPED GAUSSIAN DISTRIBUTION

Algorithm 3 gives a full pseudocode for calculating the distributional consistency loss for a Gaussian distribution clipped to $[0, 1]$.

## C.3 POISSON CDF

### C.3.1 POISSON-GAMMA DUALITY

We consider how to efficiently evaluate the CDF of a one-dimensional Poisson random variable at a measured value, i.e. compute $s_i$ values as defined in Section 3.2. Suppose $m \in \mathbb{N}$ is drawn from a Poisson distribution with unknown mean, and $\hat{y}$ is a predicted mean value. We consider the following Poisson-Gamma duality lemma that shows we may equivalently use an appropriately defined Gamma CDF instead of a Poisson CDF:

**Lemma C.1.** *For all $\hat{y} > 0$ and $m \in \mathbb{N}$,*

$$s(\hat{y}, m) = 1 - t(\hat{y}, m), \tag{60}$$

*where*

$$s(\hat{y}, m) = \sum_{k=0}^{m} \frac{e^{-\hat{y}}\hat{y}^k}{k!} , \quad t(\hat{y}, m) = \int_0^{\hat{y}} \frac{e^{-u}u^m}{m!} \, du . \tag{61}$$

*Proof.* Let $X \sim \text{Poisson}(\hat{y})$ and $Y \sim \text{Gamma}(m + 1, 1)$. Then:

$$s(\hat{y}, m) = \mathbb{P}(X \leq m) , \tag{62}$$
$$t(\hat{y}, m) = \mathbb{P}(Y \leq \hat{y}) . \tag{63}$$

To justify the identity $\mathbb{P}(X \leq m) = \mathbb{P}(Y > \hat{y})$, we appeal to the structure of a homogeneous Poisson process $\{N(t)\}_{t \geq 0}$ with unit rate. Then:

- The time $Y$ of the $(m+1)^{\text{th}}$ event follows a Gamma distribution: $Y \sim \text{Gamma}(m+1, 1)$.

- The count $X = N(\hat{y})$, the number of events up to time $\hat{y}$, is Poisson distributed with mean $\hat{y}$.

By construction, the events of the Poisson process satisfy:

$$\{N(\hat{y}) \leq m\} \iff \{Y > \hat{y}\} , \tag{64}$$

since if the $(m+1)^{\text{st}}$ event has not occurred by time $\hat{y}$, there can be at most $m$ events up to that time. Therefore,

$$\mathbb{P}(X \leq m) = \mathbb{P}(Y > \hat{y}) . \tag{65}$$

---

**Algorithm 3** Distributional Consistency Loss for Clipped Gaussian Noise (with Tail Approximation)

---

**Require:** Parameter estimate $\hat{\boldsymbol{\theta}}$, observed data $\mathbf{m}$, standard deviation $\sigma$, number of measurements $N$, ramp width $\epsilon$, tail threshold $\delta$

1: $\hat{\mathbf{y}} \leftarrow f(\hat{\boldsymbol{\theta}})$               ▷ Predicted signal
2: Initialize empty vector $\mathbf{r}$ of length $N$
3: **for** $i = 1$ to $N$ **do**
4:    $q \leftarrow \hat{y}_i$
5:    $m \leftarrow m_i$
6:    **if** $m = 0$ **then**
7:      $m \leftarrow \text{Uniform}(0, \epsilon)$       ▷ Perturb endpoints to avoid discrete effects
8:    **else if** $m = 1$ **then**
9:      $m \leftarrow \text{Uniform}(1 - \epsilon, 1)$
10:    **end if**
11:    $z \leftarrow \frac{m-q}{\sigma}$             ▷ Standardized residual
12:    $c_\epsilon \leftarrow \Phi\left(\frac{\epsilon-q}{\sigma}\right)$
13:    $c_{1-\epsilon} \leftarrow \Phi\left(\frac{1-\epsilon-q}{\sigma}\right)$
14:    **if** $m < \epsilon$ **then**
15:      $s \leftarrow \left(\frac{m}{\epsilon}\right) \cdot c_\epsilon$         ▷ Linear ramp left
16:    **else if** $m > 1 - \epsilon$ **then**
17:      $s \leftarrow c_{1-\epsilon} + \left(\frac{m-(1-\epsilon)}{\epsilon}\right) \cdot (1 - c_{1-\epsilon})$    ▷ Linear ramp right
18:    **else**
19:      $s \leftarrow \Phi(z)$      ▷ Standard Gaussian CDF in central region
20:    **end if**
21:    **if** $s < \delta$ **then**     ▷ Tail approximations to Logit of Gaussian CDF
22:      $r_i \leftarrow -\left(\frac{z^2}{2} + \log(|z|) + \log\sqrt{2\pi}\right)$
23:    **else if** $s > 1 - \delta$ **then**
24:      $r_i \leftarrow \frac{z^2}{2} + \log(|z|) + \log\sqrt{2\pi}$
25:    **else**
26:      $r_i \leftarrow \log\left(\frac{s}{1-s}\right)$
27:    **end if**
28: **end for**
29: Sample $\mathbf{u} \sim \text{Logistic}(0, 1)^N$        ▷ Reference distribution
30: Sort $\mathbf{r}$ and $\mathbf{u}$ in ascending order
31: $L \leftarrow \frac{1}{N}\sum_{i=1}^{N}|r_i - u_i|$        ▷ Wasserstein-1 distance
32: **return** $L$

---

Substituting equation 62, equation 63, and equation 65 gives:

$$s(\hat{y}, m) = 1 - t(\hat{y}, m) \,, \tag{66}$$

as claimed.                    $\square$

As a consequence of the Poisson-Gamma duality lemma, we may calculate $s$ in Section 3.2 by evaluating the CDF of a $\text{Gamma}(\mathbf{m} + 1, 1)$ random variable. This is beneficial because PyTorch does not support evaluating the CDF of a Poisson random variable, but does support evaluating the CDF of a Gamma random variable. (While specifying a Poisson CDF manually is possible, it is simpler to make use of optimized PyTorch functions where possible).

### C.3.2 APPROXIMATION OF LOGIT($s$) FOR POISSON CDF TAILS VIA GAMMA DUALITY AND TAIL EXPANSIONS

Building on the Poisson–Gamma duality $s(\hat{y}, m) = 1 - t(\hat{y}, m)$, where

$$t(\hat{y}, m) = \int_0^{\hat{y}} \frac{e^{-u} u^m}{m!} \, du = \mathbb{P}(Y \leq \hat{y}), \quad Y \sim \text{Gamma}(m + 1, 1) \,, \tag{67}$$

---

**Algorithm 4** Distributional Consistency Loss for Poisson Noise

---

**Require:** Parameter estimate $\hat{\boldsymbol{\theta}}$, observed data $\mathbf{m}$, number of measurements $N$, tail threshold $\epsilon$
1: $\hat{\mathbf{y}} \leftarrow f(\hat{\boldsymbol{\theta}})$         ▷ Predicted signal
2: Initialize empty vector $\mathbf{r}$ of length $N$
3: **for** $i = 1$ to $N$ **do**
4:      $q \leftarrow \max(\hat{y}_i, 0)$         ▷ Ensure non-negativity
5:      $s \leftarrow 1 - \text{GammaCDF}(q; \alpha = m + 1, \beta = 1)$         ▷ Poisson posterior tail probability
6:      **if** $s \leq \epsilon$ **then**
7:          $r_i \leftarrow m \cdot \log(q) - q - \log \Gamma(m + 1)$         ▷ Lower tail approximation
8:      **else if** $s \geq 1 - \epsilon$ **then**
9:          $r_i \leftarrow -m \cdot \log(q) + q + \log \Gamma(m + 2)$         ▷ Upper tail approximation
10:      **else**
11:          $r_i \leftarrow \log\left(\frac{s}{1-s}\right)$         ▷ Standard logit transform
12:      **end if**
13: **end for**
14: Sample $\mathbf{u} \sim \text{Logistic}(0, 1)^N$         ▷ Reference distribution
15: Sort $\mathbf{r}$ and $\mathbf{u}$ in ascending order
16: $L \leftarrow \frac{1}{N} \sum_{i=1}^{N} |r_i - u_i|$         ▷ Wasserstein-1 distance
17: **return** $L$

---

we now derive asymptotic approximations for $\text{logit}(s)$ in the extreme left and right tails, i.e., when $\hat{y} \gg m$ or $\hat{y} \ll m$, respectively.

LEFT TAIL: ($\hat{y} \gg m$, SO $s \approx 0$)

In this regime, the CDF $s$ is small, and we approximate it by the leading term of the Poisson probability mass function:

$$s \approx \frac{\hat{y}^m e^{-\hat{y}}}{m!}. \tag{68}$$

Apply logit, we obtain:

$$\text{logit}(s) \approx \log s = m \log \hat{y} - \hat{y} - \log m! . \tag{69}$$

RIGHT TAIL: ($\hat{y} \ll m$, SO $s \approx 1$)

In this case, the upper tail $1 - s = t(\hat{y}, m)$ is small. We approximate the incomplete Gamma integral by truncating its series expansion after the first term beyond the threshold $m$:

$$t(\hat{y}, m) \approx e^{-\hat{y}} \cdot \frac{\hat{y}^{m+1}}{(m + 1)!} . \tag{70}$$

Then,

$$\text{logit}(s) \approx -\log(1 - s) \approx -\log\left(e^{-\hat{y}} \cdot \frac{\hat{y}^{m+1}}{(m + 1)!}\right) = -(m + 1) \log \hat{y} + \hat{y} + \log(m + 1)! . \tag{71}$$

SUMMARY:

The full approximation for $\text{logit}(s)$, where $s = \mathbb{P}(X \leq m)$ and $X \sim \text{Poisson}(\hat{y})$, is given by

$$\text{logit}(s(\hat{y}, m)) \approx \begin{cases} m \log \hat{y} - \hat{y} - \log m! & \text{if } \hat{y} \gg m , \\ -(m + 1) \log \hat{y} + \hat{y} + \log(m + 1)! & \text{if } \hat{y} \ll m , \\ \log\left(\frac{s(\hat{y}, m)}{1 - s(\hat{y}, m)}\right) & \text{otherwise} . \end{cases} \tag{72}$$

### C.3.3 PSEUDOCODE FOR CALCULATING DISTRIBUTIONAL CONSISTENCY LOSS WITH POISSON DISTRIBUTION

Algorithm 4 gives a full pseudocode for calculating the distributional consistency loss when $\mathcal{D}_i(\hat{y}) = \text{Poisson}(\hat{y})$.

# D  1D DECONVOLUTION APPLICATION

We include the following example application to illustrate the use of DC loss in a simple non-imaging context.

## D.1  ILLUSTRATIVE EXAMPLE: DECONVOLUTION UNDER GAUSSIAN NOISE

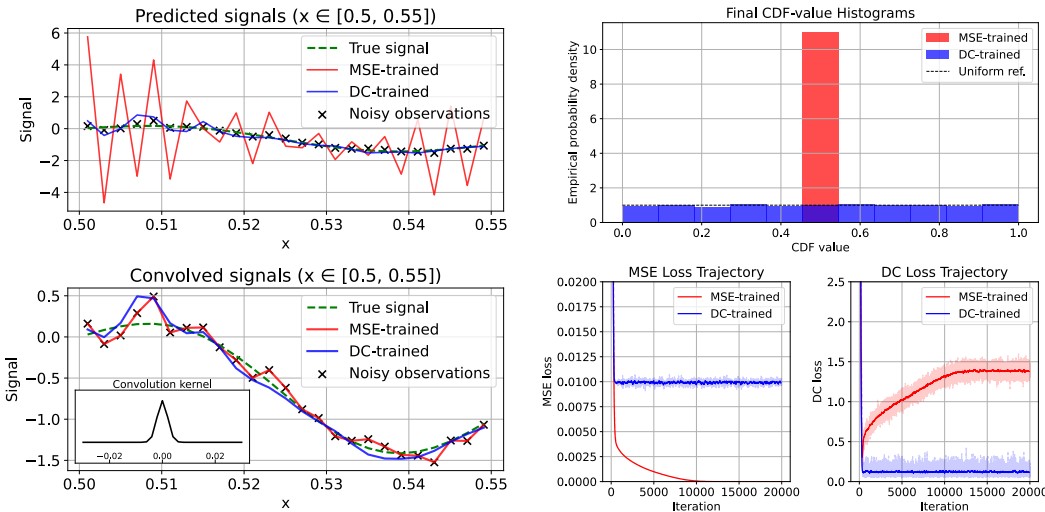

(a) Top: predicted deconvolved signals estimated with MSE and DC loss. Bottom: predicted deconvolved signals after reapplying convolution with the known operator $H$.

(b) Top: histograms of CDF values under each model's implied noise distributions. Bottom: training loss curves evaluating each of MSE and DC loss on MSE-trained and DC-trained predictions.

Figure D1: 1D deconvolution with known Gaussian blur and noise level. MSE inversion amplifies noise artifacts, while DC loss yields a cleaner, artifact-free reconstruction.

To illustrate DC loss in a classic inverse problem, we estimate a 1D signal $\boldsymbol{\theta}^*$ from blurred and noisy measurements

$$\mathbf{m} = H\boldsymbol{\theta}^* + \boldsymbol{\eta}, \tag{73}$$

where $H$ is a known Gaussian-blur convolution and $\boldsymbol{\eta} \sim \mathcal{N}(0, \sigma_{\mathrm{noise}}^2 \mathbf{I})$. A single ground-truth signal, composed of low- and high-frequency sinusoids, is forward modeled and sampled at $N = 500$ discrete timesteps to produce noisy observations. We parameterize $\hat{\boldsymbol{\theta}}$ directly as a free vector of length $N$ and optimize it under two objectives: the standard MSE $\|H\hat{\boldsymbol{\theta}} - \mathbf{m}\|^2$ and our DC loss. To demonstrate the properties of both losses, both reconstructions use Adam (Kingma and Ba, 2015) for 20,000 iterations with learning rate $5 \times 10^{-3}$.

Figure D1a shows that the MSE-trained solution (red) overfits by amplifying small, high-frequency noise producing pronounced oscillations. In contrast, the DC-trained solution (blue) remains smooth and closely tracks the true signal (green dashed).

In Figure D1b, the top row shows histograms of CDF values under each model's implied Gaussian noise distribution: the MSE-trained fit collapses its histogram at around 0.5 (indicative of overfitting), whereas the DC-trained fit remains approximately uniform (distributionally consistent). The bottom row shows the loss trajectories. In the MSE-loss plot, the MSE objective for the MSE-trained reconstruction converges towards zero, while the DC-trained reconstruction's MSE plateaus above zero. In the DC-loss plot, the DC objective converges to

$$\mathbb{E}_{u\sim\mathrm{Logistic}(0,1)}\big[|u|\big] = \ln 4 \approx 1.386 \tag{74}$$

for the MSE-trained reconstruction (Balakrishnan, 1985) (the expected distance when all CDFs collapse to 0.5) whereas the DC-trained reconstruction's DC loss converges near zero, avoiding noise amplification.

The rest of Appendix D provides implementation details and additional experiments varying the number of measurements and the noise level.

## D.2 EXPERIMENTAL DESIGN

To investigate DC loss for the task of deconvolution, a true signal was generated as $\theta(x) = \sin(10\pi x) + \frac{1}{2}\sin(40\pi x)$, and uniformly sampled at $N = 500$ points between 0 and 1.

To simulate a degraded observation, the signal was blurred using a 1D Gaussian kernel defined in domain units. Specifically, a Gaussian kernel in index-space with $\sigma_{\text{index-space blur}} = 1.0$, corresponding to a signal-space blur of $\sigma_{\text{blur}} = 0.002$ was applied with a width of 31 points (i.e. 15 neighbors of the central point on each side). The resulting kernel was applied by convolution with reflective boundary conditions.

Finally, additive Gaussian noise with standard deviation $\sigma_{\text{noise}} = 0.1$ was added independently to each measurement:

$$m_i = (H\boldsymbol{\theta}^*)_i + \eta_i, \qquad \eta_i \sim \mathcal{N}(0,\, \sigma_{\text{noise}}^2). \tag{75}$$

The deconvolution task was then to recover the original signal (pre-convolution) from the noisy measurements.

For each experiment, Adam (Kingma and Ba, 2015) with a learning rate of 0.005 was run for 20,000 iterations. A vector of zeros was used as the starting set of parameters. Experiments were conducted with a 24GB Nvidia GeForce RTX 3090 GPU; running deconvolution with MSE and then DC loss took $\sim 4$ minutes (with many experiments able to be run in parallel).

Except where otherwise defined, we took $\sigma_{\text{noise}} = 0.1$, $\sigma_{\text{blur}} = 0.002$ and $N = 500$.

## D.3 ASSESSING DISTRIBUTIONAL CONSISTENCY OF THE TRUE SIGNAL

We investigated the assumption that the true clean image should theoretically induce a uniform histogram of CDF values (asymptotically). In Figure D2, we compared evaluating the DC loss on the true convolved signal and on the noisy convolved signal for a varied number of measured datapoints $N$.

For high $N$, we see that evaluating the DC loss on the convolved true signal yields an approximately uniform distribution of CDF values (and correspondingly low DC loss value), while evaluating the DC loss on the noisy image yields a histogram with a peak at 0.5 (and correspondingly high DC loss value of approximately $\ln 4$ - see Equation 74). For lower values of $N$, we obtain higher average DC loss values and less uniform histograms for the true convolved signal, although still more uniform than for the noisy signal.

## D.4 VARYING NOISE LEVEL

In Figures D3, D4 and D5, we investigated varying the noise applied to the convolved signal (with the values $\sigma_{\text{noise}} = 0.01$, 0.05 and 0.2 respectively). We observed that at the lowest noise level, both loss functions resulted in a near-perfect fit to the true signal, but only the DC-trained prediction avoided noise amplification effects in the estimate of the pre-convolved signal. This trend was seen more dramatically as the noise level was increased. The fit to the true convolved signal worsened for both loss functions, but only the MSE-trained vector experienced extreme noise amplification in the pre-convolved signal (as a result of overfitting).

The loss trajectories and CDF histograms observed in these cases were consistent with those seen in Section D.1.

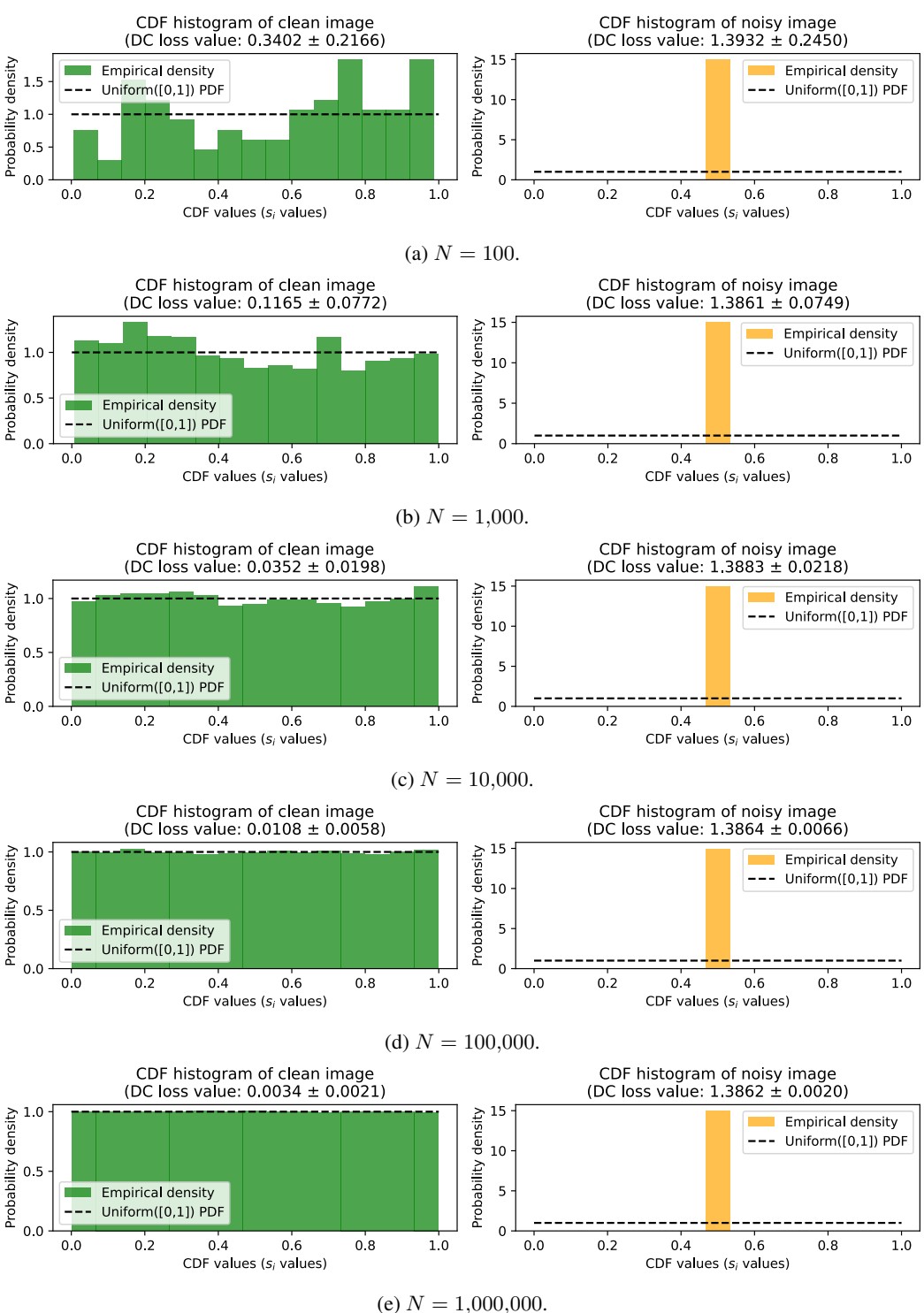

Figure D2: Investigating the DC loss values obtained and the uniformity of CDF histograms in the ideal and worst cases of evaluating the DC loss against the true convolved signal or noisy convolved signal respectively. Subplots show the effect of increasing the number of measurements from $N = 100$ to $N = 1,000,000$. Uncertainty values are given as $1.96\times$ the standard deviation over 100 evaluations of the (stochastic) loss on different instantiations of noise.

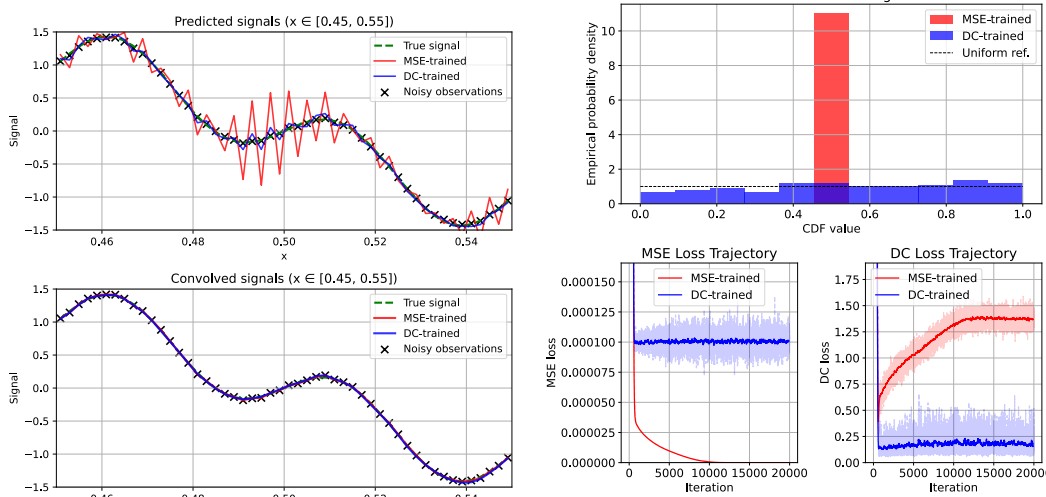

(a) Top: predicted signals estimated with MSE and DC loss. Bottom: signals after convolution with the known operator $H$.

(b) Top: histograms of CDF values under each model's implied noise distributions. Bottom: training loss curves evaluating each of MSE and DC loss on MSE-trained and DC-trained predictions.

Figure D3: Results for 1D deconvolution with known Gaussian blur with $\sigma_{\text{blur}} = 0.02$ and noise with $\sigma_{\text{noise}} = 0.01$.

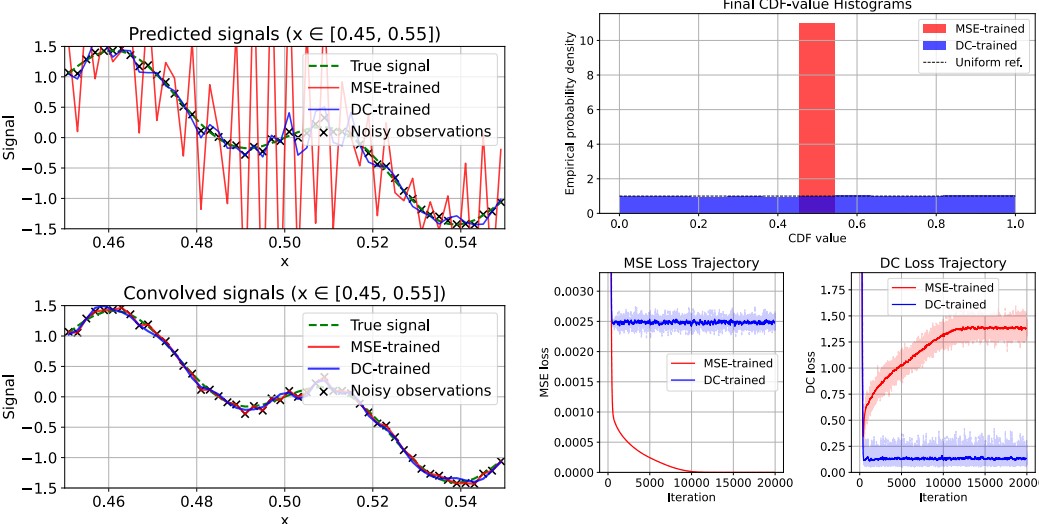

(a) Top: predicted signals estimated with MSE and DC loss. Bottom: signals after convolution with the known operator $H$.

(b) Top: histograms of CDF values under each model's implied noise distributions. Bottom: training loss curves evaluating each of MSE and DC loss on MSE-trained and DC-trained predictions.

Figure D4: Results for 1D deconvolution with known Gaussian blur with $\sigma_{\text{blur}} = 0.02$ and noise with $\sigma_{\text{noise}} = 0.05$.

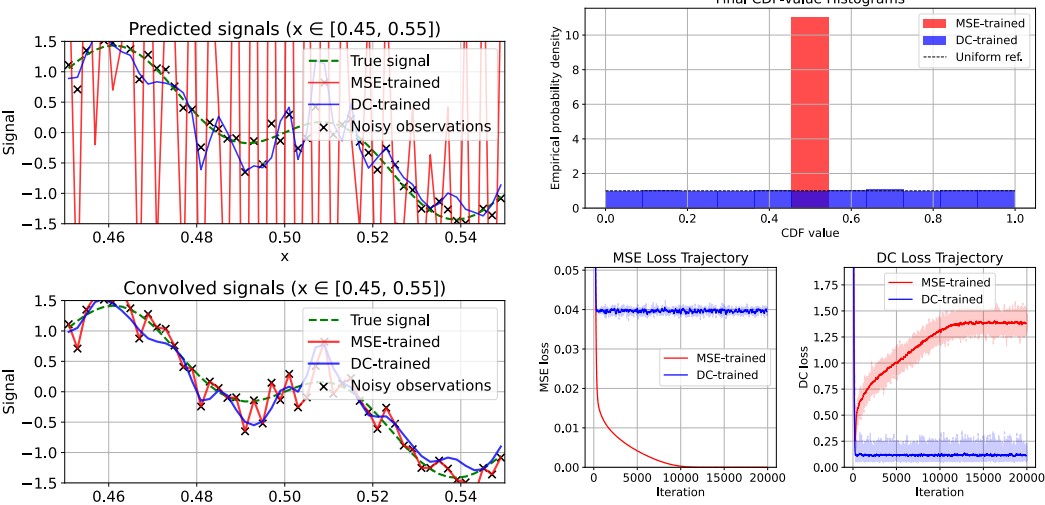

(a) Top: predicted signals estimated with MSE and DC loss. Bottom: signals after convolution with the known operator $H$.

(b) Top: histograms of CDF values under each model's implied noise distributions. Bottom: training loss curves evaluating each of MSE and DC loss on MSE-trained and DC-trained predictions.

Figure D5: Results for 1D deconvolution with known Gaussian blur with $\sigma_{\text{blur}} = 0.02$ and noise with $\sigma_{\text{noise}} = 0.2$.

# E   DIP APPLICATION

## E.1   EXPERIMENTAL DESIGN

We followed the experimental design of Ulyanov et al. (2020) to specify a denoising task for this work. We used the same hourglass architecture with skip connections described in their paper, using their code made available at `https://github.com/DmitryUlyanov/deep-image-prior`. In all experiments, we used Adam (Kingma and Ba, 2015) with learning rate $1 \times 10^{-3}$. Experiments were conducted with a 24GB Nvidia GeForce RTX 3090 GPU and run for 10,000 iterations; for the largest image size $512 \times 512$, running DIP-MSE and then DIP-DC took $\sim 40$ minutes (with $\sim 4$ experiments able to be run in parallel).

Images were first normalized to [0,1]. Then, Gaussian noise was added independently pixel- and channel-wise to 3-channel RGB images with a fixed standard deviation $\sigma$. Pixel intensities were then clipped to [0,1], as in Ulyanov et al. (2020). The MSE and DC loss were then evaluated after flattening channel and spatial dimensions to a 1D vector.

The clipping function induces a non-continuous noise distribution $\mathcal{D}_i$ for each measurement (i.e. one channel of one pixel), with probability mass at 0 and 1. However, the CDF of a clipped Gaussian distribution may still be well approximated, and we use such an approximation in our experiments in this section (details given in Appendix C.2).

We investigated varied noise levels from $\sigma = \frac{10}{255}$ to $\sigma = \frac{100}{255}$. We fixed the random image and network input for experiments comparing DIP-MSE to DIP-DC; for repeatability, we ran experiments with 5 random seeds. Where error bars are shown in Figure 5d, E2 and E3, these are with respect to 5 random initializations of random noise and network parameters. The results shown in the main body of the paper are for random seed 0 and $\sigma = \frac{75}{255}$, for the image of an F-16 aircraft.

The F-16 aircraft image is from a standard dataset[6] intended for research purposes. The train, plane and castle images are from Martin et al. (2001) (and our usage complies with the custom license, which states use is permitted "for non-commercial research and educational purposes"). The cat image is reproduced with permission from the photographer.

## E.2   ASSESSING THE DISTRIBUTIONAL CONSISTENCY OF THE CLEAN IMAGE

We empirically validate the assumption that the true clean image should theoretically induce a uniform histogram of CDF values (asymptotically). In Figure E1, we see that evaluating the DC loss on the clean image yields an approximately uniform distribution of CDF values (and correspondingly low DC loss value), while evaluating the DC loss on the noisy image yields a histogram with a peak at 0.5 (and correspondingly high DC loss value of approximately $\ln 4$ - see Equation 74).

## E.3   EFFECT OF VARIED NOISE LEVEL

We varied $\sigma$ from $\sigma = \frac{10}{255}$ to $\sigma = \frac{100}{255}$ and performed DIP with each loss at each $\sigma$.

Figure E2 shows the peak SSIM achieved by each method on the F-16 image. Figure E3 shows the peak PSNR, the PSNR of the final image and the PSNR of the mean of the images at the $9100^{\text{th}}, 9200^{\text{th}}, \cdots, 10000^{\text{th}}$ iterations.

In Figures E4, E5, E6, E7 and E8, we give more detailed results (for seed 0) at different noise levels $\sigma$.

## E.4   RESULTS WITH VARIED IMAGES

We additionally show some results with varied $\sigma$ for different images, to validate that our findings generalize beyond the F-16 aircraft image used in the main text.

Figure E9 shows the output of DIP-DC and DIP-MSE on a selection of natural images at varied $\sigma$. These results are explored in more depth in Figures E10, E11, E12 and E13, showing loss curves that correspond with what was observed in Section 5.1.

---

[6]`https://sipi.usc.edu/database/database.php?volume=misc&image=11`

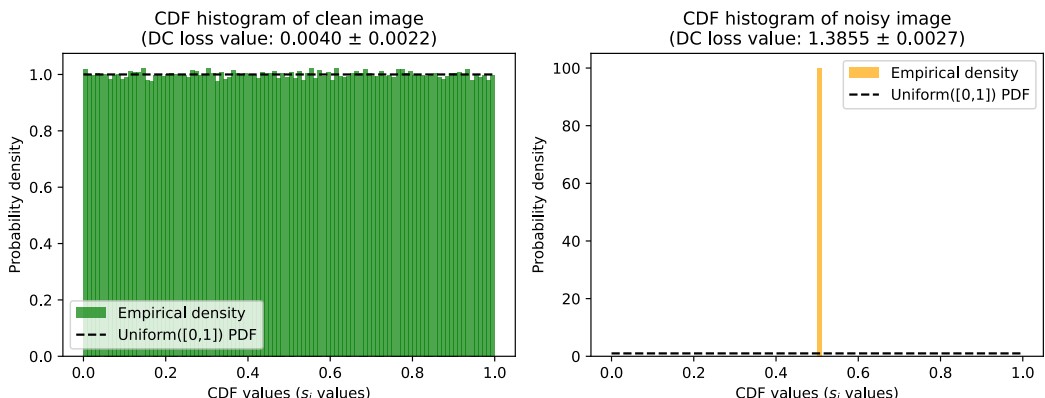

Figure E1: Histograms of CDF values obtained during the evaluation of the DC loss on the true image (left) and the noisy image (right) (with $\sigma = \frac{10}{255}$). Uncertainty values are given as $1.96\times$ the standard deviation over 100 evaluations of the (stochastic) loss on different instantiations of noise.

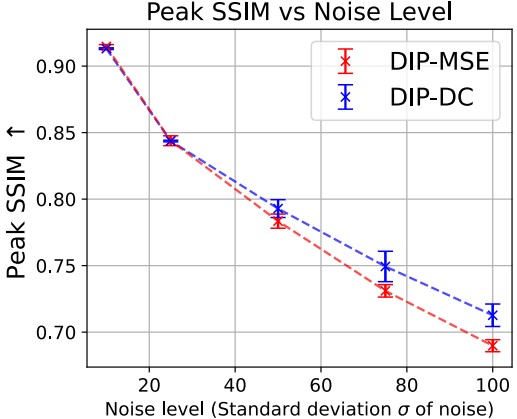

Figure E2: Peak SSIM metric values for DIP-MSE and DIP-DC. Uncertainty is shown with respect to 5 initializations of random noise and network parameters.

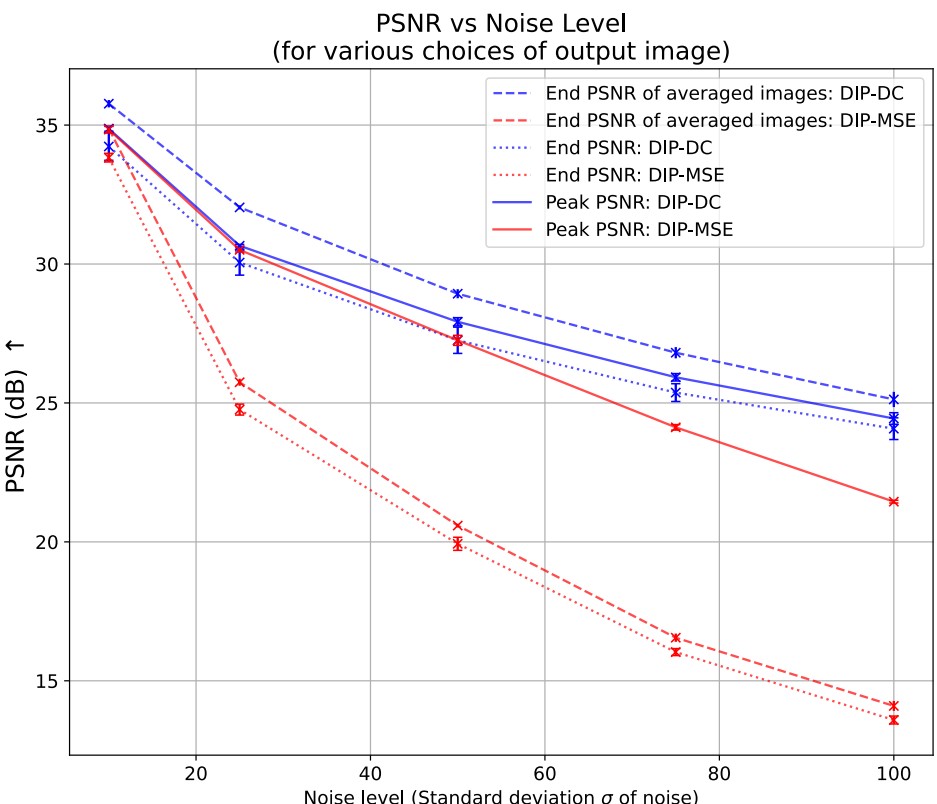

Figure E3: PSNR metric values of variant outputs for DIP-MSE and DIP-DC. Specifically, the peak PSNR achieved, the PSNR of the image at the $10000^{\text{th}}$ iteration and the PSNR of the mean of the images at the $9100^{\text{th}}, 9200^{\text{th}}, \cdots, 10000^{\text{th}}$ iterations are shown. Uncertainty is with respect to 5 initializations of random noise and network parameters.

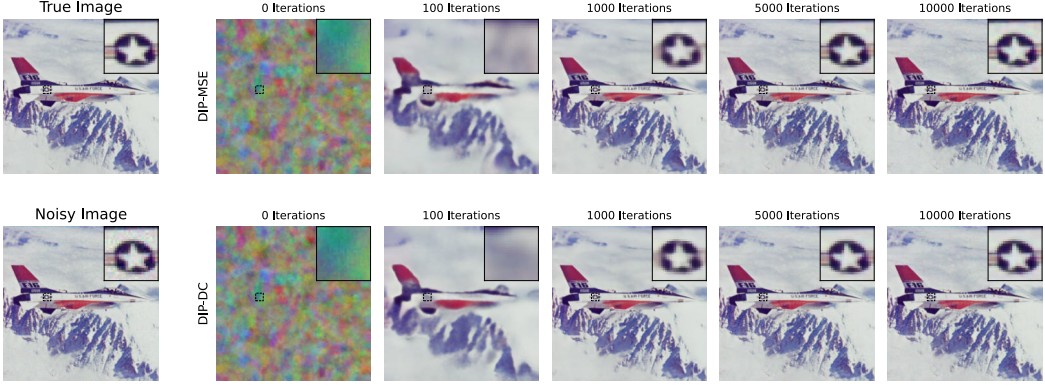

(a) Images produced by DIP-MSE and DIP-DC as a function of iteration number.

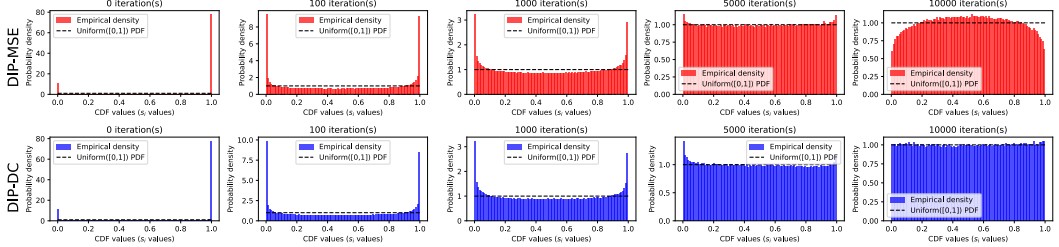

(b) Histograms associated with images produced by DIP-MSE and DIP-DC as a function of iteration number.

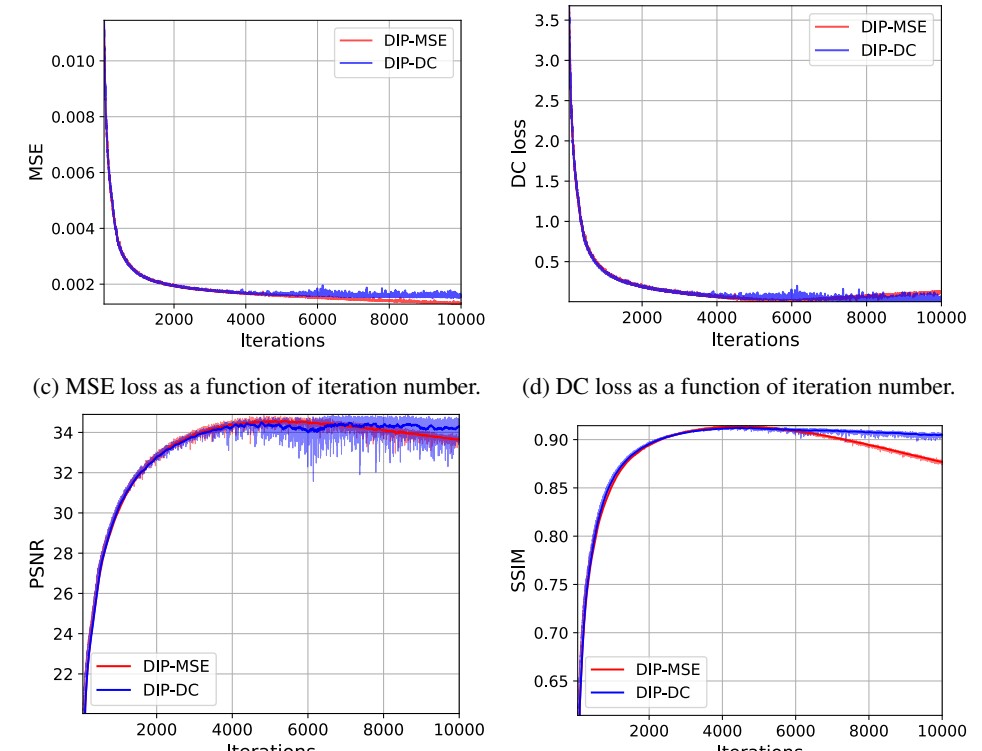

(c) MSE loss as a function of iteration number.

(d) DC loss as a function of iteration number.

(e) PSNR as a function of iteration number.

(f) SSIM as a function of iteration number.

Figure E4: Results for DIP-MSE and DIP-DC at $\sigma = \frac{10}{255}$.

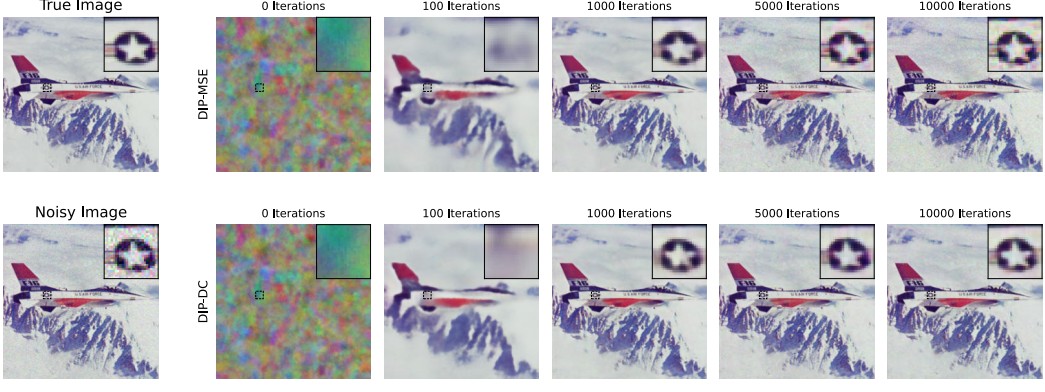

(a) Images produced by DIP-MSE and DIP-DC as a function of iteration number.

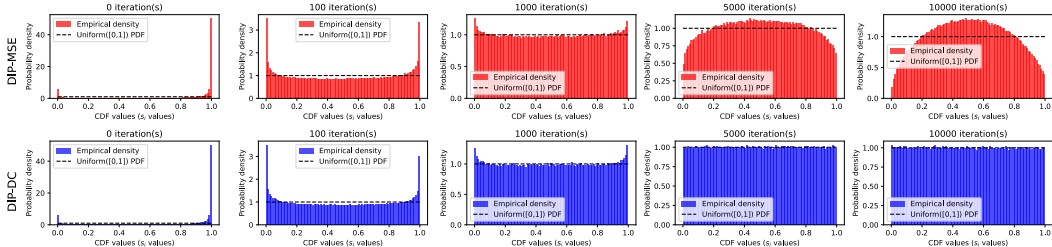

(b) Histograms associated with images produced by DIP-MSE and DIP-DC as a function of iteration number.

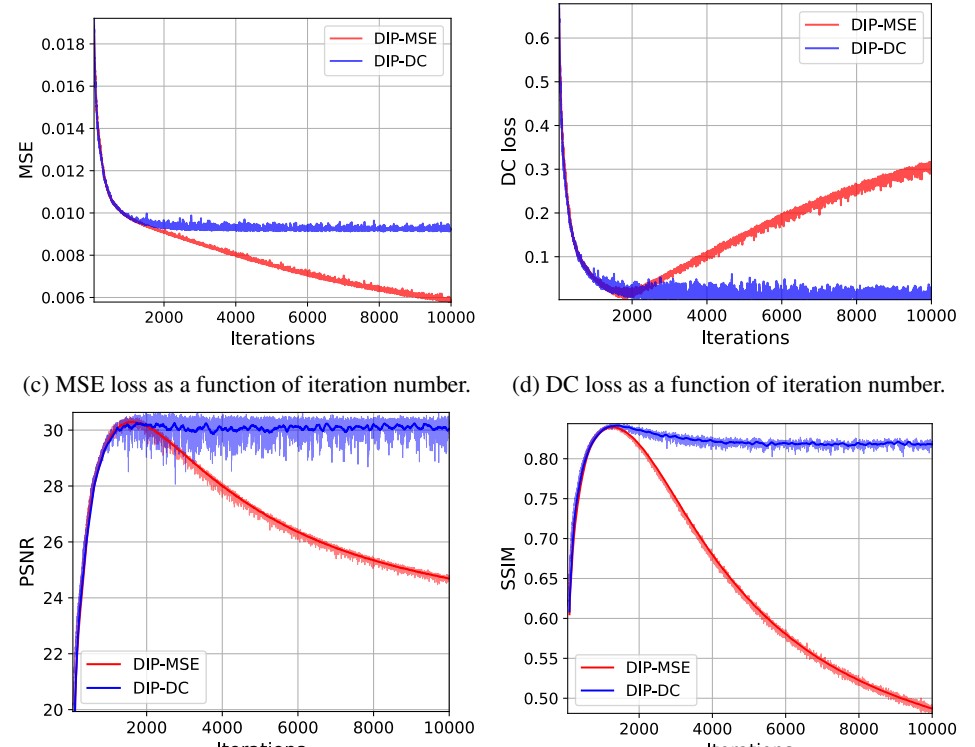

(c) MSE loss as a function of iteration number.

(d) DC loss as a function of iteration number.

(e) PSNR as a function of iteration number.

(f) SSIM as a function of iteration number.

Figure E5: Results for DIP-MSE and DIP-DC at $\sigma = \frac{25}{255}$.

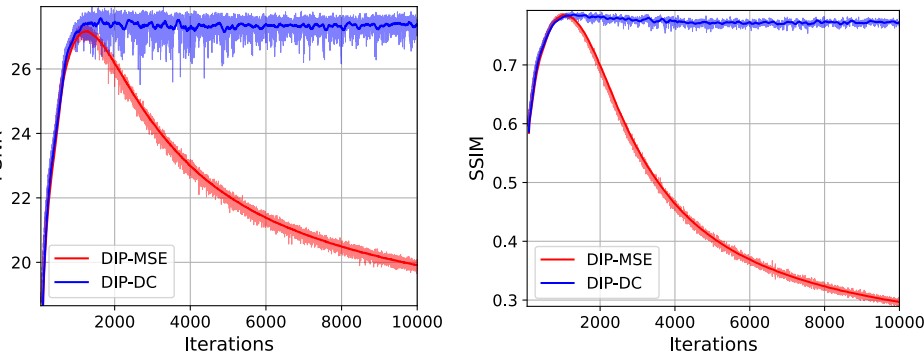

(a) Images produced by DIP-MSE and DIP-DC as a function of iteration number.

(b) Histograms associated with images produced by DIP-MSE and DIP-DC as a function of iteration number.

(c) MSE loss as a function of iteration number.

(d) DC loss as a function of iteration number.

(e) PSNR as a function of iteration number.

(f) SSIM as a function of iteration number.

Figure E6: Results for DIP-MSE and DIP-DC at $\sigma = \frac{50}{255}$.

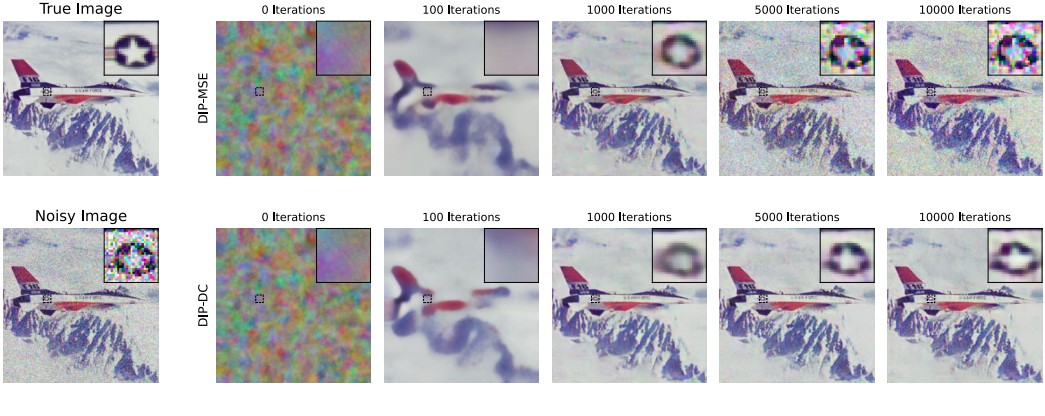

(a) Images produced by DIP-MSE and DIP-DC as a function of iteration number.

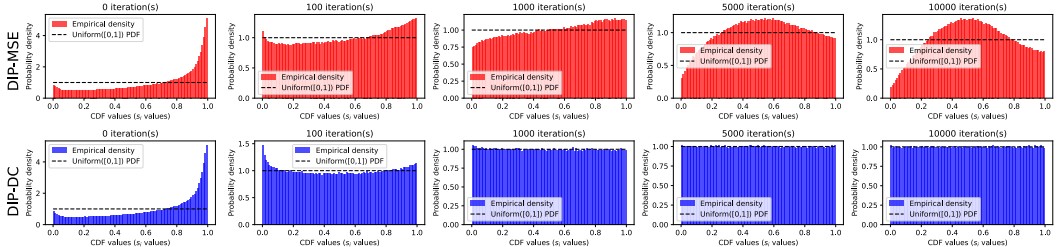

(b) Histograms associated with images produced by DIP-MSE and DIP-DC as a function of iteration number.

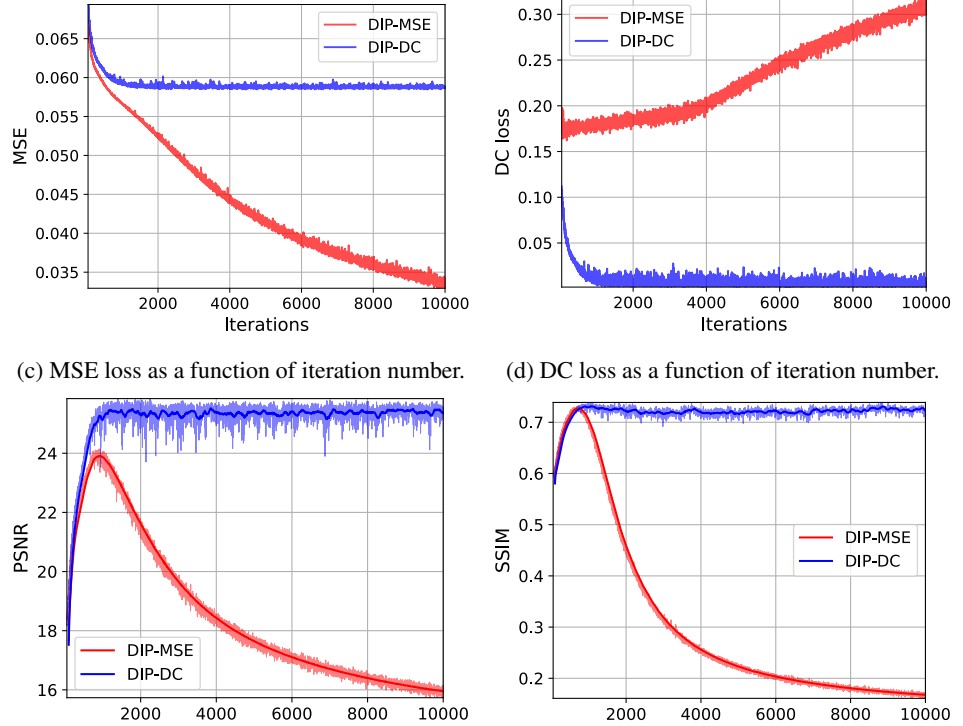

(c) MSE loss as a function of iteration number. (d) DC loss as a function of iteration number.

(e) PSNR as a function of iteration number. (f) SSIM as a function of iteration number.

Figure E7: Results for DIP-MSE and DIP-DC at $\sigma = \frac{75}{255}$.

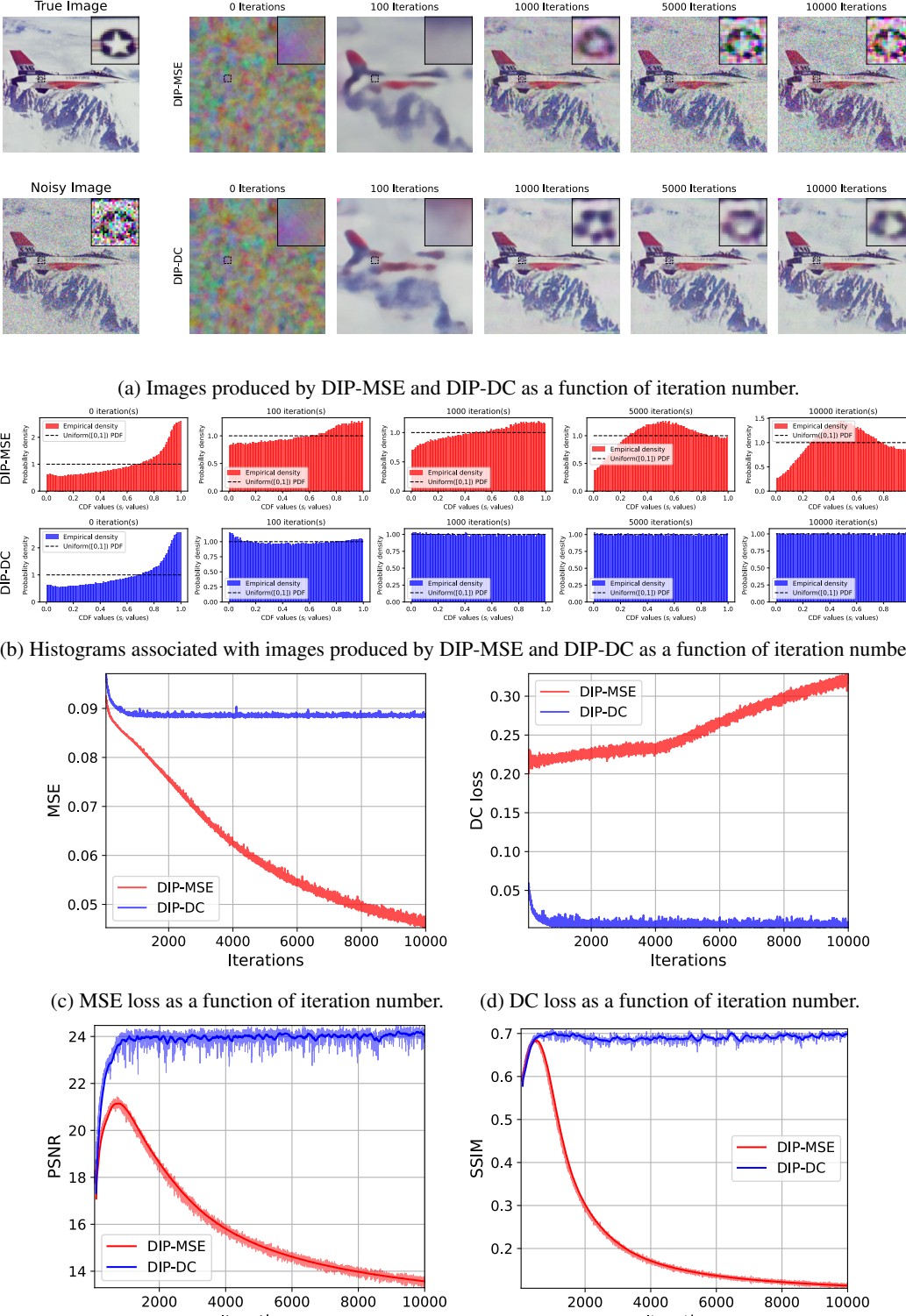

(a) Images produced by DIP-MSE and DIP-DC as a function of iteration number.

(b) Histograms associated with images produced by DIP-MSE and DIP-DC as a function of iteration number.

(c) MSE loss as a function of iteration number.

(d) DC loss as a function of iteration number.

(e) PSNR as a function of iteration number.

(f) SSIM as a function of iteration number.

Figure E8: Results for DIP-MSE and DIP-DC at $\sigma = \frac{100}{255}$.

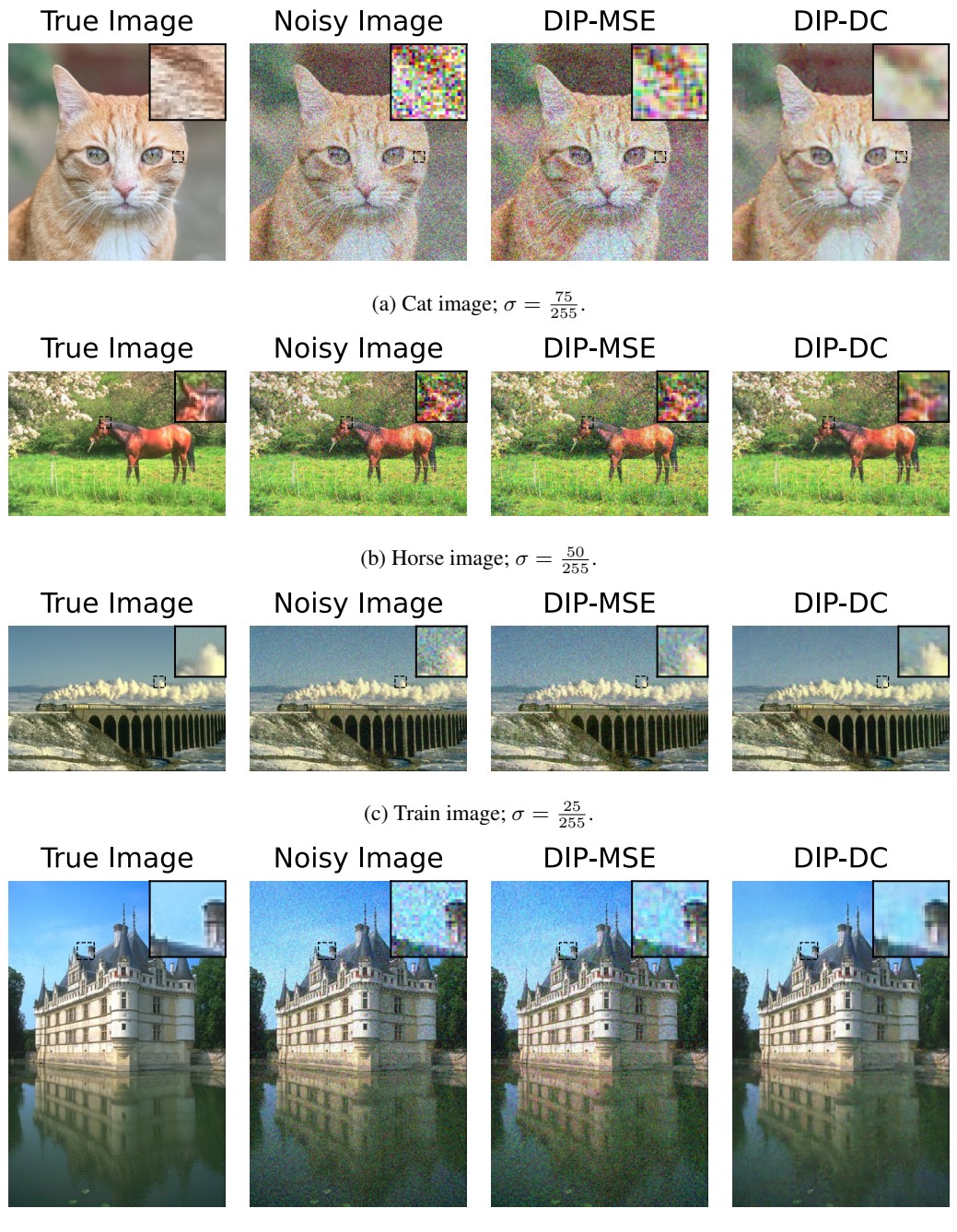

Figure E9: Further example images denoised with DIP-DC and DIP-MSE. Images shown are after 10,000 iterations of DIP.

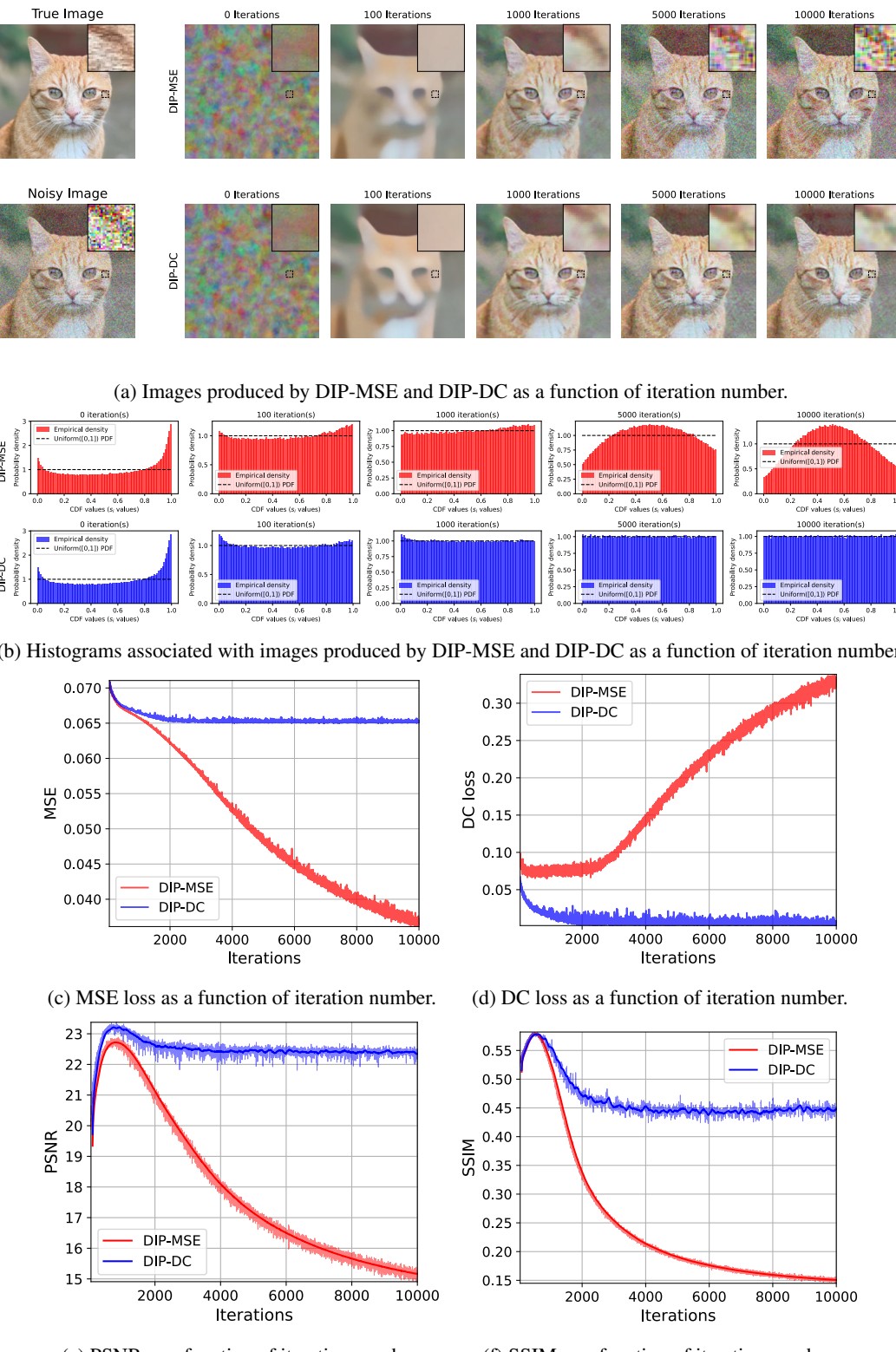

(a) Images produced by DIP-MSE and DIP-DC as a function of iteration number.

(b) Histograms associated with images produced by DIP-MSE and DIP-DC as a function of iteration number.

(c) MSE loss as a function of iteration number.

(d) DC loss as a function of iteration number.

(e) PSNR as a function of iteration number.

(f) SSIM as a function of iteration number.

Figure E10: Results for DIP-MSE and DIP-DC on cat image.

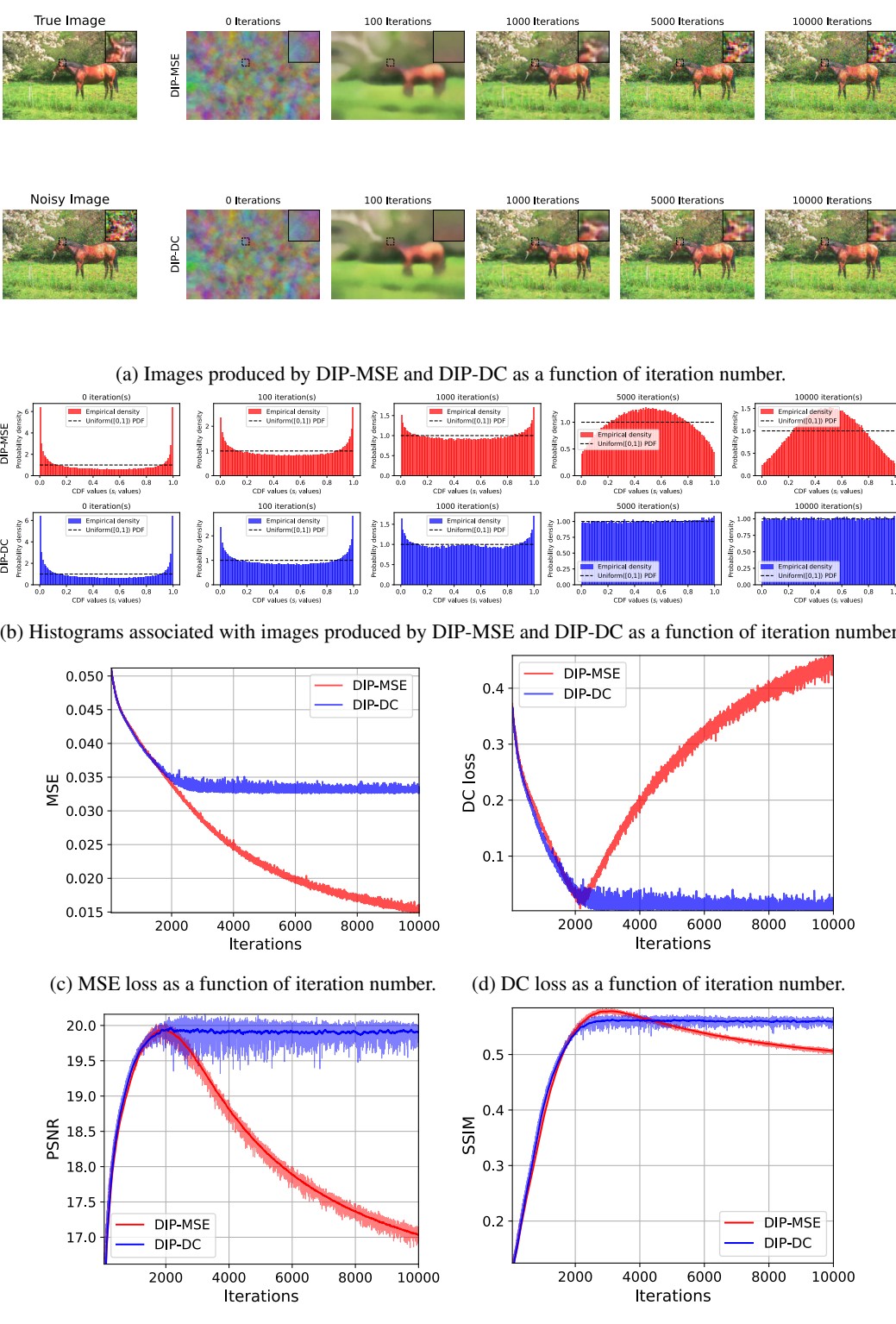

(a) Images produced by DIP-MSE and DIP-DC as a function of iteration number.

(b) Histograms associated with images produced by DIP-MSE and DIP-DC as a function of iteration number.

(c) MSE loss as a function of iteration number.

(d) DC loss as a function of iteration number.

(e) PSNR as a function of iteration number.

(f) SSIM as a function of iteration number.

Figure E11: Results for DIP-MSE and DIP-DC on horse image.

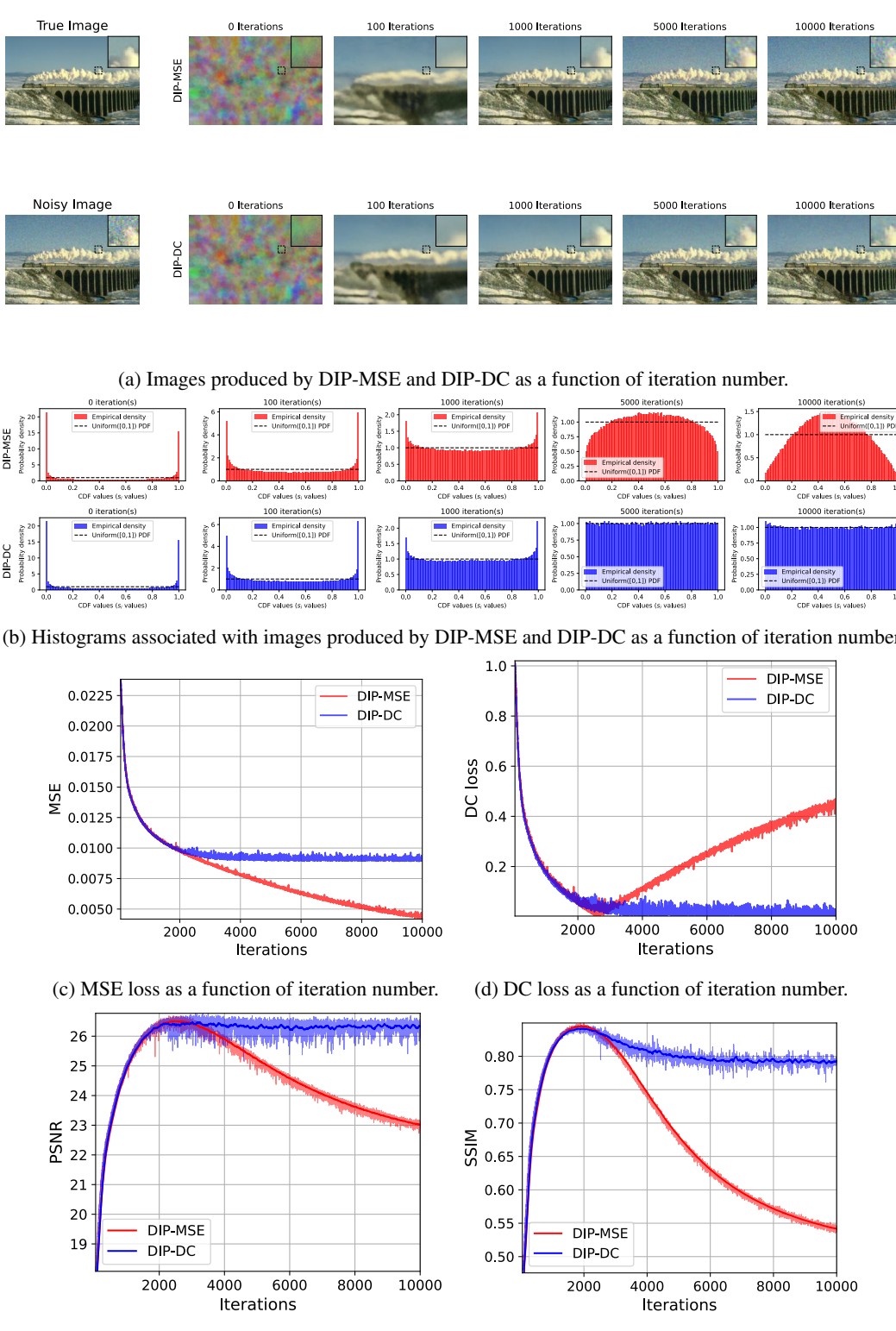

(a) Images produced by DIP-MSE and DIP-DC as a function of iteration number.

(b) Histograms associated with images produced by DIP-MSE and DIP-DC as a function of iteration number.

(c) MSE loss as a function of iteration number.

(d) DC loss as a function of iteration number.

(e) PSNR as a function of iteration number.

(f) SSIM as a function of iteration number.

Figure E12: Results for DIP-MSE and DIP-DC on train image.

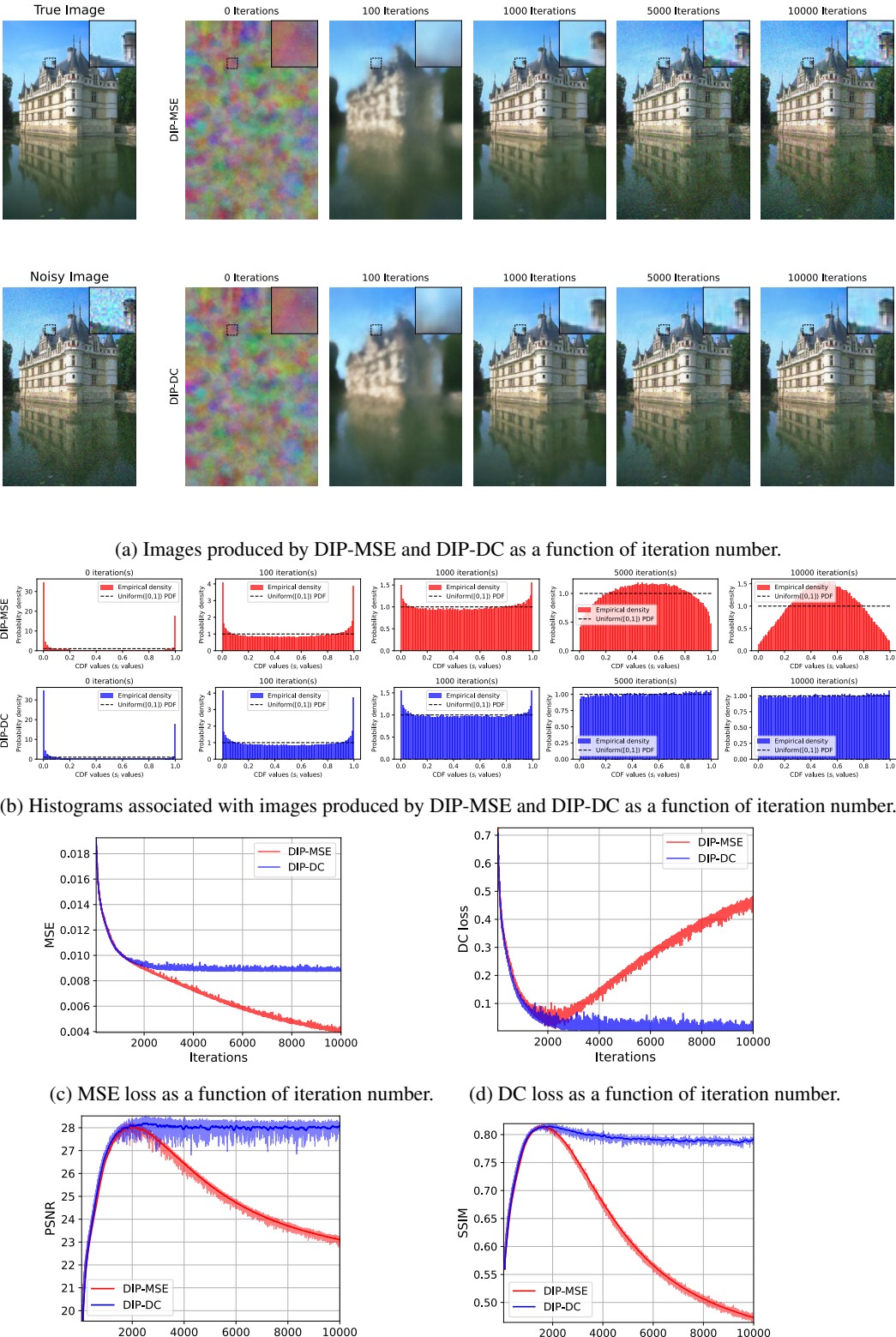

(a) Images produced by DIP-MSE and DIP-DC as a function of iteration number.

(b) Histograms associated with images produced by DIP-MSE and DIP-DC as a function of iteration number.

(c) MSE loss as a function of iteration number.

(d) DC loss as a function of iteration number.

(e) PSNR as a function of iteration number.

(f) SSIM as a function of iteration number.

Figure E13: Results for DIP-MSE and DIP-DC on castle image.

### E.5 EFFECT OF MISSPECIFIED NOISE

We considered the behavior of DC loss when the assumed noise model is systematically violated. For all experiments, DC loss assumed the same noise model of a clipped Gaussian with $\sigma_{\text{assumed}} = \frac{75}{255}$, matching the setting used in the main paper.

To probe robustness, we generated data from three alternative families of noise distributions, each controlled by a single scalar parameter that smoothly modulates the degree of mismatch. In every case, the clean image and optimization settings were kept fixed.

**Outliers** We first investigated the effect of sparse, heavy-tailed corruptions. For each pixel, with probability $p$, we inflated the true noise standard deviation to $\sigma_{\text{true}} = 5\sigma_{\text{assumed}}$, drawing the measurement from a heavier-tailed distribution than the one assumed by the loss.

We varied $p$ from 0% to 25%. Figure E14 shows that the performance of DC loss degraded gracefully over this range, with similar degradation observed relative to MSE loss.

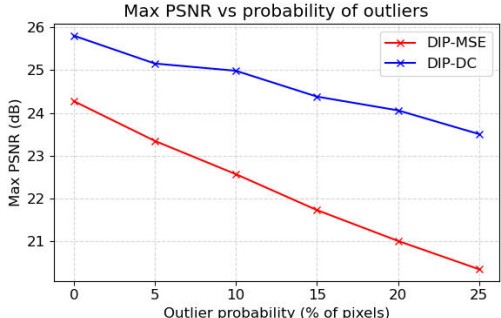

(a) Max PSNR (with respect to iteration) as a function of $p$, the probability that a pixel's measurement is drawn from an outlier heavy-tailed distribution.

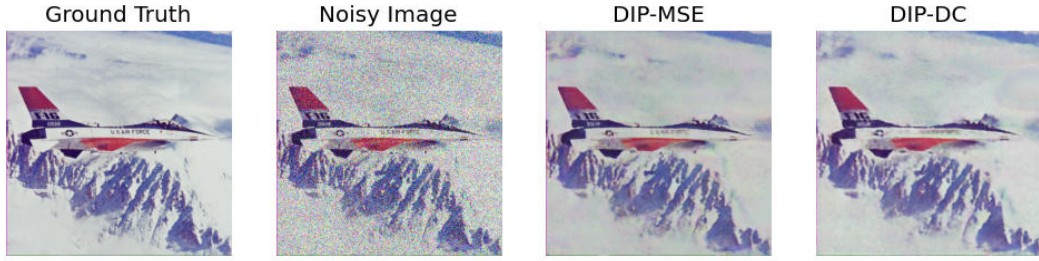

(b) Images corresponding to $p = 25\%$ on the above chart.

Figure E14: Results for outlier ablation study, investigating the effect of mispecifying the noise distribution for $p$ pixels.

**Correlated noise** We next examined noise with spatial correlation structure. Here, we drew an i.i.d. Gaussian field $\epsilon \sim \mathcal{N}(0, \sigma^2_{\text{assumed}})$ of the same shape as the image, and then spatially smoothed it using an isotropic Gaussian kernel. Specifically, letting $g_\tau$ denote a Gaussian blur with standard deviation $\tau$ pixels, the true noise was generated as

$$n_{\text{true}} = g_\tau * \epsilon.$$

The parameter $\tau \geq 0$ controls the strength of correlation: $\tau = 0$ recovers the matched i.i.d. model, while increasing $\tau$ broadens the correlation structure, thereby violating the independence assumption

underlying the MSE and DC losses. After smoothing, $n_{\text{true}}$ was rescaled to have standard deviation $\sigma_{\text{assumed}}$, ensuring that only the correlation structure was altered, not the noise amplitude.

We varied $\tau$ from 0 to 2.5. Figure E15 shows that the benefit of DC loss relative to MSE loss degrades severely when correlation is introduced. However, the performance of our method nonetheless degrades at a similar rate to MSE loss in this setting.

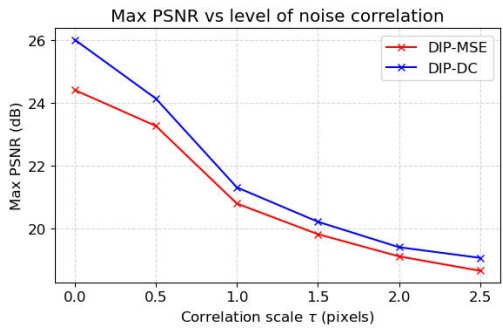

(a) Max PSNR (with respect to iteration) as a function of $\tau$, the correlation scale.

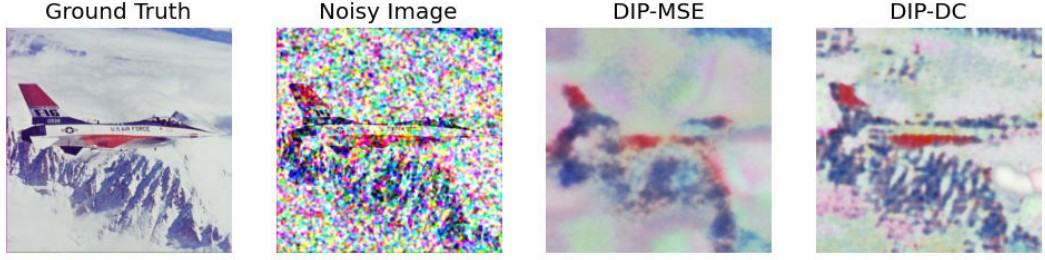

(b) Images corresponding to $\tau = 2.5$ on the above chart.

Figure E15: Results for correlated noise ablation study, investigating the effect of mispecifying the noise distribution by introducing correlated noise.

**Heteroscedastic noise** Thirdly, we tested a form of noise whose variance depends on the underlying clean intensity.

For a pixel with clean value $x \in [0, 1]$, we defined the true noise level as

$$\sigma_{\text{true}}(x) = \sigma_{\text{assumed}} \big(1 + \alpha(2x - 1)\big),$$

where $\alpha \in [-1, 1]$ determines the severity of heteroscedasticity. Thus $\alpha = 0$ recovers the matched homoscedastic model, while increasing $|\alpha|$ creates regions of inflated or reduced variance depending on image brightness. For each pixel we then sampled

$$n_{\text{true}}(x) \sim \mathcal{N}\big(0, \, \sigma_{\text{true}}(x)^2\big),$$

and constructed the observed image as $y = x + n_{\text{true}}(x)$ with clipping to $[0, 1]$. The parameter $\alpha$ therefore provides a single scalar that smoothly controls the degree of mismatch, analogous to the outlier rate $p$ and correlation scale $\tau$ above.

Our ablation test again demonstrated graceful degradation in the performance of both DC loss and MSE loss with respect to the noise modelling mismatch.

**Conclusion** Taken together, these three experiments explore mismatches in tail behavior, spatial correlation, and variance structure. Each setting introduces a controlled departure from the assumed

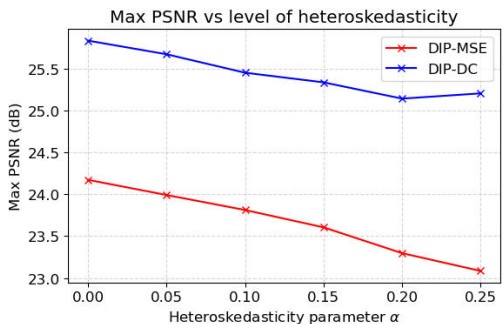

(a) Max PSNR (with respect to iteration) as a function of $\alpha$, the level of heteroscedasticity.

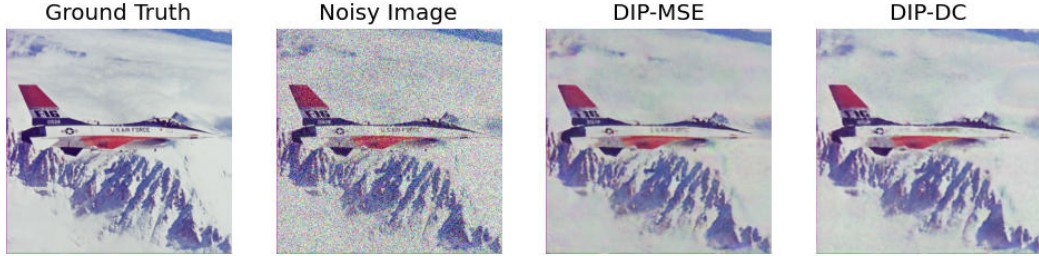

(b) Images corresponding to $\alpha = 0.25$ on the above chart.

Figure E16: Results for heteroscedastic noise ablation study, investigating the effect of mispecifying the noise distribution by introducing heteroscedastic noise.

noise model, allowing us to assess how reconstruction quality evolves as the data deviate from the idealised conditions encoded in the loss.

Under the noise model mismatch settings studied, DC loss's performance degraded gracefully and comparatively to MSE loss.

# F  PET RECONSTRUCTION APPLICATION

## F.1  EXPERIMENTAL DESIGN

A ground truth BrainWeb phantom was generated (Cocosco et al., 1997) with a width and height of $N_d = 256$ voxels, and multiplied by a scale factor of 0.1 (resulting in a mean voxel intensity of 2.42). A forward model was specified using ParallelProj (Schramm and Thielemans, 2023), approximating the dimensions of a single ring of a Siemens Biograph mMR PET scanner. Lines of response were over-sampled by a factor of 4, and pairs of lines were then added together, so as to simulate thicker lines of response, ensuring the projected rays intersected all voxels in the image space (which avoided a null space in the projector for smaller voxel sizes).

This setup produced sinograms with 252 projection angles and 688 radial bins (for a total of 173376 lines of response, of which 54773 intersected non-zero voxels in the $N_d = 256$ ground truth phantom).

For the case of regularized PET image reconstruction (corresponding to Section 5.2.1), the intensity of the phantom was 25% of the unregularized case, to simulate a lower-dose acquisition. The edge-preserving TV penalty used is defined in subsection F.4, and $N_d = 256$ was used only.

For each of the algorithms NLL-Adam, DC-Adam and MLEM, our starting image iteration was an image of uniform intensity equal to the mean intensity of non-zero voxels in the phantom.

For DC-Adam and NLL-Adam only, at each iteration, we multiplied the image estimate by a mask representing the convex hull of the non-zero voxels in the phantom (information that can be derived physically in a real PET scan from the attenuation map). Final images were passed through a ReLU function to remove negative values (which are not physically realistic for a PET image). We preconditioned each gradient update by dividing by the sensitivity image (i.e. the backprojected image obtained by backprojecting a sinogram of ones).

For the experiments in subsection F.3, the total sum of counts in the ground truth phantom was held constant as the width of the phantom was varied from $N_d = 64$ to $N_d = 512$. In this appendix, learning rate 0.01 was used for $N_d = 512$, 0.005 for $N_d = 256$ and 0.0025 for $N_d \leq 128$.

Experiments were conducted with a 24GB Nvidia GeForce RTX 3090 GPU and run for 10,000 iterations; for the largest image size $512 \times 512$, running DC-Adam, NLL-Adam and MLEM consecutively took $\sim 30$ minutes.

## F.2  ASSESSING THE DISTRIBUTIONAL CONSISTENCY OF THE CLEAN PHANTOM

In contrast to the Gaussian setting, with a Poisson noise model we have a discrete CDF. As a result, the theoretical guarantee of the probability integral transform does not strictly hold. However, the true image estimate should still induce an approximately uniform histogram of CDF values, and we investigate this assumption in this subsection.

Figure F1 explores how changing the simulated radioactivity level (often referred to as the dose, affecting the total number of measured counts) influences the histogram of CDF values for both the ground-truth sinogram and the corresponding noisy sinogram. We also include an alternative reference distribution: the CDF values of Poisson-distributed samples generated by treating each noisy measurement as the Poisson mean. All count levels in this figure are expressed relative to the baseline count level used for the PET results in the main paper.

These results demonstrate that targeting a uniform histogram of CDFs is an appropriate approximation for higher doses (analogously to how the Normal distribution becomes a good approximation for the Poisson distribution for higher means). Further work is required to determine how to best adjust for the discrete nature of the Poisson distribution at lower doses, although the alignment shown between the alternative target (blue line) in Figure F1 and the true clean histogram is promising, as is the option of using the randomized PIT. (We hypothesize that the peaks at 0.5 and 1.0 are primarily due to very low measured values of 0 and 1, where the Poisson distribution is not well approximated by any continuous distribution.)

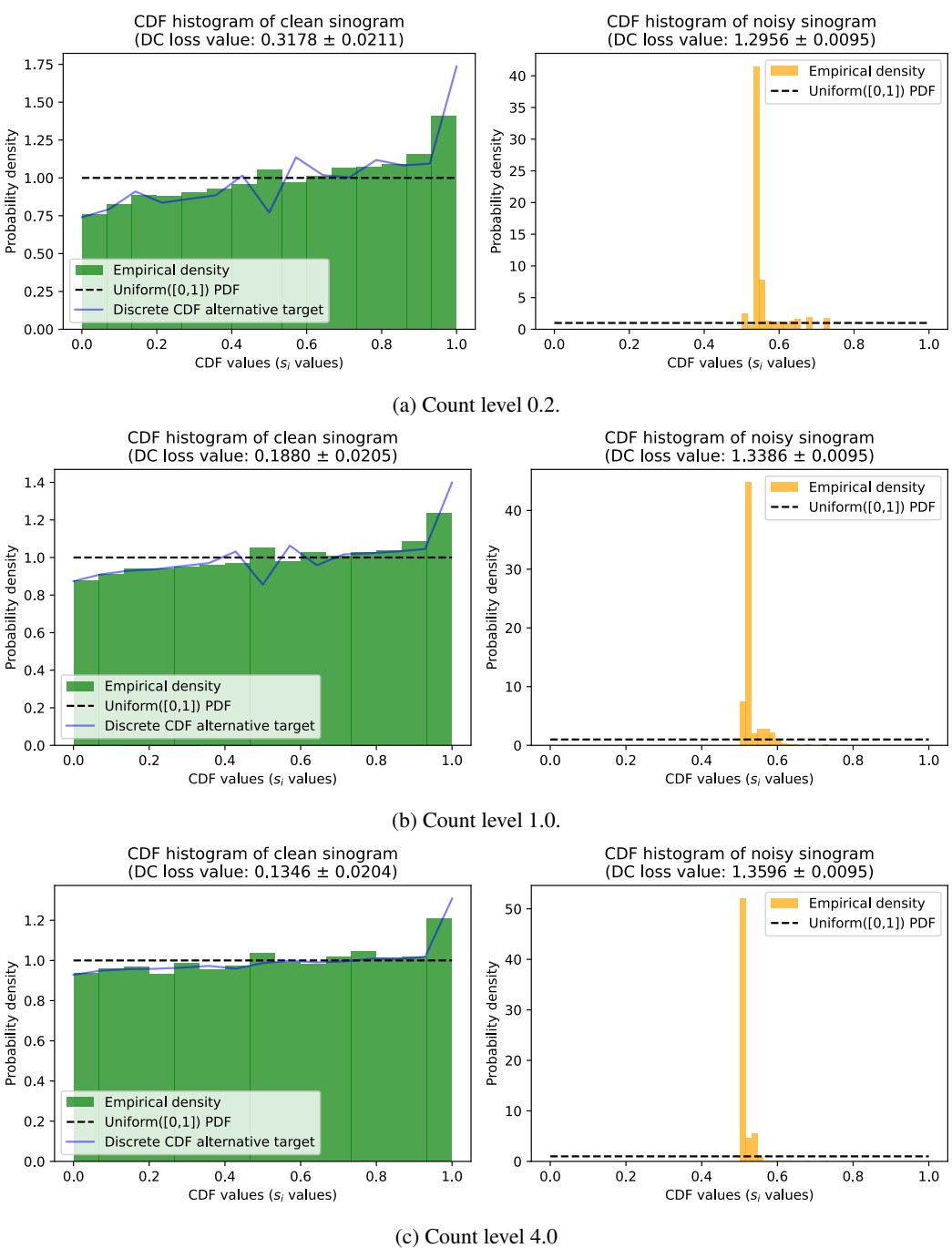

Figure F1: Investigating the DC loss values obtained and the uniformity of CDF histograms in the ideal and worst cases of evaluating the DC loss against the true measured data or Poisson-noisy measured data respectively. Subplots show the effect of increasing the number of measured counts (and thereby reducing the impact of noise introduced by the Poisson noise model). Uncertainty values are given as $1.96\times$ the standard deviation over 100 evaluations of the (stochastic) loss on different instantiations of noise.

## F.3 VARYING VOXEL SIZE AND THE NUMBER OF PARAMETERS TO ESTIMATE

We investigated the effect of over-parameterization on the PET reconstruction process with the DC loss, by varying the width of the phantom from $N_d = 64$ to $N_d = 512$. Figures F2, F3, F4, and F5 show the results of these experiments for $N_d = 64$, 128, 256 and 512 respectively.

It is clear that as the number of voxels increases (i.e. as we move into an over-parameterized regime), the ability of traditional algorithms to overfit the noisy data increases, thereby leading to more noisy reconstructions without early stopping. This pattern is particularly obvious in the histograms for MLEM and NLL-Adam in Figures F2–F5, which become less uniform as the voxel size decreases.

NLL-Adam and DC-Adam do not build in non-negativity as a prior, whereas MLEM does, which may explain its better peak metric values for NRMSE and SSIM.

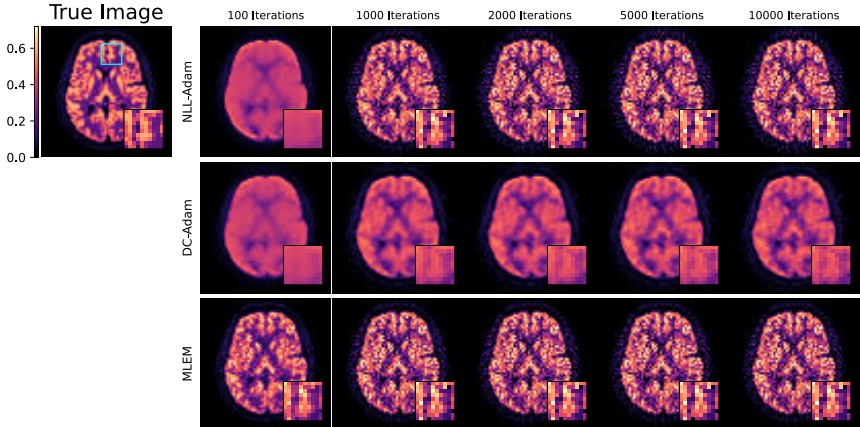

(a) Images produced by PET reconstruction algorithms as a function of iteration number.

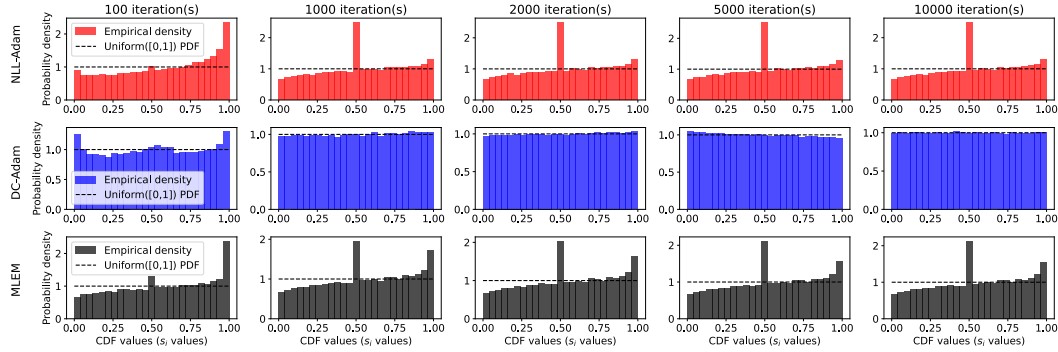

(b) Histograms associated with PET reconstruction outputs as a function of iteration number.

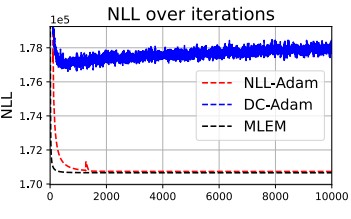

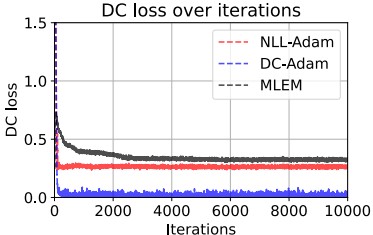

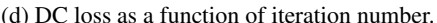

(c) NLL loss as a function of iteration number.

(d) DC loss as a function of iteration number.

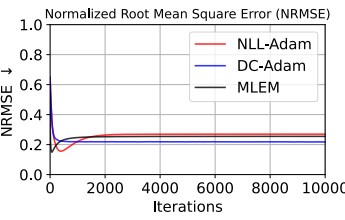

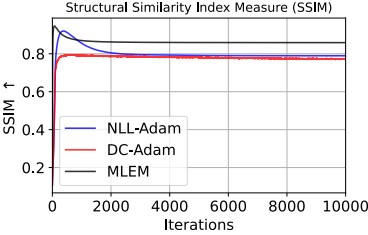

(e) NRMSE as a function of iteration number.

(f) SSIM as a function of iteration number.

Figure F2: Results for PET reconstruction with image size $N_d = 64$.

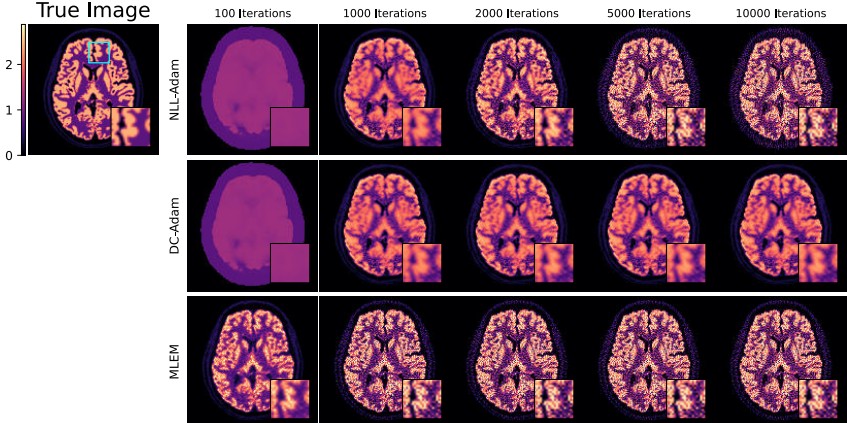

(a) Images produced by PET reconstruction algorithms as a function of iteration number.

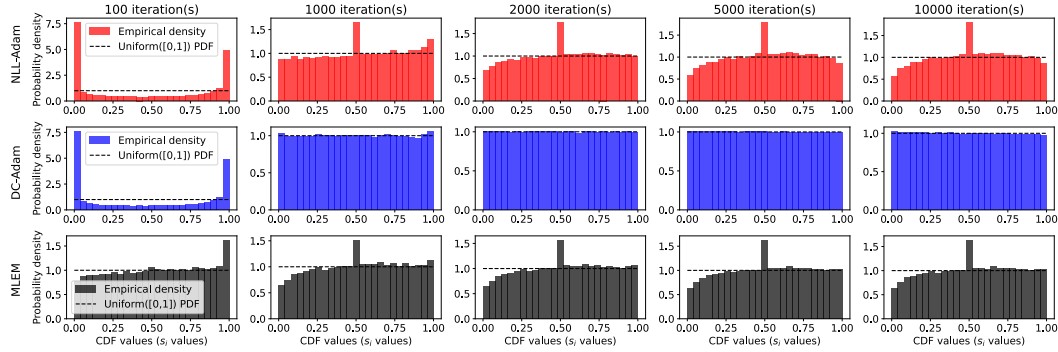

(b) Histograms associated with PET reconstruction outputs as a function of iteration number.

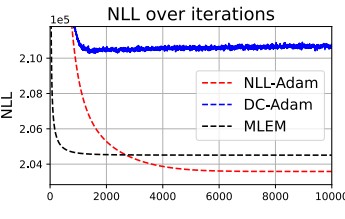

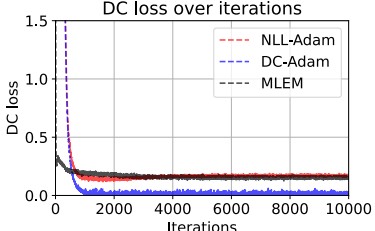

(c) NLL loss as a function of iteration number.

(d) DC loss as a function of iteration number.

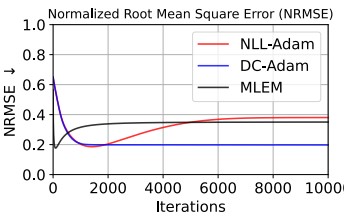

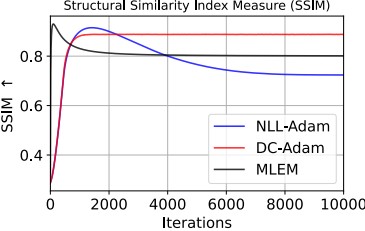

(e) NRMSE as a function of iteration number.

(f) SSIM as a function of iteration number.

Figure F3: Results for PET reconstruction with image size $N_d = 128$.

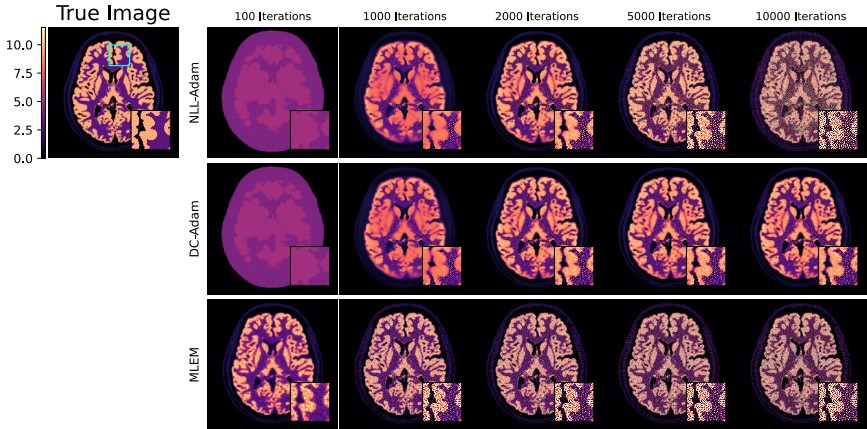

(a) Images produced by PET reconstruction algorithms as a function of iteration number.

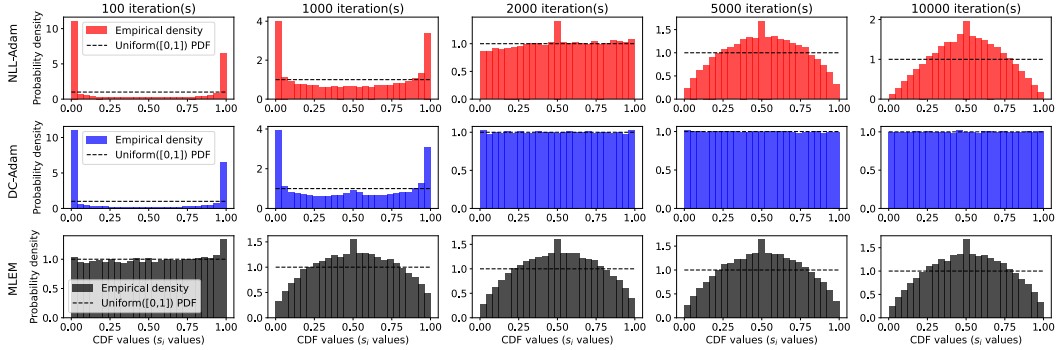

(b) Histograms associated with PET reconstruction outputs as a function of iteration number.

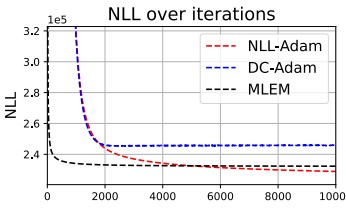

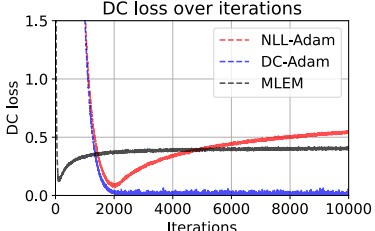

(c) NLL loss as a function of iteration number.    (d) DC loss as a function of iteration number.

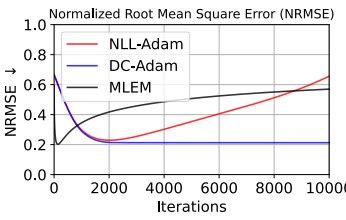

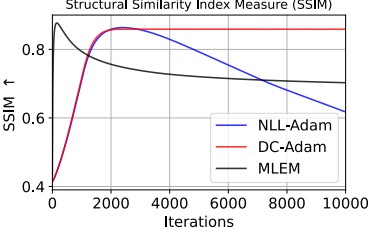

(e) NRMSE as a function of iteration number.    (f) SSIM as a function of iteration number.

Figure F4: Results for PET reconstruction with image size $N_d = 256$.

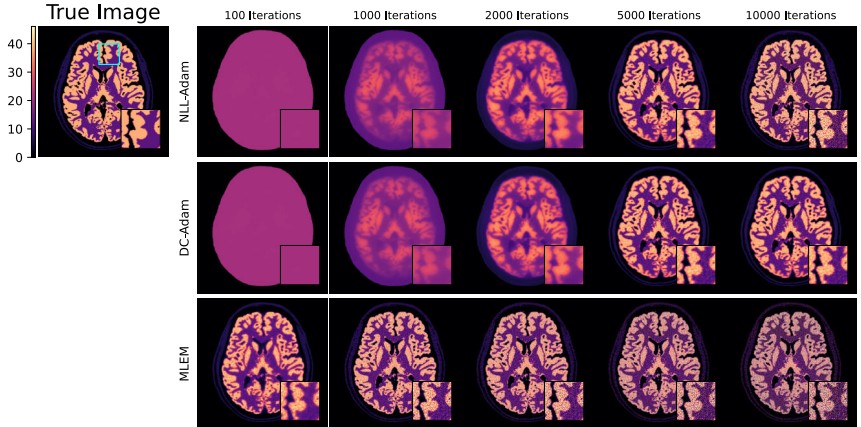

(a) Images produced by PET reconstruction algorithms as a function of iteration number.

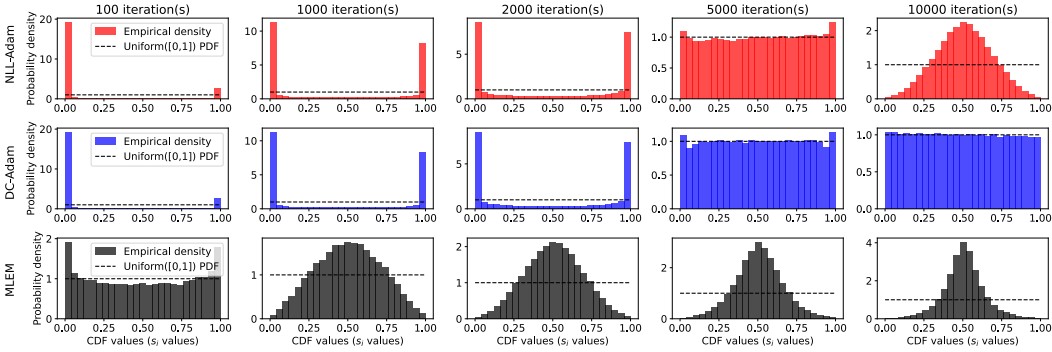

(b) Histograms associated with PET reconstruction outputs as a function of iteration number.

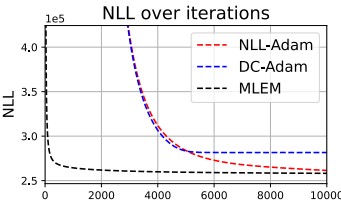

(c) NLL loss as a function of iteration number.

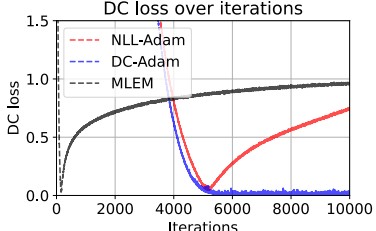
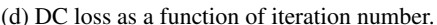

(d) DC loss as a function of iteration number.

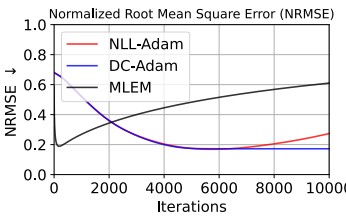

(e) NRMSE as a function of iteration number.

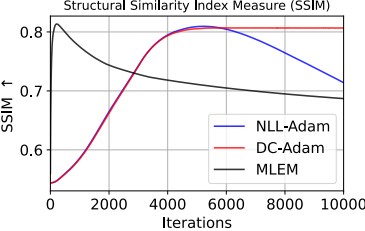

(f) SSIM as a function of iteration number.

Figure F5: Results for PET reconstruction with image size $N_d = 512$.

## F.4 FURTHER RESULTS WITH A REGULARIZED OBJECTIVE FUNCTION

We defined our edge-preserving TV prior as

$$\text{EPTV}(\mathbf{x}) = \frac{1}{N} \sum_{i,j} w_{i,j} \left( |\mathbf{x}_{i,j+1} - \mathbf{x}_{i,j}| + |\mathbf{x}_{i+1,j} - \mathbf{x}_{i,j}| \right), \tag{76}$$

with edge-aware weights defined as

$$w_{i,j} = \frac{1}{1 + \left( \frac{\sqrt{(\mathbf{x}_{i,j+1} - \mathbf{x}_{i,j})^2 + (\mathbf{x}_{i+1,j} - \mathbf{x}_{i,j})^2 + \varepsilon}}{\kappa} \right)^2} . \tag{77}$$

In this work, $\kappa = 0.1$ and $\varepsilon = 1e-8$ were used. The width and height of each image was $N_d = 256$.

In Figure F6, we show the edge-preserving TV prior values obtained by image estimates found by DC+TV and NLL+TV against NLL, DC loss and error values (NRMSE). In line with what was expected, these charts exhibit different trends for DC+TV and NLL+TV.

Figure F6a shows that for high levels of regularization (i.e. large values of $\beta$) both DC+TV and NLL+TV exhibit higher NRMSE ($\sim 40\%$ or greater), as the strong regularization overpowers the data-fidelity term, seeking to drive the edge-preserving TV prior (penalty) value closer to zero. Interestingly, for DC+TV, we find many different image estimates (for a range of different regularization strengths $\beta$) that give an NRMSE close to DC+TV's minimum NRMSE, and only once an image estimate's value for edge-preserving TV (i.e. the prior or penalty value) dips too low does it come into conflict with the data-fidelity constraint, resulting in worse NRMSE. In contrast, with NLL+TV, the NRMSE changes with greater sensitivity to the regularization strength $\beta$, with only a very limited range of $\beta$ values for which minimum NRMSE is achieved. It is precisely this lack of conflict between the DC loss data-fidelity and the regularization prior which we theorize has led to the improved NRMSE values obtained by DC+TV in Figures 8a and F6a.

With lower regularization strengths, as expected, DC+TV retains a low value of the NRMSE while NLL+TV exhibits increasing NRMSE (as overfitting to noise occurs). Figure F6b displays a similar chart, now showing DC loss against prior EPTV values for a range of image estimates obtained for various $\beta$ values. Both methods show a similar trend to that seen in Figure F6a (presumably because DC loss and NRMSE are correlated)[7].

For completeness, Figure F6c also shows a similar trend with NLL in place of DC loss. It is noteworthy (and in line with concepts discussed previously) that the estimates found by DC+TV exhibit relatively constant NLL loss values as regularization strength increases, before leaving this "stable region" when balancing data-fidelity and regularization is no longer feasible at higher $\beta$ values. This point occurs at very similar $\beta$ values in Figure F6c as in Figure F6b.

The images shown with varying regularization strengths $\beta$ in Figure F7 underline these trends. With low (or no) regularization strength, DC+TV finds a much better estimate than NLL+TV, and it retains some improvement still at optimum regularization strengths (as seen in Figure 8). At high regularization strengths, both methods over-smooth and succumb to artifacts induced by the prior.

---

[7]The trend for NLL+TV surprisingly shows a relatively low prior penalty for the no regularization case. Considering the images in Figure F7, we attribute this to an imperfect prior not perfectly correlating with error. The prior value for highly regularized images is still lower than for the unregularized case. Clearly, the prior is having the desired effect, as increasing it reduces overfitting and promotes smoothness (with edges respected). Further work could investigate this phenomenon with different regularization hyperparameters and penalty choices).

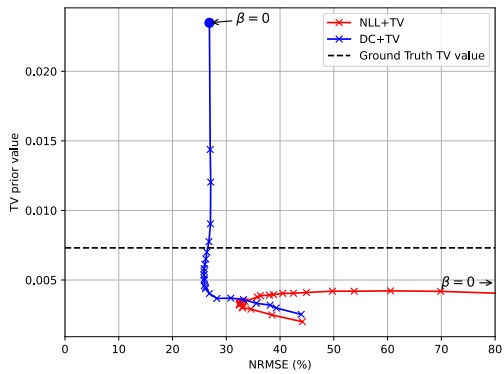

(a) Value of edge-preserving TV prior obtained against NRMSE value obtained by DC+TV estimates and NLL+TV estimates.

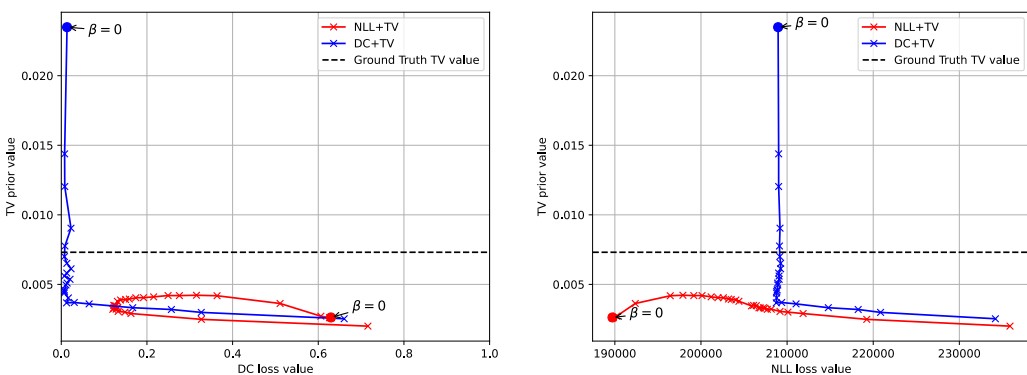

(b) Value of edge-preserving TV prior obtained against DC loss value obtained by DC+TV estimates and NLL+TV estimates.

(c) Value of edge-preserving TV prior obtained against NLL loss value obtained by DC+TV estimates and NLL+TV estimates.

Figure F6: Edge-preserving TV prior values obtained by image estimates found by DC+TV and NLL+TV against NLL, DC loss and error values.

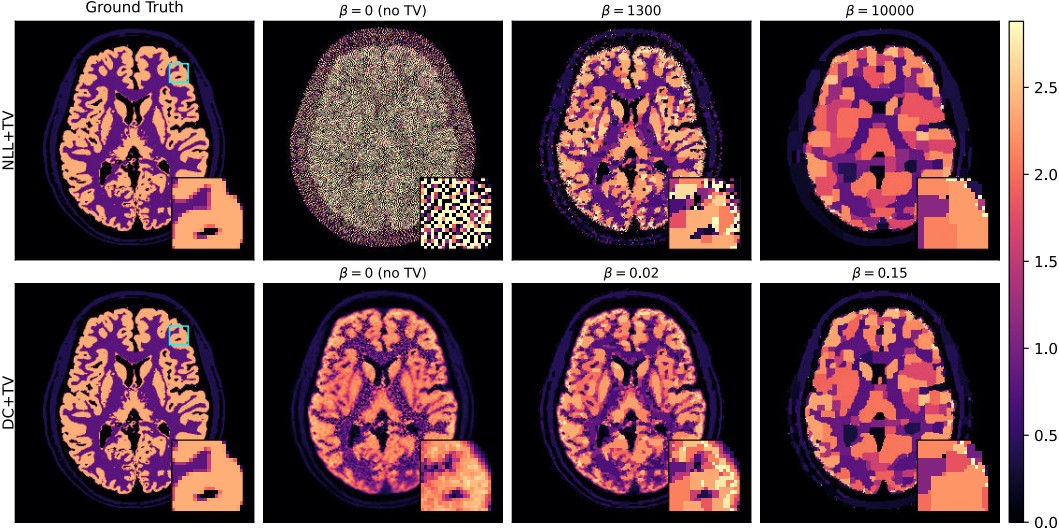

Figure F7: Example images produced with DC+TV and NLL+TV for regularized PET image reconstruction with an edge-preserving total variation penalty, at increasing regularization strengths $\beta$. First column: ground truth images. Second column: image results with no explicit regularization. Third column: image results with near-optimal choice of $\beta$. Fourth column: indicative image results with $\beta$ chosen to be "too large", resulting in regularization-induced artifacts for both methods.

## F.5  REAL PET DATA: EXPERIMENTAL SETUP

This section provides the full experimental details for the real-data reconstruction reported in Section 5.2.2. We reconstructed a single 3D clinical PET brain scan acquired on a Siemens Biograph mMR system (non-time-of-flight), using the same Adam-based optimization framework as in the synthetic PET experiments but with an approximation of the full physical forward model used clinically.

**Dataset and corrections.**   The dataset consisted of approximately 70 million lines of response, axially compressed into span-11 sinograms of dimension $252 \times 344 \times 837$. Vendor-provided software supplied the corrective factors required for forward modelling: attenuation correction $\mathbf{L}$ (from the MR-derived $\mu$-map), detector normalization $\mathbf{N}$, scatter estimation $\mathbf{s}$, and randoms estimation $\mathbf{r}$.

**Forward model.**   We implemented the tomographic projection operator $\mathbf{P}$ using the open-source ParallelProj library (Schramm and Thielemans, 2023), parameterised with scanner-specific geometry to match the real system. The complete forward model $\mathbf{A}$ combined the projection operator with all corrections:

$$\mathbf{Ax} = \mathbf{NLC_{11}Px} \; + \; \mathbf{r} \; + \; \mathbf{s},$$

where $\mathbf{C}_{11}$ denotes the manufacturer's span-1 to span-11 axial compression matrix.

We restricted consideration of sinogram bins to those lines-of-response that intersect the $\mu$-map, i.e. lines of response that actually intersect matter.

**Reconstruction grid.**   All reconstructions were performed directly on the same voxel grid as the vendor reconstruction: $128 \times 128 \times 120$ voxels with physical voxel dimensions $(2.08626\,\text{mm}) \times (2.08626\,\text{mm}) \times (2.03125\,\text{mm})$.

**Optimization.**   The initial image was defined as a uniform field on a binary support mask derived from the $\mu$-map (after hole-filling). Voxels outside the support were assigned a constant value equal to $1\%$ of the mean activity inside the mask. Although not required for reconstruction, this warm start stabilised the earliest iterations and expedited prototyping.

We compared Poisson NLL and DC as alternative data-fidelity terms, both optimised with Adam using a learning rate of $10^{-4}$ and a positivity projection applied after each update. In line with standard practice for likelihood-based PET reconstruction, we accelerated computation by splitting the sinogram into 21 ordered subsets and cycling through them during optimization. All methods were run for 10,000 iterations to expose the characteristic late-iteration behavior of each loss.

For both NLL and DC, the gradient was preconditioned by dividing by a stabilised sensitivity image $A^\top \mathbf{1} + 10^{-3}$, matching the effective diagonal scaling used in the MLEM algorithm.

NLL-Adam reconstruction took 0.037 sec/it compared to 0.057 sec/it for the DC-Adam reconstruction.

**Notes.**   All reconstructions operated on the same PET grid, used identical physical corrections, and differed only in the choice of data-fidelity term. The vendor image in Fig. 9 was the scanner's MLEM reconstruction, which may have included additional proprietary corrections.

## G  PLUG-AND-PLAY WITH A LEARNED PRIOR

To demonstrate that DC loss applies cleanly to modern learned priors, we carried out an additional experiment on image deblurring/denoising using a plug-and-play (PnP) framework with a pretrained denoiser prior. Concretely, we consider a standard Gaussian deblurring model

$$\mathbf{y} = \mathbf{A} * \mathbf{x}_{\text{true}} + \boldsymbol{\eta}, \qquad \boldsymbol{\eta} \sim \mathcal{N}(0, \sigma^2 I), \tag{78}$$

where $\mathbf{x}_{\text{true}}$ is a clean grayscale natural image, $\mathbf{A}$ is a spatially invariant Gaussian blur kernel, and $\boldsymbol{\eta}$ is additive white Gaussian noise. Pixel values are scaled to $[0, 1]$ and clipped after noise is added, as in the DIP experiments.

**PnP–ADMM with a DnCNN prior.**  We adopt a plug-and-play ADMM scheme in which the prior is represented by an off-the-shelf DnCNN denoiser (Zhang et al., 2021), while the data term is either the usual quadratic loss or DC loss. Introducing auxiliary and dual variables $v$ and $u$, we iterate

$$\mathbf{x}^{t+1} \approx \arg\min_{\mathbf{x}} \left\{ \mathcal{L}_{\text{data}}(\mathbf{A} * \mathbf{x}; \mathbf{y}) + \frac{\rho}{2} \|\mathbf{x} - v^t + u^t\|_2^2 \right\}, \tag{79}$$

$$v^{t+1} = D_\tau(\mathbf{x}^{t+1} + u^t), \tag{80}$$

$$u^{t+1} = u^t + \mathbf{x}^{t+1} - v^{t+1}, \tag{81}$$

where $\rho > 0$ is the ADMM penalty parameter and $D_\tau$ is a DnCNN denoiser trained for a nominal Gaussian noise level $\tau$. The $x$–update in equation 79 is carried out by a small fixed number of gradient steps with Adam, while the $v$–update equation 80 is a pure denoising step (no backpropagation through $D_\tau$).

**Data terms: MSE vs. DC.**  We compare two choices for the data-fidelity term:

$$\mathcal{L}_{\text{MSE}}(\mathbf{A} * \mathbf{x}; \mathbf{y}) = \frac{1}{2\sigma^2} \|\mathbf{A} * \mathbf{x} - \mathbf{y}\|_2^2, \tag{82}$$

$$\mathcal{L}_{\text{DC}}(\mathbf{A} * \mathbf{x}; \mathbf{y}) = L_{\text{DC-Gauss}}(\mathbf{A} * \mathbf{x}, \mathbf{y}; \sigma), \tag{83}$$

where $L_{\text{DC-Gauss}}$ is the clipped-Gaussian DC loss from the DIP experiments, applied to the flattened residuals $(\mathbf{A} * \mathbf{x} - \mathbf{y})$ with a Gaussian noise model of variance $\sigma^2$ and intensities clipped to $[0, 1]$. In the implementation, the DC loss is evaluated directly on the $(B, C, H, W)$ tensors by flattening all spatial and channel dimensions.

We refer to the two PnP variants as:

- **PnP–MSE**: ADMM with quadratic data term $\mathcal{L}_{\text{MSE}}$.
- **PnP–DC**: ADMM with DC data term $\mathcal{L}_{\text{DC}}$.

**Experimental setup.**  We use the same F16 image as in the DIP experiments, converted to single-channel grayscale. The forward operator $\mathbf{A}$ is a depthwise Gaussian convolution with kernel size $15 \times 15$ and kernel standard deviation 1.6 pixels. We simulate clean data $\mathbf{y}_{\text{clean}} = \mathbf{A} * \mathbf{x}_{\text{true}}$ and then add i.i.d. Gaussian noise $\boldsymbol{\eta} \sim \mathcal{N}(0, \sigma^2 I)$ with $\sigma = 25/255$, clipping the result back to $[0, 1]$.

The PnP–ADMM scheme is initialized with $\mathbf{x}^0 = v^0 = \mathbf{y}$ and $u^0 = 0$. We run 2500 ADMM iterations, and use 5 inner gradient steps with learning rate $10^{-3}$ for the $\mathbf{x}$–update. The denoiser $D_\tau$ is a standard 17-layer DnCNN trained for $\tau = 25/255$ and kept fixed during the experiment (Zhang et al., 2021). Reconstruction quality is measured against $\mathbf{x}_{\text{true}}$ using PSNR and SSIM.

**Results.**  We varied the hyperparameter $\rho$ from 0.1 to 1000, carrying out a PnP reconstruction for each of MSE and DC loss. Figure F8 shows the PSNR achieved by each reconstruction against $\rho$.

PnP with DC loss achieves a higher peak PSNR than with MSE (27.5 dB versus 26.9 dB; images shown in Figure F9), and the DC curve is markedly flatter, indicating substantially greater robustness to the choice of $\rho$.

This behavior follows directly from the properties of DC loss. In PnP–MSE, higher $\rho$ places more weight on the denoiser prior, while lower $\rho$ places more weight on the data fidelity. With an MSE

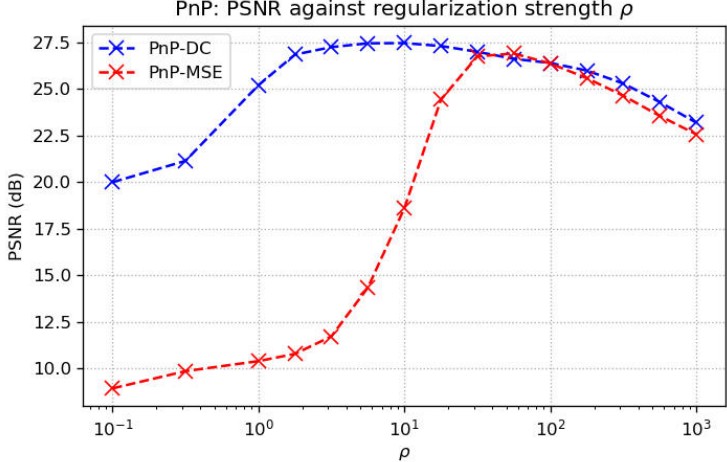

Figure F8: PSNR values for PnP reconstruction with varied regularization strength $\rho$, for PnP-DC and PnP-MSE.

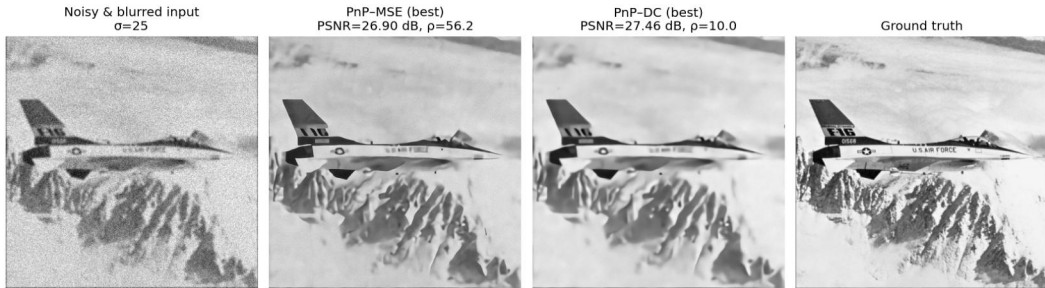

Figure F9: Images obtained with maximum PSNR when reconstructing from blurred and noisy images, for PnP-MSE and PnP-DC.

data term, varying $\rho$ changes this balance in a way that often leads to overfitting the specific noise realization in $\mathbf{y}$ (when the data term dominates) or oversmoothing (when the denoiser dominates). As a result, PnP–MSE exhibits a narrow PSNR peak.

In contrast, DC loss stops rewarding further reductions in the residual once its distribution matches the noise model. Consequently, the $x$–update no longer chases the measured noise, regardless of whether $\rho$ emphasises the data term or the prior. This makes the PnP–DC fixed point far less sensitive to $\rho$ and yields both the higher optimal PSNR and the observed hyperparameter stability.

This reinforces the earlier observation that DC behaves like MSE when far from the solution but self-regularizes once noise consistency is reached, making it naturally compatible with PnP methods. Further work remains to explore these effects in a wider range of modern learned denoising frameworks.

# H TIMING

## H.1 EVALUATION SPEED OF DC LOSS

We benchmarked the computational cost of evaluating the DC loss against conventional pointwise objectives (NLL and MSE) for both Gaussian and Poisson noise models. Timings were obtained on a single NVIDIA RTX 3090 GPU, averaged over 1000 runs. The same pairs of uniform random arrays (in the range [0,1]) were used to measure each pair of methods (DC / NLL or DC / MSE).

Results are reported separately for forward and backward passes, and for two representative dataset sizes, $N = 1,000$ and $N = 1,000,000$.

**Smaller scale** ($N = 1,000$). At this moderate sample size, DC loss is measurably slower than NLL/MSE but still lightweight in absolute terms: forward passes are on the order of $7 \times 10^{-4}$ s, versus $3 \times 10^{-5}$ s for NLL/MSE; backward passes show a similar gap.

Table 1: Timing results (seconds) for $N = 1,000$ (mean $\pm$ std).

| Noise | Pass | Loss | Mean | Std |
|-------|------|------|------|-----|
| Poisson | Forward | DC | 0.000749 | 0.000476 |
| | | NLL | 0.000032 | 0.000176 |
| | Backward | DC | 0.000237 | 0.000426 |
| | | NLL | 0.000042 | 0.000201 |
| Gaussian | Forward | DC | 0.000704 | 0.000478 |
| | | MSE | 0.000030 | 0.000171 |
| | Backward | DC | 0.000266 | 0.000443 |
| | | MSE | 0.000026 | 0.000159 |

**Larger scale** ($N = 1,000,000$). At one million samples, the gap widens: DC loss forward passes are about half a second, reflecting per-sample CDF/logit evaluations and sorting for the Wasserstein term, whereas NLL/MSE remain in the millisecond range. Backward passes for DC are faster than forwards but still notably slower than NLL/MSE.

Table 2: Timing results (seconds) for $N = 1,000,000$ (mean $\pm$ std).

| Noise | Pass | Loss | Mean | Std |
|-------|------|------|------|-----|
| Poisson | Forward | DC | 0.503176 | 0.152972 |
| | | NLL | 0.002166 | 0.001515 |
| | Backward | DC | 0.042866 | 0.022514 |
| | | NLL | 0.001780 | 0.001442 |
| Gaussian | Forward | DC | 0.500350 | 0.154613 |
| | | MSE | 0.000825 | 0.000608 |
| | Backward | DC | 0.031831 | 0.017249 |
| | | MSE | 0.000965 | 0.000960 |

**Context** Solving inverse problems with modern unsupervised methods typically involves costly regularizers (e.g.neural network evaluations) and forward operator evaluations (e.g. tomographic projection). For context, the subsetted 3D tomography operator in Section 5.2.2 required approximately 0.012 s per evaluation, DC loss on the same problem required 0.017 s per evaluation, and a single reverse-diffusion step with the architecture and methodology of (Singh et al., 2024) required 1.05 s.

DC loss therefore introduced a noticeable runtime overhead relative to NLL/MSE, but its cost remained comparable to other common components in modern inverse-problem pipelines.

**Summary.** DC loss carries a runtime overhead versus NLL/MSE. The overhead is small at $N = 1{,}000$, but becomes more noticeable at $N = 1{,}000{,}000$, driven chiefly by distribution evaluations and sorting in the Wasserstein step. It may be possible to reduce this overhead with approximation strategies and GPU-optimized kernels for DC loss at scale.

## I   LLM USAGE

Large-language models (LLMs) were used to sanity check ideas, polish writing and assist with converting code to algorithm pseudocode.

