# OpenReview forum: "Distributional Consistency Loss: Beyond Pointwise Data Terms in Inverse Problems"
_ICLR.cc/2026/Conference — ICLR 2026 Poster_

### Official Review · Reviewer_3iUP · 2025-10-20

**Soundness:** 3
**Presentation:** 3
**Contribution:** 2
**Rating:** 4
**Confidence:** 4

**Summary:**

The paper introduces a new 'distributional consistency' data fidelity term that can be used for solving inverse problems. The loss cheks whether the prediction residuals follow the noise distribution by applying a variant of the probability integral transform (similar to the Von Mises statistic). Experiments on single image denoising using the deep image prior and on PET reconstruction with Poisson noise show that the proposed loss performs better than the standard negative log-likelihood setting.

**Strengths:**

- presents an interesting data fidelity term for inverse problems that exploits full knowledge of the noise distribution by checking that residuals are consistent at a distributional level.
- the proposed loss avoids over-fitting on the deep image prior and improves performance on PET (regularized) reconstruction.
- presents an analysis that motivates links with the standard negative log-likelihood loss

**Weaknesses:**

- the paper lacks a discussion of existing literature on the distributional loss, and other existing alternatives to negative log-likelihood. For example, the loss seems quite similar to the Von Mises statistic, which is well-known in statistics. It would be also good to include discussions on other related work, like robust estimators (eg Huber losses, etc).

- There is no guarantee that the proposed loss should avoid overfitting in general problems, except for the empirical evidence on the two presented 'toy' examples, which involde a single image at a time.
   - I would expect at least one experiment on learning on a dataset or its usage with modern solvers such as plug-and-play, diffusion methods, etc.
   - I find sentences like "it avoids overfitting to measurement noise even without the use of priors" problematic, since there is no theoretical guarantee that this is the case - it is easy to construct counter-examples with low distributional loss and high error (as already done in the paper in 3.2) - and the results rely on 'implicit regularizations' such as the deep image prior autoencoder architecture or the reduced number of voxels in PET.

**Questions:**

I would expect the loss to be highly sensitive to mispecified or approximate noise models - did you check the proposed method on real measurement data (not synthetically generated measurements)?

---

> ### Author Response · Authors · 2025-11-21
> **Response to Reviewer 3iUP (Part 1 of 2)**
>
> # Response to Reviewer 3iUP (Part 1 of 2)
>
> We thank the reviewer for their thoughtful and constructive review.
>
> We believe we have now obtained new results and have provided comprehensive clarifications for the questions raised, which we address point-by-point below:
>
> ---
>
> ## 1. Relation to existing distributional tests and data-fidelity literature:
>
> We agree that DC loss sits naturally alongside classical goodness-of-fit tests such as Kolmogorov–Smirnov and Cramér–von Mises that rely on the Probability Integral Transform (PIT). In response to the reviewer’s comment, we have expanded the related-work section (6.1) accordingly.
>
> Conceptually, DC can be viewed as an optimal-transport analogue of these statistics: instead of $L^\infty$ or $L^2$  discrepancies used in K-S/CvM, we use the Wasserstein-1 distance on logit-transformed PIT values. This yields a smooth, differentiable, optimization-friendly loss that retains the core idea of assessing whether PIT-transformed residuals follow the expected uniform distribution.
>
> We also now discuss robust losses such as Huber and Student’s-t (Section 6.1). These methods address model misspecification or outliers, whereas DC assumes a well-specified noise model and tests for distributional consistency, which is a complementary but distinct goal.
>
> ## 2. On overfitting and what DC loss guarantees:
>
> The reviewer’s comment here reflects a misunderstanding of the specific guarantee we claim, and hence we appreciate the chance to clarify here and in the revised paper.
>
> When we say DC “avoids overfitting to measurement noise,” we refer to the classical sense of tracking the particular noise realisation, i.e. minimizing $|Ax-m|$. Under this definition, an estimate cannot simultaneously overfit and have a low DC loss. Low DC loss implies the residuals behave like random draws from the assumed noise distribution, and not a deterministic fit to the exact noise sample. These two conditions are mutually exclusive, which is why we believe our original wording was accurate.
>
> This behaviour is a direct consequence of the PIT, which we now explain more prominently in Sections 3.1–3.2 to complement the formal treatment we provided in Appendices A.1 and B.
>
> That said, the reviewer is absolutely right that avoiding noise-overfitting does not imply zero reconstruction error. As Section 3.2 illustrates, noisy inverse problems are ill-posed, so multiple reconstructions can be statistically consistent with the noise model.
> Thus a solution may have low (even zero) DC loss yet non-zero error; this reflects ambiguity, not overfitting.
>
> In our example, even the “worst-case” low-DC solution remains within a bounded error of the ground truth, and this bound shrinks to zero as the noise level decreases. This reflects DC’s role: it identifies the set of data-consistent solutions, typically within one noise realization of the maximum likelihood estimate (see Figure 2), but does not select among them. That selection is determined by explicit or implicit regularization.
>
> In short:
> - DC loss theoretically prevents fitting the specific noise sample once statistical consistency is reached.
> - DC does not guarantee minimal reconstruction error, because many solutions may be equally consistent.
> - This behaviour is distinct from MSE/NLL, which continue to reward per-sample noise fitting and therefore exhibit classic late-iteration overfitting.
>
> ---
> Our response continues in Part 2 of 2 (due to the character limit on individual comments).

---

> > ### Author Response · Authors · 2025-11-21
> > **Response to Reviewer 3iUP (Part 2 of 2)**
> >
> > # Response to Reviewer 3iUP (Part 2 of 2)
> >
> > ---
> >
> > ## 3. Scope of evaluation and modern solvers
> >
> > Following the reviewer’s suggestion, we have now added a plug-and-play experiment (Appendix G) using a learned denoiser.  The behaviour we observed mirrors our DIP results: DC loss improves stability and noise–detail trade-offs compared with MSE loss.
> >
> > We have also clarified the intended scope of our DC loss (Section 6.2): it is designed for noise-dominated inverse problems with known noise models, but is not expected to deliver benefits for inverse problems with ill-posed operators and minimal noise (e.g. inpainting).
> >
> > ---
> >
> > ## 4. Noise-model mismatch and real data
> >
> > In response to your question, we have now carried out and included results for a real 3D PET reconstruction (Section 5.2.2 & Appendix F.5), to demonstrate that the theoretical advances in the paper carry over to real-world problems. PET acquisition pipelines explicitly estimate and calibrate key components of the noise model, making it a representative real-world testbed. Our results suggest it is feasible to apply DC loss in this setting.
> >
> > We agree that sensitivity to mismatch is important. In response, we have added new experiments (Appendix E.5) introducing correlated noise, outliers, and heteroscedastic variance. We find that DC loss is robust to moderate mismatches and performs comparably to MSE loss, degrading gracefully as the noise model mismatch becomes more severe.
> >
> > ---
> > ---
> >
> > ## Summary:
> >
> > **We thank the reviewer for the helpful and constructive feedback. In response, we have expanded the related-work discussion, clarified the applicable scope for our method, and added new experiments including plug-and-play, real PET data, and noise-model mismatch. These revisions address the reviewer’s concerns and strengthen both the scope and positioning of the paper.**

---

### Official Review · Reviewer_AQeQ · 2025-10-26

**Soundness:** 4
**Presentation:** 3
**Contribution:** 3
**Rating:** 6
**Confidence:** 4

**Summary:**

This paper proposes the Distributional Consistency (DC) Loss, a new data-fidelity objective for imaging inverse problems that goes beyond traditional pointwise losses such as MSE or negative log-likelihood. Instead of enforcing pixelwise accuracy, DC loss measures whether the residuals are statistically consistent with the assumed noise distribution. The method applies the Probability Integral Transform (PIT) to map residuals into a uniform space and computes the Wasserstein distance (after a logit transform) between the resulting distribution and a standard logistic reference. Experiments on Deep Image Prior (DIP) denoising and Poisson PET reconstruction show that DC loss effectively mitigates overfitting and improves robustness without relying on early stopping or manual regularization tuning.

**Strengths:**

- The DC loss provides a principled alternative to traditional pixelwise data terms by enforcing consistency at the distributional level. The use of PIT and Wasserstein distance forms a coherent, probabilistically interpretable framework.
- Experiments on DIP and PET reconstruction show that DC loss mitigates overfitting and achieves a better balance between noise suppression and structural preservation, illustrating strong potential for broader applications.

**Weaknesses:**

- The scope of evaluation is limited. The paper validates the proposed DC loss only on two tasks—DIP-based image denoising and Poisson PET reconstruction. While these are representative, it remains unclear whether the method generalizes to other inverse problems (e.g., deblurring, inpainting, or diffusion-based approaches) or to more complex, learned priors.

- The core concept is not immediately intuitive. The key theoretical basis—the Probability Integral Transform (PIT)—is crucial for understanding why the CDF-based approach produces uniform residuals under a correct model. However, this important explanation is only presented in the supplementary material, leaving readers unfamiliar with the concept confused at first glance. As written, the method appears pointwise rather than distributional, and the main text would benefit from a concise, self-contained explanation or visual illustration of how PIT ensures distributional consistency.

- The discussion of related work is insufficient. Those who are experts in imaging inverse problems might not necessarily experts in the design of data-fidelity terms. Thus, I find it surprising that the paper does not discuss any closely related works or precedents. This lack of context makes it difficult to judge whether the proposed formulation is genuinely new or an adaptation of existing statistical consistency ideas.

- The approach has practical limitations:
  - Computational cost is higher than standard MSE or NLL losses due to the sorting and Wasserstein distance computation required per iteration.
  - The method assumes known and independent noise models for each measurement, which may not hold in real scenarios involving correlated, unknown, or spatially varying noise.
  - The sensitivity to noise model mismatch is not studied, leaving uncertainty about the robustness of the approach in practical imaging systems.

**Questions:**

Please see the weakness section.

---

> ### Author Response · Authors · 2025-11-21
> **Response to Reviewer AQeQ (Part 1 of 2)**
>
> ## Response to Reviewer AQeQ  (Part 1 of 2)
>
> We thank the reviewer very much for the thoughtful and constructive review, and for highlighting the strength of our novel framework.
>
> We now have new results and clarifications to address the questions raised, which we cover point-by-point below:
>
> ---
>
> ## 1. Evaluation scope:
>
> We agree that demonstrating broader applicability is important. In addition to the DIP-based denoising and PET image reconstruction experiments in the main paper, the revised manuscript now highlights:
> - New Appendix G: results using a plug-and-play learned prior for image deblurring/denoising. In this modern learned-regularization setting, we again observe the same pattern as in DIP: DC loss provides improved stability (this time with respect to hyperparameters rather than iteration number) and better noise-detail trade-offs compared with MSE loss.
> - New PET real data: we give results for real 3D PET reconstruction, demonstrating the applicability of our framework to a real world problem.
> - Appendix D: a toy example on 1D noisy deconvolution, illustrating that the DC loss is not limited to imaging.
>
> As a result of these changes, we now show DC loss for Gaussian and Poisson noise models, and for diverse tasks including tomographic image reconstruction, denoising, deblurring and deconvolution.
>
> We also clarified the intended scope of applicability (Abstract & Section 6.2): DC loss is most beneficial in noise-dominated inverse problems with known noise models and many independent measurements. This is an important class of problems that includes PET/SPECT (nuclear medicine), photon-counting CT, astronomical imaging, and low-light imaging, among others. Conversely, we do not expect benefits for inverse problems where operator ill-posedness is the primary difficulty to overcome (e.g., inpainting), and we now make this explicit.
>
> ---
>
> ## 2. Explaining the PIT intuitively:
>
> We thank the reviewer for highlighting that the intuition behind PIT should appear earlier in the paper. In response, we have added a concise explanation of the PIT into Section 3.1, restructuring the order of ideas in this section, and clarified the explanation given in the caption for Figure 1.
>
> ---
>
> ## 3. Related work:
>
> In response to this point, we have expanded both the introduction (Section 1, paragraph 3) and related-work section (Section 6.1) to clarify both the novelty of our formulation and its relationship to existing statistical consistency ideas.
>
> Our approach is, to the best of our knowledge, the first to use distributional calibration with a known noise model as a differentiable data-fidelity term for inverse problems. At the same time, DC loss relates strongly to classical goodness-of-fit tests that rely on the probability integral transform, most notably the Kolmogorov–Smirnov (K–S) statistic and the Cramér–von Mises (CvM) criterion. These methods evaluate whether residuals, transformed through their model CDFs, follow the expected uniform distribution.
>
> The key distinction lies in how deviations from uniformity are quantified:
> - K–S uses an $L^\infty$ discrepancy;
> - CvM uses an $L^2$ discrepancy;
> - DC loss uses the Wasserstein-1 (optimal transport) distance in logit space.
>
> This optimal-transport view is important, because it yields a smooth, well-behaved objective that is suitable for optimization instead of hypothesis testing.
>
> We now additionally cite the related work of Llacer et al. [1], which proposes similar ideas of distributional consistency for emission tomography, albeit as a post-hoc assessment tool (similar to K-S or CvM).
>
> We also clarify how this relates to robust losses such as Huber and Student’s-t (Section 1, paragraph 3 & Section 6.1). These methods address model misspecification or outliers, whereas our focus is on distributional correctness under a well-specified noise model. They therefore address complementary concerns, and we now discuss this explicitly.
>
> ---
>
> Our response continues in Part 2 of 2 (due to the character limit on individual comments).

---

> > ### Author Response · Authors · 2025-11-21
> > **Response to Reviewer AQeQ (Part 2 of 2)**
> >
> > # Response to Reviewer AQeQ (Part 2 of 2)
> >
> > ---
> >
> > ## 4. Practical limitations:
> >
> > ### Computational cost:
> >
> > For a PET reconstruction with real 3D tomographic data, we find that using NLL takes 0.037 seconds per iterative update, and using DC takes 0.057 seconds per iterative update (Section 5.2.2 / Appendix F.5). While we acknowledge that our method is slower than calculating a likelihood, we do not consider the relative timing a prohibitive limitation for practical use.
> >
> > The reviewer is correct that DC loss introduces overhead from sorting and CDF evaluation. In practice, however, the overall runtime of most inverse problem solvers is dominated by regularization term evaluations or forward operator projections. We have now updated Appendix H with direct timing comparisons showing that the loss computation is a modest component of end-to-end runtime. For example, one reverse diffusion step with the methodology of Singh *et al.* [2] in 3D takes 1.05s, or 18x longer than one step of DC-Adam.
> >
> > ### Sensitivity to noise mismatch:
> >
> > Following the reviewer’s suggestion, we added new mismatch experiments (Appendix E.5) covering outliers, correlated noise, and heteroscedastic variance. In the settings studied, DC loss proved tolerant to moderate mismatch and behaves comparably to MSE loss, degrading gracefully as the noise model mismatch became more severe.
> >
> > We also included a real PET experiment (Section 5.2.2 & Appendix F.5), demonstrating that DC loss performs stably using a practical acquisition model.
> >
> > ---
> > ---
> >
> > ## Summary:
> > **We thank the reviewer again for the helpful feedback. In response to the comments raised, we have redrafted Sections 1, 3 and 6 of the manuscript and expanded the scope of evaluation to include additional application areas and real data results.**
> >
> > ---
> >
> > ### References:
> > [1] Llacer, J., Veklerov, E. & Nunez, J. (1989) Statistically Based Image Reconstruction For Emission Tomography. *International Journal Of Imaging Systems And Technology 1*:132--148.
> >
> > [2] Singh, I.R.D., Denker, A., Barbano, R., Kereta, Ž., Jin, B., Thielemans, K., Maass, P. & Arridge, S. (2024) Score-Based Generative Models for PET Image Reconstruction. *Machine Learning for Biomedical Imaging 2 (Special Issue for Generative Models)*:547–585.

---

### Official Review · Reviewer_KXqD · 2025-10-30

**Soundness:** 3
**Presentation:** 4
**Contribution:** 3
**Rating:** 8
**Confidence:** 4

**Summary:**

The paper introduces a loss function for inverse problems, based on matching the distribution of residual errors with a known noise distribution.

**Strengths:**

- Well written paper.

- The overall idea of matching the noise distribution to derive a loss function is very interesting.

- Results are given for several settings.

**Weaknesses:**

- The claim that the noise model is known or can be estimated in many practical settings seems overly optimistic. In practice, there can be many additional artifacts that might not be well captured by noise terms and/or are difficult to estimate/model. In the case of tomography for example, real-world data might include artifacts caused by not well calibrated or defective detector pixels (ring artifacts), isolated high-intensity values caused by stray X-rays, scatter components that are caused by non-linear interactions with the sample, parts of the data replaced by nonsensical values (caused for example by the X-ray beam going down during an experiment). It is unclear, but important, to assess how robust the proposed method is to such artifacts. In other words, it would be useful to know how well the method works when the noise model used does not match the data (in various degrees of severity).

- The authors now use the proposed loss in combination with Deep Learning based methods. It would be interesting to see its performance for traditional image reconstruction (e.g. reconstructing the pixel values of an image directly), comparing with traditional reconstruction methods.

- There is quite a bit of work in the inverse problem community about appropriate loss functions. Examples include the student's t loss [1] and the Huber loss [2]. Such works should be discussed in the introduction, and ideally compared with.

- Similarly, there is quite a bit of knowledge about the effect of noise on the convergence of iterative reconstruction methods in inverse problems [3]. Based on this, several stopping criteria have been developed [4]. It would be good to discuss this prior work, and to compare directly with such stopping criteria (as you assume known noise distribution).

[1] Kazantsev, D., Bleichrodt, F., van Leeuwen, T., Kaestner, A., Withers, P. J., Batenburg, K. J., & Lee, P. D. (2017). A novel tomographic reconstruction method based on the robust Student's t function for suppressing data outliers. IEEE Transactions on Computational Imaging, 3(4), 682-693.

[2] Mohan, K. A., Venkatakrishnan, S. V., Gibbs, J. W., Gulsoy, E. B., Xiao, X., De Graef, M., ... & Bouman, C. A. (2015). TIMBIR: A method for time-space reconstruction from interlaced views. IEEE Transactions on Computational Imaging, 1(2), 96-111.

[3] Elfving, T., Hansen, P. C., & Nikazad, T. (2014). Semi-convergence properties of Kaczmarz’s method. Inverse problems, 30(5), 055007.

[4] Hansen, P. C., Jørgensen, J. S., & Rasmussen, P. W. (2021, September). Stopping rules for algebraic iterative reconstruction methods in computed tomography. In 2021 21st International Conference on Computational Science and Its Applications (ICCSA) (pp. 60-70). IEEE.

**Questions:**

- How does the method perform when the noise model used does not match the data?

- How does the method perform in combination with classical image reconstruction?

- How does the proposed method compare with existing custom loss functions used in inverse problems?

- How does the method compare with existing work based on using known noise models, for example as a stopping criterion?

---

> ### Author Response · Authors · 2025-11-21
> **Response to Reviewer KXqD (Part 1 of 2)**
>
> # Response to Reviewer KXqD (Part 1 of 2)
>
> We thank the reviewer very much for their thoughtful and positive review. The comments helped us strengthen both the scope and positioning of the paper.
>
>
> We address each of the questions posed point-by-point below.
>
> ---
>
> ## 1. How does the method perform when the noise model used does not match the data?
>
> We agree that in some settings, real measurements often contain artefacts beyond idealized noise assumptions. In response to the reviewer’s question, we have added new experiments in Appendix E.5 introducing controlled mismatches between the assumed and true noise models, including outliers, correlated noise and heteroscedastic variance.
>
> Across these settings, we find that DC degrades gracefully compared to MSE, demonstrating tolerance to moderate mismatch. As expected, both losses break down under extreme mismatch.
>
> To complement these synthetic studies, we also added a real data 3D PET image reconstruction experiment (Section 5.2.2). PET scanners use carefully calibrated physical models that explicitly model defective detectors, normalization issues, attenuation, scatter and randoms, thereby preserving the pure Poisson noise model and making them a natural testbed for DC loss. In this setting, we demonstrate that DC loss is a feasible choice of optimization function for real data reconstruction.
>
> ---
>
> ## 2. How does the method perform in combination with classical image reconstruction?
>
> We thank the reviewer for raising this point. Our PET image reconstruction experiments already implement classical reconstruction: we directly optimize voxel values to explain tomographic measurements, either using the Poisson NLL or the DC loss. There is no learned prior or network in this experiment.
>
> ---
>
> ## 3. How does the proposed method compare with existing custom loss functions used in inverse problems?
>
> We appreciate this suggestion, and in response we have now added discussion of these methods in both the introduction and the related-work section. Robust losses target model misspecification or outliers, whereas DC loss addresses the complementary question of distributional calibration under a known noise model.
>
> ---
>
> Our response continues in Part 2 of 2 (due to the character limit).

---

> > ### Author Response · Authors · 2025-11-21
> > **Response to Reviewer KXqD (Part 2 of 2)**
> >
> > # Response to Reviewer KXqD (Part 2 of 2)
> >
> > ---
> >
> > ## 4. How does the method compare with existing work based on using known noise models, for example as a stopping criterion?
> >
> > We thank the reviewer for highlighting this connection. We have expanded the discussion (Section 6.1) to relate DC loss to classical work on semi-convergence and stopping-rule design.
> >
> > Conceptually, stopping criteria suppress late-iteration noise amplification by terminating optimization once the data term begins to overfit noise; as the cited paper Elfving et al. [3] shows, such early iterations can provide useful implicit regularization, particularly for algorithms like Kaczmarz. In our PET experiments, we observe a similar phenomenon: DC-Adam reaches an error comparable to the optimally early-stopped NLL-Adam, and both are slightly outperformed by the implicit bias of MLEM.
> >
> > However, DC loss provides a fundamentally different mechanism. Instead of preventing overfitting by halting the optimization process before it has achieved its objective, DC removes the incentive to fit individual noise fluctuations: once residuals are statistically consistent with the noise model, the data term flattens and the regularizer (implicit or explicit) becomes the dominant driver. This makes DC a continuous calibration target rather than a discrete stopping rule, and it interacts synergistically with regularization (as demonstrated in the DIP and TV-regularized PET experiments, Figures 5d & 8a). DC also evaluates the distributional calibration of an estimate with respect to the noise model, which is something that early stopping does not enforce. For example, Figure 7(b) shows that MLEM attains low error yet does not achieve distributionally consistent residuals.
> >
> > Early stopping also has known limitations in tomography, most notably spatially variant convergence rates and the difficulty of selecting iteration numbers that behave uniformly across datasets. In contrast, DC yields stable late-iteration behaviour without manual intervention: once the uniform-residual condition is reached, further optimization does not chase noise.
> >
> > Empirically, this distinction is visible in our experiments:
> > - DIP: DC loss achieves a higher peak PSNR than DIP-MSE **even when DIP-MSE is optimally early-stopped** (Figures 5(c)–(d)).
> > - PET: DC-Adam avoids the semi-convergent behaviour of NLL-Adam and converges to stable reconstructions at high iteration counts (Figure 7).
> >
> > Figure 2 provides intuition: stopping rules seek a balance between fidelity and regularization, while DC loss broadens the region of “data-consistent” solutions by enforcing distributional calibration rather than pointwise matching. This can enable algorithms to find data consistent solutions with greater fidelity than the implicit regularization of early stopped iterative solvers.
> >
> > While MSE and NLL also assume known noise models, we are not aware of prior work that uses the noise model to define a distributional fidelity term. The empirical advantages we observe arise not from a “free lunch,” but from exploiting the large number of measurements typical in inverse problems to enforce distributional agreement rather than pointwise error reduction.
> >
> > ---
> > ---
> >
> > ## Summary:
> >
> > **We thank the reviewer once again for the constructive and encouraging review. The suggestions made have prompted us to make several meaningful improvements: we demonstrated the practicality of the method with real data, we added robustness experiments with noise-model mismatch, and expanded the related-work discussion around robust losses. These revisions have helped sharpen the scope and presentation of the paper, and we are grateful for the reviewer’s insights.**

---

### Official Review · Reviewer_W1WB · 2025-11-01

**Soundness:** 2
**Presentation:** 2
**Contribution:** 2
**Rating:** 2
**Confidence:** 4

**Summary:**

This paper introduces a novel data-fidelity loss function, the Distributional Consistency (DC) loss, designed for inverse problems. The DC loss evaluates whether the entire set of observed measurements is statistically consistent with the noise distributions implied by the current signal estimate. The authors demonstrate that this method avoids overfitting, eliminates the need for early stopping in methods like Deep Image Prior (DIP) , and improves the efficacy of regularization in tasks like PET image reconstruction.

**Strengths:**

1. The paper provides a comprehensive statistical analysis of the proposed DC loss.

2. The idea of exploring distributional-level calibration as an alternative to data-fidelity loss is meaningful.

**Weaknesses:**

1. The method's core premise requires that the measurement noise distribution is known or can be well-estimated. The experiments are confined to purely synthetic data using clean Gaussian or Poisson noise models. This ideal scenario is rarely met in practice, where noise is often complex, correlated, or misspecified. The paper provides no experiments on real-world data, making it impossible to assess the method's robustness or practical generalization ability.

2. The paper makes a very broad claim to be a "performance-enhancing alternative... for inverse problems", yet the experimental validation is narrow. It is limited to only two applications: image denoisin and PET reconstruction. Furthermore, the PET experiments are entirely simulated using a Brain Web phantom. This limited testing on "toy problems" and synthetic data is insufficient to support the paper's general claims.

3. Follow my previous comments, another major limitation is the study's exclusive focus on unsupervised regularization methods like Deep Image Prior (DIP), a 2017 work. It does not demonstrate its effect in any state-of-the-art, end-to-end trained frameworks. Without even a proof-of-concept on generative methods or deep unrolling networks (which is way more common to use compared with DIP), the claim that this loss is a convincing alternative for modern inverse problems is unsupported and feels overclaimed.

4. The practical usage of DC loss is questionable due to its computational overhead. The method requires computing a CDF for every measurement and, more importantly, sorting the entire set of $N$ transformed values at each iteration to compute the Wasserstein-1 distance.

5. The paper's mathematical notation is inconsistent and confusing. For example: the unknown parameter to be estimated is introduced as $\theta$ in the general formulation (Equation 1). In the PET application, this parameter is abruptly changed to $x$ (Equation 8) without any connection. Most critically, Equation 8 introduces a term $b$ (in $[Ax+b]$) which is never defined anywhere in the paper.

**Questions:**

Same as the weakness part.

---

> ### Author Response · Authors · 2025-11-21
> **Response to Reviewer W1WB (Part 1 of 2)**
>
> # Response to Reviewer W1WB (Part 1 of 2)
>
> We thank the reviewer for their considered review, and for noting that our idea is novel and meaningful.
>
> We believe that we have strong answers for the questions raised, which we address point-by-point below:
>
> ---
>
> ## 1. Known measurement noise:
>
> While the reviewer is entirely correct that many inverse problems do not have known noise models, there is however a significant subset of important inverse problems (for example, nuclear medicine single photon imaging, positron emission tomography, photon-counting CT, astronomical imaging, low-light imaging, among others) where the noise model is physically well known or can be empirically estimated to high accuracy. It is precisely in these cases where DC loss can bring benefits, as our experiments demonstrate. We have now amended the abstract to claim applicability to an “important class of unsupervised noise-dominated inverse problems”, rather than all inverse problems.
>
> In response to the reviewer’s point about experiments on real-world data, we have further demonstrated that our method has real-world applicability by performing reconstruction from real 3D PET brain scan data with our proposed DC loss. These new results are now included in Section 5.2.2, and conclusively demonstrate the feasibility of using DC loss for a real-world inverse problem.
> Our real 3D PET data was obtained from a Siemens Biograph mMR scanner with Poisson-noisy measurements from > 70 million tomographic lines of response. As with many medical imaging setups, the acquisition process is extensively modelled, including attenuation, detector normalization, and scatter and randoms estimation. This is necessary for standard reconstruction with likelihood objectives, which also assume a Poisson noise model.
>
> In addition, to assess the robustness of the known noise assumption, in Appendix E.5 we added new synthetic experiments with a mismatch between the true and assumed noise distribution, introducing outliers, correlated noise, and heteroscedastic variance. In the settings we studied, we found that the performance of DC loss degrades gracefully compared to MSE loss.
>
> ---
>
> ## 2. Strength of claims:
>
> We have now clarified the claim highlighted in the abstract to better match the scope of our contribution, which we stated in Section 6.2. As indicated in our point #1 above, we now claim applicability to an “important class of unsupervised noise-dominated inverse problems”, rather than all inverse problems.
>
> Our set of experiments now includes Gaussian and Poisson noise models, and applications such as image denoising (Sec 5.1) and deblurring (App. G), 1D deconvolution (App. D), and tomographic reconstruction (Sec 5.2) now with both synthetic and real data. These experiments serve to validate the theoretical properties of our method.
>
> As a result of the reviewer’s suggestions, we therefore now have a greater body of empirical evidence to support a tighter set of claims.
>
> ---
>
> ## 3. Focus on unsupervised reconstruction:
>
> **On state-of-the-art approaches:** we have now added an example application demonstrating image deblurring and denoising with a state-of-the-art plug-and-play (PnP) prior in Appendix G. This is an example of a state-of-the-art reconstruction approach with a learned prior. Our results in this case align with our findings for DIP, namely that using DC loss with regularization outperforms using MSE loss with regularization. We additionally find that our method is much less sensitive to the choice of PnP’s key hyperparameter, $\rho$.
>
> The addition of Appendix G greatly strengthens our claim of applicability to modern inverse problem solving.
>
> **On end-to-end trained approaches:** our work doesn’t apply to this setting, and so we make no claims in this direction. When ample training data is available, an end-to-end pipeline can be trained by directly penalizing deviation between an estimate and the ground truth in image space, and so a data fidelity term is not required (hence not needing our specific contribution of the DC loss). However, for many inverse problems of interest, ground truth data is not available (e.g. medical imaging or astronomy problems, among others). This motivates the use of unsupervised regularization methods (such as DIP or PnP) in the literature, and it is this setting that is covered in the paper.
>
> As mentioned in our response #2, we have now reduced the breadth of the claim of applicability to modern inverse problem solving to avoid confusion. This more refined scope and clearer claim has now made clear where DC loss is applicable and useful, as well as where it doesn’t apply.
>
> ---
>
> Our response continues in Part 2 of 2 (due to the character limit).

---

> > ### Author Response · Authors · 2025-11-21
> > **Response to Reviewer W1WB (Part 2 of 2)**
> >
> > # Response to Reviewer W1WB (Part 2 of 2)
> >
> > ---
> >
> > ## 4. Computational overhead:
> >
> > We show DC loss’s use for a real reconstruction pipeline for PET data, resulting in the following timings:
> >
> > - NLL-Adam reconstruction: 0.037 sec/it
> > - DC-Adam reconstruction: 0.057 sec/it
> >
> > While we agree that the DC loss introduces overhead, and this overhead is dominated by the sorting operation, we do not find in practice that this overhead is prohibitive or questionable.
> >
> > Additionally, modern iterative inverse problem-solving approaches usually involve costly neural function evaluations or forward operator evaluations. Compared to these operations, DC loss is a modest component of the inverse-problem-solving pipeline, as we now highlight in Appendix H (“Context section”). For example, one reverse diffusion step with the methodology of Singh et al. [1] in 3D takes 1.05s, or 18x longer than one step of our DC-Adam approach.
> >
> > ---
> >
> > ## 5. Mathematical notation:
> > Thank you for pointing out this inconsistency. We have fully addressed this in our revised manuscript, which  now uses consistent notation throughout for clarity.
> >
> > ---
> > ---
> >
> > ## Summary:
> > **In response to the reviewer’s questions, we have narrowed the scope of our claims and demonstrated the use of our method with practical real-world 3D data. The paper has been strengthened further as a result. We hope these extensive revisions address the reviewer’s concerns.**
> >
> >
> >
> > ---
> >
> > ### References:
> > [1] Singh, I.R.D., Denker, A., Barbano, R., Kereta, Ž., Jin, B., Thielemans, K., Maass, P. & Arridge, S. (2024) Score-Based Generative Models for PET Image Reconstruction. *Machine Learning for Biomedical Imaging 2 (Special Issue for Generative Models)*:547–585.

---

> ### Comment · Reviewer_W1WB · 2025-11-28
> **Response to the rebuttal of Submission8871**
>
> I thank the authors for the rebuttal. The clarified scope and additional real-world experiments have successfully addressed concerns 1, 2, 4, and 5.
>
> Regarding concern 3, since the scope has shifted to unsupervised imaging inverse problems, the authors should discuss and compare their method against established unsupervised baselines. For instance, SURE-based methods [1] and Noise2Noise [2] are fundamental to this domain. Furthermore, in the specific context of deep unrolling, recent self-supervised approaches such as SSDU [3] and SPICER [4] should be discussed in the Background or Introduction.
>
> Additionally, the Introduction and Background sections require rewriting as the current background lacks sufficient detail. The authors need to formally introduce the literature on unsupervised imaging inverse problems to explicitly state the research gap and better motivate the proposed approach.
>
> As the authors have addressed most of my concerns, I will raise my score to 4. If these remaining concerns are addressed, I am open to further increasing my score. Thank you.
>
> [1] Metzler, C. A., Mousavi, A., & Baraniuk, R. G. (2018). Unsupervised learning with Stein's unbiased risk estimator. arXiv preprint arXiv:1805.10531.
>
> [2] Lehtinen, J., Munkberg, J., Hasselgren, J., Laine, S., Karras, T., Aittala, M., & Aila, T. (2018). Noise2Noise: Learning image restoration without clean data. In Proceedings of the 35th International Conference on Machine Learning (pp. 2965–2974). PMLR.
>
> [3] Yaman, B., Hosseini, S. A. H., Moeller, S., Ellermann, J., Uğurbil, K., & Akçakaya, M. (2020). Self-supervision via data undersampling for physics-guided deep learning reconstruction. Magnetic Resonance in Medicine, 84(4), 2107–2120.
>
> [4] Hu, Y., Gan, W., Ying, C., Wang, T., Eldeniz, C., Liu, J., Chen, Y., & Kamilov, U. S. (2024). SPICER: Self-supervised learning for MRI with automatic coil sensitivity estimation and reconstruction. Magnetic Resonance in Medicine, 91(6), 2321–2336.

---

> > ### Author Response · Authors · 2025-12-02
> > **Second response to Reviewer W1WB**
> >
> > We thank the reviewer for their constructive feedback and for raising their score to 4.
> >
> > The reviewer’s feedback has convinced us that the Introduction and Background do indeed need revision to clarify the scope and more clearly situate our contribution. **In the newly revised manuscript, these sections have now been restructured, the discussion of related approaches has been front-loaded, and the research gap we are addressing is now stated explicitly at the outset.**
> >
> > Regarding the remaining reviewer concern, we believe this very valid concern arises from two different uses of the term “unsupervised”.
> >
> > In our paper, we use the term “unsupervised” for image reconstruction in the sense of the classical inverse-problems context: an algorithm operates solely on the measurements of the dataset to reconstruct from (i.e. a single test dataset), with strictly no paired measurement/ground-truth data and no measurement corpus for training. Clean images may be used as priors or regularizers (e.g. as with plug-and-play or diffusion model methods), but there are no datasets containing measurements involved.
> >
> > Rather than this classical unsupervised inverse-problem context, the reviewer’s concern relates instead to the broader and equally common use of the term “unsupervised” in the self-supervised learning literature. In this setting, a model is trained on a dataset of corrupted measurements without access to clean targets. Under this interpretation, the reviewer’s cited methods are natural representatives of that setting. The assumptions that these methods employ are however different from those we study in our work, for the following important reasons:
> > - **Noise2Noise [2]** requires multiple independent corrupted measurements of each underlying signal. In contrast, our inverse-problems setting only has access to a single noisy measurement.
> > - **SSDU [3]** addresses undersampled MRI and assumes access to a dataset of many corrupted k-space measurements. Again, in contrast, our setting concerns having only one single noisy observation with no training corpus available.
> > - **SPICER [4]** similarly requires multiple corrupted acquisitions of the same object and is specific to parallel MRI with unknown coil sensitivities.
> > - **SURE-based methods [1]** are the closest in spirit to our approach and our problem context, and therefore we now cite and discuss these [1] in the revised manuscript (and we thank the reviewer for drawing our attention to these approaches). These methods stabilise the reconstruction by estimating risk and implicitly constraining the estimator itself through the divergence term, which reflects the architecture’s sensitivity to noise. In this sense, SURE acts chiefly as a model-side constraint driven by the chosen parametrization. Our DC loss takes a complementary route: it does not regularise the architecture but instead uses the known noise model to enforce distributional consistency of the residuals, which is the regime of interest in our noise-dominated applications.
> >
> > The majority of the cited approaches operate in regimes where training datasets of corrupted measurements are available. In contrast our work studies reconstructions from a single noisy instance without any paired training corpus (not that this requirement does not exclude the use of a learned image functional or pre-trained generative model, as in many state-of-the-art regularization approaches). As a result, the cited methods are only tangentially related to our work and problem context. **Thanks to the reviewer’s pointing out of these apparently very related methods, we have now made this distinction explicit in the revised manuscript (in Section 2.2), to avoid ambiguity around terminology and expected baselines.**
> >
> > **We really hope this helps to resolve the remaining reviewer concern and that the revised manuscript now more accurately positions our contribution within the inverse-problems context. Thanks to the clarifications and extensions already made following the reviewer’s initial round of feedback, we trust our manuscript now addresses all major concerns and presents a clearer and better-supported account of our proposed approach.**

---

### Author Response · Authors · 2025-11-12
**Reproducibility**

As mentioned in the paper, an anonymous repo for reproducibility is provided at the following link: https://github.com/Anonymous-Ocelot/DC-Loss

I want to thank the reviewers for their considered reviews; we are working hard on our replies now and will post them as soon as possible.

---

### Author Response · Authors · 2025-12-02
**Summary comment for AC**

Given the unexpected reset of the discussion phase, we wanted to compose a concise summary of the clarifications and additions made during the discussion.

---

## Real-world practicality of assuming the noise distribution is known

Several reviewers questioned whether our proposed DC loss applies in real-world scenarios: whether it is actually practical to assume that the noise model is known. We clarified that many inverse problems do in fact fall into exactly this regime (e.g., medical imaging – the Poisson model is known and trusted for raw PET and SPECT projection data, and a Poisson–Gaussian noise model is known to be appropriate for low-light imaging). We added new material to support this:

- Section 5.2.2: real-data 3D PET reconstruction under a clinical acquisition model.
- Appendix E.5: noise-mismatch ablations showing that our proposed DC loss degrades gracefully and comparably to standard data fidelity losses when the assumed noise model is imperfect.

Reviewer W1WB noted that these additions “successfully addressed” their concerns on this point.

---

## Related work and positioning of our contribution

All reviewers asked for broader contextualization. We therefore moved our related work section to the front of the paper (Section 2.2), and expanded it to cover robust losses, early-stopping and semi-convergence, self-supervised and risk-estimation methods (e.g., SURE, Noise2Noise), and classical goodness-of-fit tests (Kolmogorov–Smirnov, Cramér–von Mises).

Following reviewer AQeQ’s suggestion, we also clarified that DC loss can be viewed as a directly optimizable analogue of these goodness-of-fit tests (by replacing their non-differentiable metric with a differentiable optimal transport metric that allows direct optimization).

---

## Scope and clarity improvements

We refined the abstract and introduction to specify that our DC loss targets noise-dominated inverse problems from a single instance of measured data, where the measured data is composed of many independent measurements and has a known (or well-estimated) noise distribution. We moved the explanation of the probability integral transform into the main method section.

---

## Expanded experimental validation

Beyond deep image prior (DIP) and simulated PET, we also now include:

- real 3D PET reconstruction (Section 5.2.2),
- plug-and-play reconstruction with a learned prior (Appendix G),
- 1D deconvolution (Appendix D).

These results demonstrate that the behavior observed in DIP and PET is not specific to those two initial examples.

---

## Reviewer assessments

Reviewer KXqD (8) was positive about the idea and presentation. Their questions on robustness, robust losses, and early-stopping have now been addressed through our new mismatch experiments and expanded related work.

Reviewer AQeQ (6) also evaluated the work favorably. Their requests for broader empirical evidence, clearer explanation of the probability integral transform (PIT), and improved positioning have now been addressed with the new experiments and restructuring of Sections 2 and 3.

Reviewer 3iUP (4) raised questions about our use of the term “overfitting.” We clarified that our proposed DC loss prevents fitting the specific noise sample once residuals are statistically consistent, and therefore does theoretically prevent overfitting, but that this does not guarantee minimal error. The reviewer’s requests for discussion of robust estimators and a modern-solver experiment were all addressed via our expansion of the related work and adding a new plug-and-play example respectively.

Reviewer W1WB (2 → 4) was the only reviewer able to reply before discussion was unexpectedly frozen. They wrote that our rebuttal had “successfully addressed” their concerns about the known-noise assumption, breadth of claims, computational overhead, and notation, and indicated openness to raising their score still further. Their one remaining concern was the need for a clearer introduction and positioning of our method relative to current literature, which we addressed by rewriting those sections.

---

## Restating our core contribution

Our proposed DC loss provides a distributional data-fidelity term that evaluates whether residuals match the assumed noise distribution. This reframing:
- prevents late-iteration noise chasing,
- aligns naturally with regularization rather than opposing it,
- can be interpreted as an optimization-friendly analogue of classical goodness-of-fit tests,
- integrates cleanly with existing regularization frameworks.

We hope this summary helps contextualize the discussion and the resulting revisions. Thank you for your time and consideration.

---

### Meta-Review · Area_Chair_5tfL · 2025-12-27

**Summary:**

This paper introduces a novel data-fidelity loss for inverse problems that operates at the distributional level rather than enforcing pointwise agreement, leading to improved stability during optimization. The authors demonstrate that the proposed loss significantly improves performance in Deep Image Prior (DIP) denoising on natural images and in PET image reconstruction under Poisson noise.

Reviewers raised concerns regarding presentation clarity, the limited scope of experiments, additional computational overhead, the reliance on a known noise distribution, and the absence of certain baseline comparisons. During the rebuttal, the authors adequately addressed the technical questions, added new experiments, incorporated additional baselines, and clarified the scope and limitations of the method. Given the demonstrated novelty and the strengthened experimental section, the overall evidence now appears sufficient.

That said, I recommend acceptance provided the authors slightly weaken or rephrase the claim that DC loss avoids overfitting whereas standard losses cannot in the main paper. In the DIP setting, the noise level is assumed known, and baseline losses such as MSE can also mitigate noise overfitting when appropriately tuned to exploit this information (e.g., via noise-aware loss or related methods). Since the baselines do not utilize the known noise level, the current comparison may overstate this aspect of the advantage. A more precise framing would emphasize that DC loss avoids overfitting without requiring explicit noise-level–based heuristics, rather than implying that such behavior is unique to the proposed loss.

**Reviewer Concerns:**

Reviewers raised concerns regarding presentation clarity, the limited scope of experiments, additional computational overhead, the reliance on a known noise distribution, and the absence of certain baseline comparisons.

During the rebuttal, the authors substantially improved the presentation and clarified several technical points. In particular, they demonstrated that the additional computational cost is relatively modest, and explicitly acknowledged that assuming a known noise distribution is a strong requirement, while providing evidence that performance degrades mildly when this assumption is violated. The authors also added new experiments and included additional baseline comparisons. Overall, the previously raised concerns have been adequately addressed, with no major outstanding technical issues remaining.

**Reviewer Scores:**

Reviewer KXqD (score: 8) was positive about the paper. Given that the initial score is already high, a further increase in the score is unlikely.

Reviewer AQeQ (score: 6) was generally positive. However, since one of the main concerns involves an intrinsic limitation of the proposed method, namely, the requirement that the noise distribution be known, the reviewer is unlikely to increase the score to 8.

Reviewer 3iUP (score: 4) requested additional discussion of related work and further comparisons with plug-and-play solvers. As these points were addressed during the rebuttal, I would expect the score to increase to 5 or possibly 6.

Reviewer W1WB (initial score: 2 → 4) responded during the rebuttal and explicitly stated that the score would be raised to 4.

---

### Decision · Program_Chairs · 2026-01-26

Accept (Poster)